# Max Planck Institute Earth System Model (MPI-ESM1.2) for High-Resolution Model Intercomparison Project (HighResMIP)

Oliver Gutjahr[1], Dian Putrasahan[1], Katja Lohmann[1], Johann H. Jungclaus[1], Jin-Song von Storch[1], Nils Brüggemann[1,2], Helmuth Haak[1], and Achim Stössel[3]

[1]Max Planck Institute for Meteorology, Hamburg, Germany
[2]University of Hamburg, Hamburg, Germany
[3]Department of Oceanography, Texas A&M University, College Station, Texas, USA

**Correspondence:** O. Gutjahr (oliver.gutjahr@mpimet.mpg.de)

**Abstract.** As a contribution towards improving the climate mean state of the atmosphere and the ocean in Earth System Models (ESMs), we compare several coupled simulations conducted with the Max Planck Institute for Meteorology Earth System Model (MPI-ESM1.2) following the HighResMIP protocol. Our simulations allow to analyse the separate effects of increasing the horizontal resolution of the ocean ($0.4°$ to $0.1°$) and atmosphere (T127 to T255) submodels, and the effects of

substituting the Pacanowski and Philander (PP) vertical ocean mixing scheme with the K-Profile Parameterization (KPP).

The results show clearly distinguishable effects from all three factors. The high resolution in the ocean removes biases in the ocean interior and in the atmosphere. This leads to the important conclusion that a high-resolution ocean has a major impact on the mean state of the ocean and the atmosphere. The T255 atmosphere reduces the surface wind stress and improves ocean mixed layer depths in both hemisphere. The reduced wind forcing, in turn, slows the Antarctic Circumpolar Current (ACC)

reducing it to observed values. In the North Atlantic, however, the reduced surface wind causes a weakening of the subpolar gyre and thus a slowing down of the Atlantic Meridional Overturning Circulation (AMOC), when the PP scheme is used. The KPP scheme, on the other hand, causes stronger open-ocean convection which spins up the subpolar gyres, ultimately leading to a stronger and stable AMOC, even when coupled to the T255 atmosphere, thus retaining all the positive effects of a higher resolved atmosphere.

*Copyright statement.* TEXT

## 1 Introduction

The evolving computational power allows for ever higher resolutions of earth system models (ESM). High resolution ESMs are able to explicitly resolve processes that are subgrid-scale and parameterized in low-resolution models. Optimally, better resolved processes would improve atmosphere and ocean dynamics and thus reduce biases in the mean state and in the variability

of key quantities. In this manuscript, we separately increase the horizontal resolution of the atmosphere and ocean submodels and analyse the effects on the mean states. Besides increasing the resolution of the major model subcomponents, new strategies

and model developments, such as improved physics, are required for improving ESMs. Therefore, we also analyse the effects of a more sophisticated vertical mixing parameterization in the ocean submodel.

Specifically, this paper describes the adaptation of the Max Planck Institute - Earth System Model (MPI-ESM, Giorgetta et al., 2013) to higher horizontal resolutions and the implementation of improved ocean physics within the PRIMAVERA project (https://www.primavera-h2020.eu/). A key aspect of the project is on improving the simulation of the European climate, which is why we put a focus on the North Atlantic and the Atlantic Meridional Overturning circulation (AMOC). We investigate separately the effects of increasing horizontal resolution of the atmosphere and the ocean, and of exchanging the vertical mixing parameterization in the ocean and sea ice submodel MPIOM (Jungclaus et al., 2013).

All our simulations follow the High Resolution Model Intercomparison Project (HighResMIP) protocol (Haarsma et al., 2016) and provide climate simulations with varying horizontal resolutions that are higher than the standard resolution of the Coupled Model Intercomparison Project - Phase 6 (CMIP6; Eyring et al., 2016). An overview of all performed simulations for this study is shown in Tab. 1.

Our reference model is the MPI-ESM1.2-HR (or HR in the remainder of the manuscript) that was recently described by Müller et al. (2018) and contributes to CMIP6. HR is the higher resolution version of the former MPI-ESM1.2-LR (or LR), with 1.5 times as high (T127, $\sim 100\,\mathrm{km}$) horizontal resolution for the atmospheric submodel ECHAM6.3 (Hertwig et al., 2015; Mauritsen et al., 2018) and a $0.4\,^\circ (\sim 40\,\mathrm{km})$ ocean on an eddy-permitting tripolar grid (TP04) (Jungclaus et al., 2013) compared to the LR version (T63, $\sim 200\,\mathrm{km}$ atmosphere and $1.5\,^\circ$ ocean grid). How the ocean and atmosphere mean states improve in HR compared to LR was described by Müller et al. (2018).

Further reductions of atmospheric biases were shown by Hertwig et al. (2015), who used ECHAM6.3 with a T255 ($\sim 50\,\mathrm{km}$) resolution in atmospheric model intercomparison project (AMIP) type experiments. Building on these improvements, we further use a coupled MPI-ESM1.2 version with the T255 atmosphere and the TP04 ocean grid (MPI-ESM1.2-XR or simply XR) to investigate the effect of an increased atmospheric resolution on the mean state. This XR version was already used by Putrasahan et al. (2019) and (although under a different acronym) by Milinski et al. (2016). Milinski et al. (2016) demonstrated that the sea surface temperature bias in the upwelling regions along the coast of Africa diminished because of a more detailed representation of the coastal winds with the T255 atmosphere. Although biases were reduced with a T255 version of ECHAM6.3, our XR simulation generally produces too weak surface wind speeds, in particular over the North Atlantic and the subpolar gyre (Putrasahan et al., 2019).

These weaker near-surface winds caused a slowdown of the Atlantic Meridional Overturning Circulation (AMOC) to about $9\,\mathrm{Sv}\ (\mathrm{Sv} := 10^6\,\mathrm{m}^3\,\mathrm{s}^{-1})$, as documented by Putrasahan et al. (2019). This issue was not only affecting the MPI-ESM1.2, but was also reported by other modelling centres using ECHAM6, although going from T63 to T127 (Sein et al., 2018). Sein et al. (2018) gave a possible explanation for the reduction of mean wind speeds, which they attribute to a higher cyclone activity with the T127 resolution, in particular over the North Atlantic.

The AMOC strength and its stability depend to a large extend on the vertical mixing parameterization (Gent, 2018). To investigate the sensitivity of the AMOC and the mean states, we conducted parallel experiments with HR and XR in which the modified parameterization of Pacanowski and Philander (1981) (PP), which is default in MPI-ESM1.2 (Marsland et al., 2003),

was replaced by the more sophisticated K-Profile Parameterization (KPP) scheme of Large et al. (1994). It turned out that the KPP scheme compensates for the underestimated mean winds in the high latitudes and in the tropics in the XR simulation, sustaining a stable AMOC. The reasons for this will be elaborated upon.

Finally, we adopt the $0.1°(\sim 10\,\text{km})$ tripolar grid (TP6M) of MPIOM, which was already used in an ocean-only simulation forced by NCEP, and in a coupled run with T63 and T255 versions of ECHAM6 – the so-called STORM simulations (von Storch et al., 2012; Stössel et al., 2015, 2018). With this high-resolution, mostly eddy-resolving coupled version (MPI-ESM1.2-ER or ER), we detect noticeable reductions of biases not only in the ocean and near-surface atmosphere, but also in the higher atmosphere. This leads to the important conclusion that high resolution in the ocean has a major impact on the large-scale temperature distribution in the atmosphere, consistent with recent findings (Frenger et al., 2013; Ma et al., 2016; Liu et al., 2018). The parallel simulations allow to separately analyse (1) the effects of increased atmospheric resolution (HR vs. XR), (2) the effects of increased ocean model resolution (HR vs. ER), and (3) the effect of an alternative vertical ocean mixing parameterizations (PP vs. KPP) on the mean climate.

We begin by describing the model configuration and spin-up procedure in section 2. In section 3 we present the results of the atmospheric mean state, including a description of reduced wind stress in XR. In section 4 we show the results of the ocean mean state, including the consequences of the reduced wind stress and how the KPP scheme sustains the AMOC. In section 5 we summarize all results and contrast the effects from increased resolution to improved ocean mixing.

## 2 Model, spin-up, and experiments

### 2.1 Model description

The atmospheric submodel of MPI-ESM1.2 is ECHAM6.3 (Mauritsen et al., 2018), which includes the land-surface scheme JSBACH (Stevens et al., 2013; Reick et al., 2013). The ocean and sea ice submodels are combined in MPIOM (Jungclaus et al., 2013; Notz et al., 2013). ECHAM6.3 and MPIOM are coupled via the Ocean-Atmosphere-Sea-Ice coupler version 3 (OASIS3-mct; Valcke, 2013) with a coupling frequency of 1 h. ECHAM6.3 was used with 95 vertical levels at two different spectral resolutions, truncated at T127 (~103 km) in HR and ER and T255 (~51 km) in XR. We did not change any parameter going from HR to XR, except for a reduction of the time step from 200 s (HR) to 90 s (XR) and the horizontal diffusion damping term. Both use the same eddy-permitting ocean with a resolution of 0.4°(~44 km) on a tripolar grid (TP04, Jungclaus et al., 2013) with 40 unevenly spaced vertical levels. The first 20 levels are distributed in the top 750 m. A partial grid cell formulation (Adcroft et al., 1997; Wolff et al., 1997) is used for a more accurate representation of the bottom topography. River runoff is calculated by a horizontal discharge model (Hagemann and Gates, 2003).

In the ER configuration, the ocean component has a nominal resolution of $0.1°$ (~10 km) on a tripolar grid (TP6M) (e.g. von Storch et al., 2012, 2016). The TP6M grid is quasi-uniform in the northern hemisphere and scales with latitude $(0.1°\cos(\phi))$ in the Southern hemisphere (von Storch et al., 2016). The grid size is thus smaller than the nominal resolution in the Southern Ocean (5 to 6 km near $60°S$) and about 2 to 3 km in the Weddell and Ross Sea. The TP6M grid resolves the major bulk of

mesoscale eddies, but not in the Arctic Ocean and the Nordic Seas, in the marginal seas of the subpolar North Atlantic, and not on continental shelves.

We did not change any parameters compared to the TP04 grid as prescribed by the HighResMIP protocol (Haarsma et al., 2016), except that we reduced the time step from 3600 s (TP04) to 240 s (TP6M). Table 1 provides an overview of the simula-

tions that we compared in this study. The HR configuration of our reference simulation is exactly the same as in Müller et al. (2018). The XR configuration was used by Hertwig et al. (2015) (denoted as VHR in their study) for AMIP simulations with ECHAM6 and in Milinski et al. (2016) (denoted as HRatm in their study) for MPI-ESM runs. The TP6M configuration was already used in stand-alone ocean simulations with 80 vertical levels (von Storch et al., 2012; Stössel et al., 2018), and in fully coupled simulations (e.g. Stössel et al., 2015).

All simulations (except ER) use the thickness diffusivity $\kappa_{GM}$ of the Gent et al. (1995) (GM) parameterization to account for the diffusion and tracer advection induced by unresolved mesoscale eddies in the ocean. For the TP04 grid, $\kappa_{GM}$ is constant and chosen to be proportional to the grid spacing. A value of $\kappa_{GM} = 250\,\mathrm{m^2\,s^{-1}}$ is chosen for a 400 km wide grid cell and it reduces linearly with increasing resolution. That is, for the eddy-permitting TP04 grid $\kappa_{GM}$ is only about 10 % of this value. We had two reasons for keeping this rather low GM coefficient compared to what is used by e.g. Marshall et al. (2017).

First, we want to be consistent with previous MPI-ESM simulations, and second, the GM coefficient was used to tune the AMOC, which became too weak in the TP04 configuration with higher values (von Storch et al., 2016). We note that finding an optimal configuration is challenging and still an open issue, in particular for grids that are a mixture of non-eddy-resolving, eddy-permitting and eddy-resolving as our tripolar grids. The TP04 grid is eddy-permitting, so that eddies are partly resolved and partly parameterized. Strictly, the same applies to the TP6M grid, which does not resolve eddies in all parts of the ocean,

but we decided to switch off GM in this configuration. There are also strategies to completely switch off GM already for eddy-permitting grids (Delworth et al., 2012).

The lateral eddy diffusivity is parameterized by an isopycnal formulation (Redi, 1982) and is set to $\kappa_{Redi} = 1000\,\mathrm{m^2\,s^{-1}}$ for a 400 km wide grid cell, again reducing linearly with increasing resolution. In ER, $\kappa_{GM}$ is set to zero, but $\kappa_{Redi}$ is unchanged (von Storch et al., 2016).

An innovation over previous versions of HR and XR is that we used two different diapycnal mixing schemes (see section 2.2): the PP scheme as default, and the KPP scheme. The diapycnal mixing scheme used in a simulation is indicated by subscripts: $HR_{pp}$, $HR_{kpp}$, $XR_{pp}$, $XR_{kpp}$, and $ER_{pp}$. Note that the model was not retuned when the KPP scheme was used, to account for the pure effect of a changed ocean mixed layer scheme. For all our comparisons, $HR_{pp}$ is our reference simulation.

We follow the HighResMIP protocol (Haarsma et al., 2016) for initialising and forcing our coupled control simulations. The

coupled runs used fixed 1950 forcing that consists of greenhouse gases, including ozone and aerosol loadings of the 1950s climatology ($\sim$ 10 year mean). The HR simulations were initialised from an HR control simulation that was nudged to the averaged state of 1950 to 1954 of the UK MetOffice Hadley Centre EN4 observational data set (version 4.2.0; Good et al. (2013)). The XR runs were initialised from the same ocean state, but from an atmospheric state that has been spun up for 10 years from a dry state. ER was initialised from the HR atmospheric state and directly from EN4 (averaged state from 1950-

54) for the ocean. We integrated the HR and XR control simulations for 150 years and the ER simulation for 80 years (see Tab 1). We cut off the first 30 years as spin-up and used the following 50 years from the control runs for the analysis.

## 2.2 Diapycnal mixing

Previous MPI-ESM versions used a modified version of the Richardson-number dependent formulation of Pacanowski and Philander (1981) (PP scheme). The modification of the original PP formulation consists of a parameterization for wind-induced mixing that decays exponentially with depth (Marsland et al., 2003). Convection is parameterized by enhanced eddy diffusivity ($k_v = 0.1 \, \mathrm{m^2 \, s^{-1}}$). For our simulations, we corrected a bug associated with the vertical viscosities, which were only about 50 % of the correct solution from the PP scheme. This error was then also corrected in the HR version described by Müller et al. (2018). The background value for the vertical diffusivity is constant and was set to $1.05 \cdot 10^{-5} \, \mathrm{m^2 \, s^{-1}}$ and to $5 \cdot 10^{-5} \, \mathrm{m^2 \, s^{-1}}$ for the viscosity. The background values represent the breaking of internal waves, which provide the mechanical energy for diapycnal mixing in the interior of the ocean. The PP scheme is the default option in MPI-ESM1.2 and is thus used in our reference simulation (HR$_{\mathrm{pp}}$).

To improve the diapycnal mixing in MPIOM, we implemented the non-local 'K-Profile parameterization' (KPP, Large et al., 1994). The KPP scheme was implemented by adding the Community Vertical Mixing (CVMix) project library (Griffies et al., 2013) to MPIOM. In the KPP scheme, the turbulent transports do not only depend on local gradients of the properties, but also on the overall state of the boundary layer, that is the surface fluxes and the boundary layer depth (Large et al., 1994). The non-local turbulent transport represents how surface properties are redistributed from the surface layer into the boundary layer, for example by buoyant plumes, Langmuir cells, or mesoscale cellular convective elements.

The non-local fluxes are non-zero only for tracers in unstable forcing conditions, i.e. for negative surface buoyancy fluxes. They then directly depend on the net heat and freshwater fluxes crossing the ocean surface multiplied by the local vertical diffusivities, a vertical shape function, and some constants (Griffies et al., 2013). For this non-local fluxes, the same vertical diffusivities are assumed as for the local tracer diffusion. In contrast to the PP scheme, these diffusivities are not limited to a user specified value, but depend on a depth-dependent turbulent vertical velocity scale, on a vertical shape function, and on the mixed layer depth (Griffies et al., 2013).

Below the mixed layer, we use the PP scheme with the same constant background diffusivity and viscosity. The diffusivities are not matched at the base of the mixed layer to avoid potential overshooting of the non-local transport terms, which might produce extrema in the tracer field (Griffies et al., 2013). Under sea ice, we reduce the wind-induced mixing in the PP and in the KPP scheme, so that the surface friction velocity $u_*$ decreases quadratically with increasing sea ice concentration. For simplicity, we neglect that the momentum flux from the atmosphere into the ocean could be even stronger when sea ice is present, because of additional momentum flux at the interface of sea ice and the underlying sea water.

## 3 Evaluation of the atmospheric mean state

For the evaluation, the MPI-ESM1.2 simulations were averaged over the first 50 model years after the spin-up. We used the ERA-Interim reanalysis data (Dee et al., 2011) averaged from 1979–2005 as reference for the atmospheric mean state, as HR was tuned to this period (Mauritsen et al., 2012).

### 3.1 Surface quantities

#### 3.1.1 10 m wind speed

The time-mean of the simulated 10 m scalar wind speed agrees well with ERA-Interim for large parts of the world's oceans and over the continents (Fig. 1). Consistently too low wind speeds, however, evolve over the northern parts of America and Europe, over South America, and over Greenland and Antarctica. Too strong winds are simulated by all model configurations over the subtropical oceans north and south of the equator. Models with the T127 atmosphere further simulate too strong winds speeds over the Weddell Sea (Fig. 1b,c,f).

Overall, the KPP scheme has only a minor effect on the 10 m wind speed. At the equatorial Pacific, KPP reverses the negative bias to a positive wind speed bias. Further, the negative bias in the Denmark and Fram Strait is reduced because of lower sea ice concentration in this area (see section 4.4).

Increasing the horizontal resolution from T127 to T255 in $XR_{pp}$ (Fig. 1d) introduces a negative wind speed bias over the Antarctic Circumpolar Current (ACC) because of a reduced meridional pressure gradient. The near-surface wind speeds are further too low over the subpolar gyre in the North Atlantic, and over the Nordic Seas. This reduced wind stress over the subpolar gyre causes a slowdown of the AMOC in $XR_{pp}$, as described in detail by Putrasahan et al. (2019).

By using the KPP scheme in the XR model ($XR_{kpp}$; Fig. 1e), the wind speed is too weak, but not as weak as in $XR_{pp}$. However, the wind speed is still lower over the Nordic Seas and in the Pacific sector of the ACC.

Increasing the horizontal resolution of MPIOM from $0.4°$ to $0.1°$ ($ER_{pp}$; Fig. 1f) reduces the positive bias over the Indian Ocean, over the Greenland Sea, and over the subtropical Atlantic. Despite these improvements, a high ocean resolution does have only a minor effect on the near-surface wind speed, when coupled to a rather coarse T127 atmospheric resolution.

#### 3.1.2 2 m temperature

In contrast to the near-surface wind speed, the 2 m temperature distribution (Fig. 2) is strongly affected by changing the horizontal resolution of the submodels or by replacing the vertical ocean mixing parameterization. Over the ocean, it closely resembles the bias of the sea surface temperature (section 4.1.1). Again, all models (except $XR_{pp}$) agree well with ERA-Interim over the continents and over large parts of the world's oceans, in particular over the tropical and subtropical oceans and in the Arctic Ocean.

An area with larger discrepancies across all models is the North Atlantic. Here, all simulations show a cold bias, which is a common error in state-of-the-art ESMs (Randall et al., 2007) that is mostly caused by a too zonal North Atlantic Current

(NAC) (Dengg et al., 1996), or by insufficient northward heat transport by the AMOC (Wang et al., 2014a). Drews et al. (2015) demonstrated that correcting the flow field removed the cold bias in the North Atlantic almost completely. Another area of cold near-surface air temperature biases is the region around the Antarctic peninsula. In contrast, all models (except $XR_{pp}$) simulate a consistent warm bias over the Canadian Archipelago, central Africa and central Asia. Although reduced in their magnitudes, all these biases remain in the higher resolution models or when KPP is used.

Our models with the T127 atmosphere (Fig. 2b,c,f) simulate a warm bias over the Weddell Sea, which is caused by too frequent open polynyas (see section 4.5). This warm bias vanishes or partly changes its sign in the western Weddell Sea, when increasing the atmospheric resolution to T255 in the XR models (Fig. 2d,e). This is because the frequency of open-ocean polynyas reduces (see section 4.5), so that the Weddell Sea is more often covered with thicker ice (not shown), causing colder near-surface temperatures. However, a severe cold bias develops over the North Atlantic and the Nordic Seas in $XR_{pp}$, as mentioned before. As a consequence, the temperatures over Europe decrease as well.

Using the KPP scheme in HR (Fig. 2c) results in warmer 2 m temperatures in the northern hemisphere, so that cold biases reduce, but warm biases become stronger. The reason is a stronger northward heat transport into the North Atlantic (see section 4) and thus a stronger heat release to the atmosphere. In $XR_{kpp}$ (Fig. 2e), the warming caused by the KPP scheme and the cooling caused by the T255 atmosphere compensate, so that the bias pattern in the northern hemisphere is comparable to that of $HR_{pp}$. The cold bias along the ACC, however, is not affected by KPP and is similar to $XR_{pp}$.

Compared to $HR_{pp}$, most of the cold biases vanish in $ER_{pp}$; in the region of the ACC, this is partly due to resolved eddies and improved mean flow. The warm bias in the Weddell Sea, however, is considerably enhanced in the Atlantic sector of the Southern Ocean, because of more frequent open-ocean polynyas in $ER_{pp}$.

## 3.2 Vertical structure of zonal wind speed and temperature

### 3.2.1 Zonal wind speed

Fig. 3 shows the ERA-Interim climatology of the time-averaged zonal-mean wind speed (u-velocity) and the model biases. Overall, the vertical structure of the zonal wind speed is well represented in MPI-ESM1.2. A consistent bias in all simulations are too strong subtropical jets (centred at $\sim 200\,\mathrm{hPa}$). These too strong jets contribute further to higher zonal wind speeds extending into the upper troposphere at 40 to $45\,°\mathrm{S}$ and 40 to $45\,°\mathrm{N}$, as also found by Müller et al. (2018). Furthermore too strong zonal wind speeds are simulated in troposphere in the tropics at roughly $400\,\mathrm{hPa}$.

All models simulate consistently too low zonal wind speeds over the Southern Ocean at $\sim 60\,°\mathrm{S}$ throughout the whole troposphere. The overall bias pattern in $HR_{kpp}$ (Fig. 3c) is very similar to to $HR_{pp}$ (Fig. 3b), although the bias in the over the Southern Ocean reduces and increases in the upper troposphere. The T255 atmosphere in the XR models amplifies all biases (Fig. 3d-e). That is, the subtropical jets become stronger and shift equatorwards and the zonal wind speed over the Southern Ocean reduces further.

Important for the ocean is the extension of the negative bias over the Southern Ocean down to the surface in both XR simulations (stronger in $XR_{pp}$ than in $XR_{kpp}$), which reduces the zonal wind stress driving the ACC and the upwelling of

Circumpolar Deep Water (CDW). However, this wind bias in XR was found not to be the cause of the AMOC collapse (Putrasahan et al., 2019). Note that the near-surface negative bias for the North Atlantic cannot be seen here, as discussed above, because it cancels in the zonal mean.

The bias pattern in $ER_{pp}$ (Fig. 3f) is similar to $HR_{pp}$ and $HR_{kpp}$, which indicates that the ocean resolution does not have a large impact on the mean zonal wind speed. However, both the positive bias in the subtropical jet in the northern hemisphere and the negative bias north of $60\,^{\circ}$N are slightly amplified.

### 3.2.2 Zonal temperature

The cross-sections of the global time-mean zonal-mean temperature (Fig. 4) show cold biases in the upper troposphere (at $\sim 250\,$hPa) in both hemispheres. In the HR/XR simulations with PP (Fig. 4b,d), the cold bias extends to the surface in both hemispheres (Fig. 4b,d). In $HR_{kpp}$, however, this bias disappeared (Fig. 4c), and emerges only weakly in $XR_{kpp}$ (Fig. 4e).

In $XR_{pp}$ the surface-extending cold bias becomes larger in the lower troposphere compared to $HR_{pp}$, because of the weaker AMOC and the freezing of the Labrador and Nordic Seas (see section 4 below). In contrast, the AMOC remains stable in $XR_{kpp}$ (Fig. 4e), so that no severe cold bias evolves in the lower troposphere of the northern hemisphere. However, the KPP scheme does not affect the cold bias in the southern hemisphere, as already found for the 2 m temperature. A clear improvement can be seen in $ER_{pp}$ (Fig. 4f), which removes both biases in the lower and middle troposphere in both hemispheres. We conclude that a high ocean resolution plays a major role for the mean-states of the large-scale temperature distribution in the atmosphere. Although the large cold bias above the Antarctic continent is present in all simulations, the bias is reduced in $ER_{pp}$ by about $2\,^{\circ}$C. The developing warm bias over the Weddell Sea in $ER_{pp}$ can also be seen in the cross-section at roughly $60\,^{\circ}$S.

## 4 Evaluation of the ocean mean state

### 4.1 Ocean surface temperature and salinity

#### 4.1.1 Sea surface temperature

The sea surface temperature bias of MPI-ESM1.2 with respect to the UK MetOffice EN4 data (version 4.2.0; Good et al. (2013), averaged from 1945–1955) is shown in Fig. 5. We used this period for EN4 since our HR simulations were initialised from a simulation that was nudged to the averaged EN4 state of 1950–54, and we further allow for some variance. The results differ only marginally if another period is chosen (not shown). In general, biases occur in prominent areas and are affected by both changing the model resolution and the vertical ocean mixing scheme.

All simulations (except $XR_{pp}$) simulate realistic sea surface temperatures in comparison to EN4 (Fig. 5). About 1 to $2\,^{\circ}$C colder sea surface temperatures than in EN4 are simulated in the northern hemisphere by $HR_{pp}$ (Fig. 5b). The strongest cold bias of up to $-7\,^{\circ}$C occurs in the North Atlantic between $40\,^{\circ}$N to $50\,^{\circ}$N, centred at about $30\,^{\circ}$W. A similar magnitude was described by Müller et al. (2018) for MPI-ESM1.2-HR. The main explanation for this cold bias, as given in section 3.1.2, is a too zonal NAC (Dengg et al., 1996), causing a too far southward intrusion of fresh and cold Labrador Sea water (Müller et al.,

2018) and insufficient northward heat transport by the AMOC (Wang et al., 2014a). Another reason could be too much export of Mediterranean water at about 1000 m depth (Fig. 8), thus leading to a too strong halocline that inhibits vertical mixing.

Too cold sea surface temperatures are further simulated along the ACC (bias of $\sim 2\,^\circ$C). Coastal upwelling areas west of Africa and South America are about 1 to $2\,^\circ$C too warm in all simulations with the T127 atmosphere (Fig. 5b,c,f), as found by
Milinski et al. (2016)

Increasing the atmospheric resolution from T127 to T255, while using the PP scheme (XR$_{pp}$), causes a severe cold bias in the whole northern hemisphere (Fig. 5d), strongest in the North Atlantic ($-9\,^\circ$C). This cooling was already described by Putrasahan et al. (2019) and is caused by a slowed AMOC due to weak wind stress over the subpolar gyre and weak northward heat and salt transports (Tab. 4, Fig. A1, and section 4.6). Although the reduced wind stress over the Southern Ocean (Fig. 1)
might also contribute to a weakening of the AMOC (Toggweiler and Samuels, 1995) in XR$_{pp}$, Putrasahan et al. (2019) found no effect of this negative wind bias on the AMOC slow down, and argue that the timescale of the slowing AMOC is much faster than any feedback from the Southern Ocean to the North Atlantic.

On the other hand, the biases in the coastal upwelling areas diminished to some extent, because of the better resolved coastal wind systems. This warm bias reduction in the upwelling areas is consistent with other studies (Putrasahan et al., 2013; Small
et al., 2015; Milinski et al., 2016). Furthermore, the Pacific cold-tongue almost disappears, but now the tropical Pacific becomes too warm south of the equator.

The cold bias in the North Atlantic diminishes drastically with the KPP scheme in HR$_{kpp}$ (Fig. 5c), but the warm bias in the Labrador Sea and in the Nordic Seas is enhanced because of an increased heat transport into the North Atlantic and its ambient seas (Fig. A1c). Moreover, a warm bias evolves in the tropical Pacific north and south of the equator. However, the KPP scheme
simulates a stable AMOC in XR$_{kpp}$ (Fig. 5e), because of a stronger subpolar gyre (see Tab. 2). The enhanced deep convection and North Atlantic Deep Water (NADW) formation in the Labrador Sea (section 4.5) sustains a strong enough upper cell of the AMOC (section 4.6) and thus a sufficient northward transport of heat and salt (see Tab. 4 and Fig. A1c-d). This surplus in heat and salt transports, compared to XR$_{pp}$, prevents the Labrador Sea from freezing over. This finding is an important result and provides a solution to the declining AMOC strength for MPI-ESM1.2-XR. In addition, enhanced upwelling in the Southern
Ocean further strengthens the northern cell of the AMOC (Marshall et al., 2017), although it is not the main reason in our model.

The cold bias along the ACC is clearly reduced in ER$_{pp}$ (Fig. 5f), because of resolving eddies that flatten and shift the outcropping isopycnals southwards. Furthermore, the cold biases in the North Atlantic, in the North Pacific, and in the Mediterranean Sea are reduced. The warm biases in the upwelling regions, however, remain because of the coarse T127 atmosphere.

**4.1.2 Sea surface salinity**

As with sea surface temperature, the sea surface salinity is well simulated by MPI-ESM1.2 for most parts of the ocean with respect to EN4 (Fig. 6). However, in some areas we find larger discrepancies. In the North Atlantic, the surface waters are too fresh where we already found a cold bias. This fresh bias is again caused by the too zonal NAC and the entrainment of fresher water masses from the Labrador Current. Although all models produce this bias, it is most pronounced in XR$_{pp}$, likely due to a

too stable stratification in association with excessive export of salty water from the Mediterranean (compare with Fig. 8d and Fig. A6d)

The fresh bias in the North Atlantic (Fig. 6c) diminishes with using the KPP scheme or the high-resolution ocean. In both cases a stronger northward salt transport is simulated in the Atlantic (Fig. A1d). In case of $ER_{pp}$, the Gulf Stream separation is better represented, which further reduces the bias in the North Atlantic (Fig. 6f). The resolved eddies further remove the fresh bias along the ACC. The water masses in the Mediterranean Sea become more saline, which removes the fresh bias that the HR and XR models produce.

Increasing the atmospheric resolution from T127 to T255 enhances the fresh bias in $XR_{pp}$ (Fig. 6d) because of the above described AMOC slow down, with the consequence that less salt is transported by the Gulf Stream and the NAC into the North Atlantic (Fig. A1c-d). In $XR_{kpp}$ (Fig. 6e), both effects work in opposite directions and almost balance each other, so that the bias is similar to that in $HR_{pp}$.

Another bias present in all simulations is a too saline near-surface Arctic Ocean, originating from the Siberian coast that extends across the Transpolar Drift, but also into the Canadian basin. These too saline waters indicate too little freshwater input from the Siberian rivers, in particularly from the Lena river (Laptev Sea). Another effect that enhances this error could be too little barotropic tidal mixing along the Arctic shelves and thus too little horizontal spreading of the river waters (Wang et al., 2014b).

Finally, a strong fresh bias is simulated in the western tropical Pacific. The KPP scheme does not ameliorate this problem as the surface waters become severely fresher in both XR simulations (Fig. 6d-e). In general, all models simulate too little precipitation or too much evaporation for most parts of the globe (Fig. A2). In the western Pacific, the XR models even simulate slightly less precipitation (Fig. A2d-e), so that we suspect that the supply of salty waters from the east is reduced in XR thus enhancing the fresh bias.

## 4.2 Ocean interior

Figure 7 shows the time-mean zonal-mean temperature bias of the MPI-ESM1.2 simulations to EN4 for the Atlantic and the Arctic Ocean. The bias of the HR and XR simulations are very similar and show a maximum warm bias at roughly $40\,°\,S$, continuing to $30\,°\,N$ at depths of the AAIW (about 800 to 1000 m). These biases are thought to be caused by erroneous interior circulation, tracer advection and mixing due to unrepresented eddy-induced tracer transports (Griffies et al., 2009; Jungclaus et al., 2013).

The warm bias at $40\,°\,S$ is related to enhanced advection of warm and salty waters from the Indian Ocean (Fig. A5 and Fig. A6), because the resolution is still too low to represent the Agulhas Current system (Jungclaus et al., 2013), with its retroflection and intermittent eddy shedding that transfers heat and salt into the Atlantic. The retroflection is not well present in HR/XR with the TP04 grid, so that a constant Agulhas leakage transports too warm and too salty water into the South Atlantic (Fig. A7). Neither the KPP scheme (Fig. 7c) nor the T255 atmosphere (Fig. 7d,e) reduces this warm bias. On the contrary, with the KPP scheme, the inflow becomes stronger so that more heat and salt is exchanged (Fig. A1a-b and Fig. A7b,d). The warm bias and the high salinity bias (Fig. 8) vanish only with the TP6M grid in $ER_{pp}$ (Fig. 7f), which is also clearly visible at

740 m depth (Fig. A5 and Fig. A6), because less warm and salty water from the Agulhas Current flow into the South Atlantic (Fig. A7e). This improvement was also reported by von Storch et al. (2016) for ocean-only simulations. There are two reasons for this warm bias reduction in $ER_{pp}$: (1) the Agulhas Return Current, Agulhas Retroflection and the Agulhas leakage are now better resolved, producing a more realistic circulation and water mass transfer from the Indian Ocean into the South Atlantic,

as seen in other similar studies (McClean et al., 2011; Putrasahan et al., 2016; Cheng et al., 2018); and (2) the eddy-induced cooling and freshening of the intermediate ocean (von Storch et al., 2016) further reduces the warm bias.

The warm bias in Fig. 7a-e stretches northward at the depth of the Antarctic Intermediate Water (AAIW) and shows another maximum at $30°$ N that is related to the spreading of Mediterranean waters. The HR and XR models use the same TP04 ocean grid and simulate both the observed net volume transport through the Strait of Gibraltar (net inflow of about $0.04$ Sv; see

Tab. 3). The outflowing Mediterranean water is too warm and too saline in all HR and XR simulations compared to EN4 (see Fig. A5 and Fig. A6), which explains the warm and saline bias (Fig. 8a-e). The Mediterranean water is slightly more saline in $HR_{kpp}$ than in $HR_{pp}$, so that the water spreading northward along the European continental shelf becomes also more saline and contributes to saltier Northeastern Atlantic Deep Water. This enhanced flow of saline water into the subpolar gyre explains the reduced salinity bias at 40 to $50°$ N at a depth of 1000-1500 m (Fig. 8c). The main spreading pathway in all HR and XR

models, however, is to the southwest into the open Atlantic.

As with the warm biases, the salinity biases disappear in $ER_{pp}$ (Fig. 7f and Fig. 8f). A fresher water mass at intermediate depth reflects a much more realistic representation of the AAIW (Fig. 8 and in detail in Fig. A8) and of the outflow of Mediterranean water. The latter is less saline and about 2 to $3°$ C colder (also shown at a depth of 740 m; Fig. A5f and Fig. A6f), reducing the warm and saline bias at $30°$ N. The reason for this major improvement is the better resolved bathymetry of the Strait of

Gibraltar, which is 12 km wide and has a sill depth of $\sim 300$ m in the present day real world. In the two ocean configurations discussed in this paper this Strait is about 24 km wide with a shallowest sill depth of about 230 m in the TP6M grid, compared to about 54 km and same sill depth in TP04. Although the salinity maximum of the overflow water is about 100 m shallower than in EN4 (not shown), $ER_{pp}$ produces more realistic properties of upper and intermediate depth water masses.

Although the Gulf Stream separates earlier from the American coast in $ER_{pp}$ (not shown), its flow path is still too zonal, such

that the cold bias in the North Atlantic at around $50°$ N (Fig. 7) is not removed. This indicates that a high-resolution ocean alone does not solve the cold bias at 740 m in the North Atlantic. In fact, Fig. A5f suggests that the cold bias is substantially larger in $ER_{pp}$ than in any of the other simulations.

The too warm and saline subpolar gyre causes a warm and saline bias in the deep convection areas of the Labrador and Irminger Seas, centred around $60°$ N (Fig. 7, Fig. 8, and Fig. A5). The bias is larger in $HR_{kpp}$ because of the increased transport

of heat and salt from the subtropical gyre into the subpolar gyre. The bias is reduced in the XR models because of the weaker subpolar gyre and the reduced salt transport by the gyre. However, from Fig. 8d, we see that the reduced salinity is the main factor causing the reduced convection in $XR_{pp}$ (also supported by Fig. A1d), as described by Putrasahan et al. (2019). Another contribution is too warm overflow waters from the Nordic Seas, an issue that was also present in coarser MPI-ESM versions (Jungclaus et al., 2013). This warm bias of the overflow waters is mostly unaffected in $ER_{pp}$.

The Atlantic water entering the Arctic Ocean ($0\,^{\circ}$C potential temperature bounds in Fig. 7a) is too warm and its layer is too thick in all HR and XR simulations (Fig. 7b-e), causing a warm bias within the Atlantic layer between 200 m to 1000 m. This is a common error in ocean general circulation models (Ilicak et al., 2016), which is thought to be caused by spurious numerical mixing of the advection operator (Holloway et al., 2007). Zhang and Steele (2007) further found a direct impact of the vertical mixing strength on the circulation of the Atlantic Water into the Arctic Ocean. Reducing the vertical mixing in the European Basin reduces the diffusion of the Atlantic Water and results in a thinner layer. By comparing the vertical mixing across all our simulations (Fig. A3) we see that $ER_{pp}$ simulates less vertical mixing in the Arctic Ocean at the depth of the Atlantic Water layer (as well as in the deeper layers of the Arctic Ocean and Atlantic), thereby readily removing the warm bias in the Atlantic Water layer. At 740 m depth, $XR_{pp}$ shows an even fresher Atlantic water layer throughout the Arctic Ocean and the GIN Sea (Fig. A6). Combined with the high salinity bias at the surface (Fig. 6d) in the Arctic Ocean, this implies a weakening of the Arctic halocline, also reflected by strong vertical mixing in the upper layers of the Arctic Ocean (Fig. A3c).

Further, less vertical mixing in the Fram Strait can reduce the inflow of Atlantic Water into the Arctic Ocean (Zhang and Steele, 2007) and thus reduce the warm bias as in $ER_{pp}$. In fact, Zhang and Steele (2007) recommend to reduce the background diffusivity to $1\cdot10^{-6}\,\mathrm{m^2\,s^{-1}}$ and viscosity to $1\cdot10^{-5}\,\mathrm{m^2\,s^{-1}}$. The background value for diffusivity is thus an order of magnitude lower than in our configuration. Sein et al. (2018) used an even lower background diffusivity in the Arctic Ocean of about $1\cdot10^{-6}\,\mathrm{m^2\,s^{-1}}$ in FESOM that is one order of magnitude lower than in the default version 1.4 (Wang et al., 2014b). However, our results show that a high resolution in the Arctic Ocean removes the warm and saline bias in the Atlantic Water layer, without changing any background values for vertical mixing. The benefit of a very high-resolution for the Arctic Ocean was recently demonstrated by Wang et al. (2018), who used a background diffusivity of $1\cdot10^{-5}\,\mathrm{m^2\,s^{-1}}$, which is close to what we chose.

## 4.3 Ocean circulation

To evaluate the large-scale ocean circulation, we compared barotropic volume transport stream functions of selected regions, transports through straits, and the AMOC. Overall we find three effects: (1) increasing the atmospheric resolution to T255 reduces the gyre strengths, (2) the KPP scheme enhances the strength of all gyres, and (3) the effect of a high-resolution ocean is bi-directional.

The simulated subpolar gyre strengths in the North Atlantic range from 31.0 to 40.6 Sv and are all within the observational range of 26.0 to 40.0 Sv (Tab. 2). $HR_{kpp}$ simulates a stronger subpolar gyre ($+6$ Sv) than the reference simulation $HR_{pp}$. Both $XR_{pp}$ and $XR_{kpp}$ show weaker gyres compared to their respective HR counterpart, whereas $ER_{pp}$ simulates a slight increase of the gyre strength.

The volume transport of the subtropical gyre in the North Atlantic, however, reacts more sensitively to the chosen vertical ocean mixing scheme and to resolving of ocean eddies. Compared to the reference of 48.2 Sv ($HR_{pp}$), the gyre strength decreases slightly to 44.0 Sv with a higher atmospheric resolution ($XR_{pp}$). By using the KPP scheme, however, the gyre strength increases to 64.9 Sv ($HR_{kpp}$) and remains similarly high with a T255 atmosphere ($XR_{kpp}$). $ER_{pp}$ produces a gyre strength as strong as with the KPP scheme. With that, the strength of the North Atlantic subtropical gyre of the KPP and ER simulations

slightly exceeds the bound of the observed range, while that of the PP simulations hovers around the other end of the observed range. In the case of $HR_{kpp}$ and $XR_{kpp}$ the too strong volume transport of the subtropical gyre might further contribute to the positive salinity bias in the subpolar gyre at a depth of 500 to 1000 m (Fig. 8 and Fig. A6). The results for the subtropical gyre of the North Pacific reveal a similar picture as in the North Atlantic with stronger transports in the KPP simulations. One exception is a markedly reduced gyre strength in $ER_{pp}$. Furthermore, all simulations produce a considerably stronger North Pacific gyre than what has been derived from observations

Tab. 3 summarizes the transports through important passages. The net volume transport through the Bering Strait is of the same magnitude (0.6 to 0.7 Sv) for $HR_{pp}$, $HR_{kpp}$ and $XR_{kpp}$, which is on the lower side of the observations (0.7 to 1.1 Sv). The transport is even lower (0.5 Sv) in $XR_{pp}$, which indicates a low exchange between the Arctic and the Pacific Ocean. Increasing the ocean resolution leads instead to a higher transport of 0.9 Sv in $ER_{pp}$. As with the improved outflow of Mediterranean Water through Strait of Gibraltar in $ER_{pp}$, this improvement is due to a better resolved Bering Strait.

The simulated net transport through Fram Strait is in the range of the observations ($-1.75 \pm 5.01$ Sv), which show a strong interannual variability (Fieg et al., 2010). A possible explanation for the somewhat lower transport with KPP is given by Zhang and Steele (2007). They found that strong vertical mixing, as with the KPP scheme in our HR and XR simulations, deepens the Atlantic Water layer, but simultaneously weakens the inflow of Atlantic Water and the outflow of Arctic Water.

In our KPP simulations, the outflow becomes weaker compared to the PP simulations, whereas the inflow is of similar magnitude, so that the net transport is lower. However, in comparison to the HR and XR simulations, the net transport in $ER_{pp}$ is only half on average. In agreement with Fieg et al. (2010), the West Spitsbergen Current (WSC) is better resolved in $ER_{pp}$. The WSC, and thus the inflow of Atlantic Water into the Arctic Ocean, is much stronger in $ER_{pp}$, as is its return circulation north of 80 °N. This intensified WSC and its recirculation cause a reduction of the net volume transport through Fram Strait. Considering the high uncertainty of the net transport from observations, all simulations give realistic estimates but the most realistic simulation with respect to the temperature and salinity structure and to the circulation is $ER_{pp}$ (not shown).

The overflows through Denmark Strait and across the Iceland-Scotland ridge are important deep water connections for the Arctic and the Atlantic. All simulations produce realistic overflow volumes with respect to observations, which are on average slightly higher in $ER_{pp}$, but still within the standard deviation of the coarser simulations. The higher transport in $HR_{kpp}$ versus $HR_{pp}$ is caused by enhanced deep convection in the Nordic Seas, particularly in the Greenland Sea (Fig. 10).

In all HR and XR simulations, the volume transport of the Florida Current is only about half the observed value of roughly 32 Sv (Tab. 3). Although the transport increases with the KPP scheme, only $ER_{pp}$ simulates a considerably (about 10 Sv) stronger transport, amounting to about 25 Sv. We found similar results for the Indonesian throughflow, which is important for climate because it connects the Pacific with the Indian Ocean and closes the upper warm branch of the MOC. Again KPP enhances the transports slightly, but only $ER_{pp}$ simulates a transport strength that is similar to observed values.

The Mozambique channel is an example where both a T255 atmosphere and KPP show a reduction in the transports. In $ER_{pp}$, however, the transport is about twice as high as in the other simulations and more realistic with respect to recent observations of $16.7 \pm 8.9$ Sv (Ridderinkhof et al., 2010). The ability to resolve eddies, particularly the Mozambique eddies along with a better

resolved southward advection through the Mozambique Channel, contributes to the more realistic transport of about $14\,\text{Sv}$ in $\text{ER}_{\text{pp}}$ (Putrasahan et al., 2016; Ridderinkhof et al., 2010).

The observed baroclinic transport through the Drake Passage was commonly estimated at roughly $140\,\text{Sv}$. However, a new estimate reveals a much higher transport volume of about $173.3 \pm 10.7\,\text{Sv}$, when adding the barotropic transport (Donohue et al., 2016). With regard to this estimate, the models are within or close to the observed range. However, compared to the reference simulation $\text{HR}_{\text{pp}}$ ($161.1\,\text{Sv}$) the transport weakens to about $150.0\,\text{Sv}$ in $\text{XR}_{\text{pp}}$, and from $191.9\,\text{Sv}$ in $\text{HR}_{\text{kpp}}$ to $170.3\,\text{Sv}$ in $\text{XR}_{\text{kpp}}$. In $\text{ER}_{\text{pp}}$ the transport is lower than in all other simulations (about $141\,\text{Sv}$). These results confirm that a higher atmospheric or ocean resolution reduces the transport in the Drake Passage, consistent to what has been found Stössel et al. (2015). In contrast, the transport through Drake passage is enhanced when using the KPP scheme, probably because of enhanced deep convection in the Weddell Sea (Fig. 11) that steepens the isopycnals across the ACC and thus increases the geostrophic flow (Stössel et al., 2015; Naughten et al., 2018) (see section 4.5.2).

## 4.4 Sea ice

### 4.4.1 Arctic Ocean

The spatial distribution of sea ice thickness (Fig. 9) agrees well with the PIOMAS reanalysis (averaged from 1979–2005) (Zhang and Rothrock, 2003; Schweiger et al., 2011) and is comparable to the MPI-ESM1.2-HR simulation described by Müller et al. (2018). The sea ice extent is in good agreement with the observations from the EUMETSAT OSI SAF (OSI-409-a; v1.2) product (averaged from 1979–2005) (EUMETSAT Ocean and Sea Ice Satellite Application, 2015), except for $\text{XR}_{\text{pp}}$ in which the Labrador Sea freezes over. In general, the maximum ice thickness (multi-year ice) in March is found along the north coast of Greenland and of the Canadian Archipelago, and reaches about $5\,\text{m}$ in PIOMAS but only $3\,\text{m}$ in $\text{HR}_{\text{pp}}$. The ice is slightly thicker in this area in the simulations with $\text{HR}_{\text{kpp}}$. In the Iceland Sea, $\text{HR}_{\text{kpp}}$ simulates less sea ice, which is in better agreement with the observations in that the ice cover does not reach as far south as Iceland as in $\text{HR}_{\text{pp}}$ (Fig. 9b). The enhanced northward heat transport into the Nordic Seas in $\text{HR}_{\text{kpp}}$ results in warmer sea surface temperatures there, leading to a northward shift of the winter ice edge. Further, a stronger recirculating branch of the West Spitsbergen Current in the Fram Strait (not shown) in $\text{HR}_{\text{kpp}}$ pushes the East Greenland Current westwards to the east coast of Greenland, thereby becoming narrower and faster, so that sea ice is constrained to a narrower band along the coast. In $\text{XR}_{\text{kpp}}$, however, the sea surface temperature is colder than in $\text{HR}_{\text{kpp}}$, so that the sea ice reaches Iceland as in the reference simulation. Compared to $\text{HR}_{\text{pp}}$, the sea ice thickness of $\text{HR}_{\text{kpp}}$ is slightly lower in the Eurasian Basin, although it becomes thicker in the Canadian Basin. $\text{XR}_{\text{pp}}$ (Fig. 9c) simulates more, although thin, sea ice in the Labrador Sea because of the above described fresher and colder North Atlantic and the resulting freeze-over. The sea ice cover in the Iceland Sea reaches even further south than in the reference simulation $\text{HR}_{\text{pp}}$. In contrast, in $\text{XR}_{\text{kpp}}$ (Fig. 9d) the ice thickness and extent in the Labrador Sea is similar to that in $\text{HR}_{\text{pp}}$. However, due to colder sea surface temperatures in the Denmark and Fram Strait than in $\text{HR}_{\text{kpp}}$, a southern tongue of sea ice extends to Iceland as in $\text{HR}_{\text{pp}}$. Further, in contrast to $\text{HR}_{\text{kpp}}$ the recirculating branch of the West Spitsbergen Current does not become stronger in the XR simulations (not shown).

In addition, the near-surface circulation in the Arctic Ocean changes with a T255 atmosphere from a more anticyclonic circulation in the Makarov and Canadian Basin in HR, to a more cyclonic circulation in XR (not shown). A cyclonic circulation enhances the export of cold Arctic Water via the East Greenland Current, causing colder sea surface temperatures in the Nordic Seas. The XR simulations and $ER_{pp}$ produce thinner winter ice in the Canada Basin, which may be related to the changed circulation, but has to be further investigated. $ER_{pp}$ produces in general a much lower sea ice volume the Arctic Ocean than the HR/XR simulations.

The extent of the Arctic summer ice cover in September is less and thus more realistic in the XR than in the HR simulations (not shown), in particular over the Siberian shelves, which is probably caused by the better resolved T255 atmosphere. KPP again simulates thinner ice in the Canada basin (about $-0.5\,\mathrm{m}$).

### 4.4.2 Southern Ocean

The spatial distribution of austral winter (September) sea ice thickness in the Southern Ocean of $HR_{pp}$ (not shown) is similar to the MPI-ESM1.2-HR simulations described by Müller et al. (2018). The ER and both HR simulations produce an overabundance of open-ocean polynyas in the Weddell Sea (see section 4.5.2). $HR_{kpp}$ simulates less and thinner ice in the Weddell Sea than $HR_{pp}$, but otherwise the spatial distribution of sea ice in the Southern Ocean is very similar.

Both XR simulations, but more so $XR_{pp}$, produce thicker sea ice than the other simulations, in particular in the Weddell Sea and close to Antarctica's coasts. The thicker ice in the Weddell Sea emerges in concert with a reduced number of polynyas, so that the warm bias seen in Fig. 2 vanishes. This less frequent occurrence of Weddell Sea polynyas is probably related to a reduced meridional pressure gradient across the Weddell Sea and the ACC (not shown), which in turn reduces the near-surface wind speed bias (as seen in Fig. 1). However, a more detailed investigation is required to explain circulation differences between the T127 and the T255 atmospheres over the Weddell Sea. In austral summer, both XR models produce thicker ice in the Weddell Sea (not shown), so that the ocean is insulated from the cold atmosphere above, resulting in less convective mixing.

## 4.5 Mixed layer depth and diapycnal mixing

### 4.5.1 Northern hemisphere

Fig. 10 shows the average mixed layer depths in March for the northern North Atlantic. We diagnosed the mixed layer depth as the depth where the density deviates from the surface density by $\sigma_t = 0.01\,\mathrm{kg\,m^{-3}}$. This diagnostic was computed from monthly means. As observations, we use the mixed layer depth retrieved from Argo floats by the density threshold method ($\sigma_t = 0.03\,\mathrm{kg\,m^{-3}}$) from the gridded $1° \times 1°$ monthly climatology (Jan 2000 to April 2018) from Holte et al. (2017). We interpolated the Argo mixed layer depths onto the TP04 grid. Missing values were filled by the nearest non-missing neighbour and values south of $60°$S and north of $80°$N were discarded and masked, because of the sparseness of Argo data below sea ice.

In the reference simulation $HR_{pp}$ (Fig. 10b), March-mean depths of up to $1500\,\mathrm{m}$ are simulated in the Labrador Sea, which is deeper than the observed $1200\,\mathrm{m}$ in March from Argo (Fig. 10a). The area with deep mixed layers wraps around southern

Greenland with depths up to $1000\,\text{m}$ south of Cape Farewell, in the Irminger Sea, and in the Nordic Seas. In the Irminger Sea, $\text{HR}_{\text{pp}}$ simulates too shallow mixed layers of only about $500\,\text{m}$ depth. Similarly, too shallow mixed layers are simulated in the Greenland Sea.

As discussed before, in $\text{XR}_{\text{pp}}$ (Fig. 10d) the deep convection in the Labrador Sea ceases within the first two decades of the simulation. This collapse of deep convection (together with that in the Nordic Seas) leads to a slowing down of the AMOC (Tab. 4) (Putrasahan et al., 2019).

The KPP scheme in $\text{HR}_{\text{kpp}}$ (Fig. 10c) causes much deeper mixed layers in the Labrador Sea and in the Greenland Sea. In particular the mixed layer depths in the Labrador and Irminger Sea and south of Greenland (north of $50\,^\circ\text{N}$) become deeper compared to all other simulations. These deeper mixed layers with the KPP scheme result on one hand from the convection parameterization (i.e. the non-local fluxes) and on the other hand from a stronger and more cyclonic subpolar gyre (Tab. 2) that domes the isopycnals in the gyre centres (not shown), which preconditions the water column for convection. As mentioned in section 2.2, the non-local fluxes in the KPP scheme have the same vertical diffusivities as for the local gradient transports. These diffusivities are not limited to a user-defined maximum value during convective forcing conditions, so that much larger diffusivities can act to redistribute temperature and salinity throughout the ocean water column, causing it to overturn faster and to produce deeper mixed layers with the KPP than with the PP scheme. We speculate that the non-local transport terms in KPP cause a more efficient convection than the enhanced wind-mixing parameterization of our PP scheme. The diffusivity in KPP is further enhanced as it also depends on the mixed layer depth, which reflects that boundary layer eddies become larger with deeper mixed layers.

On the other hand, $\text{XR}_{\text{kpp}}$ (Fig. 10e) simulates shallower mixed layers compared with $\text{HR}_{\text{kpp}}$. These shallower mixed layers result from the reduced wind stress of the T255 atmosphere by means of two processes: (1) less positive wind stress curl spins down the subpolar gyre, so that the slower cyclonic circulation reduces the isopycnal doming and the horizontal salt advection to the gyre centres (Tab. 4), leading to a more stratified surface layer; and (2) lower near-surface wind speeds reduce the turbulent air-sea fluxes via the bulk formula and the surface friction velocity ($u_*$). Lesser heat fluxes in turn reduce directly the non-local fluxes of the KPP scheme in convection areas, and lower $u_*$ reduces the turbulent vertical velocity scales, which results in lower vertical diffusivities and viscosities.

Based on these results, increasing the atmospheric resolution reduces the mixed layer depths over the North Atlantic and the Nordic Seas, whereas KPP deepens them. By combining both, the T255 atmosphere and the KPP scheme, the above effects compensate each other ($\text{XR}_{\text{kpp}}$; Fig. 10e). In contrast to $\text{XR}_{\text{pp}}$, where the convection ceases in the Labrador and GIN seas, the combination of T255 and KPP ($\text{XR}_{\text{kpp}}$) produces more realistic mixed layers depths even with reduced wind stress.

Overall, the KPP scheme modifies the large-scale circulation by simulating a stronger subpolar gyre, which in turn provides favourable conditions for deep convection in the Labrador Sea, Irminger Sea, and Nordic Seas. For this reason, $\text{HR}_{\text{kpp}}$ simulates enhanced deep convection compared with $\text{HR}_{\text{pp}}$, in particular in the Labrador and GIN Seas. In the Irminger Sea, mixed layer depths of about 400 to $500\,\text{m}$ are simulated by both $\text{HR}_{\text{kpp}}$ and $\text{XR}_{\text{kpp}}$, which is consistent with retrievals from observations (e.g. Pickart et al., 2003; Våge et al., 2008, 2011), although too shallow compared to Argo (Fig. 10a). One explanation for these too shallow mixed layers are that even the T255 atmosphere is too coarse to fully simulate Greenland tip jets (e.g.

Martin and Moore, 2007; DuVivier and Cassano, 2016; Gutjahr and Heinemann, 2018). The tip jets have a considerable impact on triggering deep convection in the Irminger Sea due to strong associated wind stress curls driving the Irminger Gyre, and turbulent fluxes of heat and momentum removing the near-surface stratification. Because of the unresolved tip jets the mixed layer depth may be underestimated in winters with high tip jet activity.

The mixed layer depths in the Labrador Sea are nevertheless too deep (excluding XR$_{pp}$). A possible explanation is the neglect of tidal mixing in MPI-ESM1.2. As shown by Müller et al. (2010), tidal mixing improves the recirculation of the Labrador Current. By entraining more freshwater into the surface layer of the Labrador Sea, it becomes more stratified which in turn reduces deep convection. Another shortcoming is probably insufficient eddy activity in the Labrador Sea so that too little freshwater is transported from the West Greenland Current into the interior of the Labrador Sea (e.g. Eden and Böning, 2002;
Kawasaki and Hasumi, 2014).

      In ER$_{pp}$ (Fig. 10f) the mixed layer depths are to a large extent similar to our reference simulation HR$_{pp}$. However, the convection centre in the Labrador Sea is confined to a more southeastern area with deeper mixed layers in ER$_{pp}$. This is due to resolved eddies, in particular Irminger Rings, that flatten the isopycnals thereby limiting the northward extent of the convection area (Rieck et al., 2019). The resolved eddy activity can be seen in the eddy kinetic energy field (Fig. A9), defined
as $eke = \frac{1}{2}(\overline{u'^2 + v'^2})$ in $\mathrm{m^2\,s^{-2}}$, where $(u', v' = u - \bar{u}, v - \bar{v})$ with $(\bar{u}, \bar{v})$ the monthly mean zonal and meridional velocity. The $eke$ shows the largest values originating in the West Greenland Current and eddy shedding into the inner Labrador Sea. These eddies confine deeper mixed layers to the area with minimum $eke$. However, a resolution of $0.1°$ is still not sufficient to resolve so-called Convective Eddies, which emerge from baroclinic instabilities at the rim of the mixed patch due to strong buoyancy gradients and are thought to be the main process for rapid restratification in spring (Rieck et al., 2019).
ER$_{pp}$ simulates the most realistic mixed layers depth in the Irminger Sea and south of Cape Farewell. The deeper mixed layers might be related to a stronger doming of isopycnals because of an enhanced cyclonic circulation or recirculating Irminger Current (Pickart et al., 2003; Våge et al., 2011). Another reason could be enhanced advection of Labrador Sea water from the Labrador into the Irminger Basin that preconditions the water south of Cape Farewell for convection. However, the processes that lead to deep convection in the Irminger Sea are complex, and it is still not fully understood how eddies affect
the preconditioning/triggering of convection and where their main formation area is (Fan et al., 2013; DuVivier and Cassano, 2016).

### 4.5.2   Southern hemisphere

In the Southern Ocean, we define the mixed layer depth as the depth where the density deviates by $\sigma_t = 0.03\,\mathrm{kg\,m^{-3}}$ from the surface. MPI-ESM1.2 simulates very deep winter mixed layers in the Weddell and Ross Sea (Fig. 11). The main reasons for the
mismatch with mixed layer depths derived from Argo floats is the lack of such floats in ice-covered regions (even though some under-ice float data has recently become available (e.g. Campbell et al., 2019)). In the Weddell Sea, the convection reaches down into the deep ocean, which is a known problem in many state-of-the-art ESMs (Sallée et al., 2013; Kjellsson et al., 2015; Heuzé et al., 2015; Naughten et al., 2018). Spurious open-ocean deep convection leads to semi-permanent Weddell Sea

polynyas, as warm Circumpolar Deep Water is continuously brought to the surface, causing sea ice to melt so that the ocean becomes exposed to the cold atmosphere.

Possible explanations for this widespread bias are: insufficient freshwater input (Kjellsson et al., 2015), in particular glacial melt water (e.g. Stössel et al., 2015), and insufficient wind mixing in summer (Timmermann and Beckmann, 2004). Reduced
wind mixing allows salt from brine rejection to accumulate in the winter water layer and eventually to erode the stratification. In both cases, salinity increases in the winter upper layer until the weakly stratified water column overturns (Naughten et al., 2018).

The diagnosed mixed layer depth, however, is very sensitive to the chosen density threshold because of the very weakly stratified water column. We decided to apply a commonly used threshold for the Southern Ocean of $\sigma_t = 0.03\,\mathrm{kg\,m^{-3}}$, but
note, however, that if a lower threshold of $\sigma_t = 0.01\,\mathrm{kg\,m^{-3}}$ is chosen, the mixed layer depth rarely exceeds 300 m, because of a shallow stratified surface layer.

Based on two simulations with the GFDL-ESM with different resolutions of their ocean component ($0.25°$ and $0.1°$), Dufour et al. (2017) found that deep convection in the Weddell Sea does not necessarily lead to open-ocean polynyas. They argue that excessive vertical mixing in the lower-resolution ocean component hinders the build-up of a heat reservoir at depth
that is necessary for Weddell Sea polynyas to occur intermittently as expected under pre-industrial conditions (e.g. de Lavergne et al., 2014; Gordon, 2014). They further argue that the more realistic representation in the higher-resolution simulation stems from (1) the fact that mesoscale eddies tend to flatten isopycnals thereby increasing the stratification, and (2) the more detailed bathymetry which allows for a better simulation of dense-water overflows.

Based on forced MPIOM and coupled MPI-ESM simulations with varying resolution, Stössel et al. (2015) found that the
Southern Ocean winter sea ice and water properties of a $0.1°$ (TP6M) ocean simulation improved considerably upon switching from a forced to a coupled mode of operation, largely due to an associated increase in surface freshwater flux. These findings are consistent with our $\mathrm{ER_{pp}}$ simulation (Fig. 11f), where the mixed layer depth in the central Weddell Sea is overall reduced in comparison with $\mathrm{HR_{pp}}$ (Fig. 11b). At the same time, the area of deep mixed layers shifts to the eastern part of the Weddell Sea, close to the Maud Rise plateau, where $\mathrm{ER_{pp}}$ still simulates very deep mixed layers in September. This, in turn, could be
a result of the better resolved bathymetry in this region. Kurtakoti et al. (2018) explained how Maud Rise polynyas formed in a high-resolution ($0.1°$ ocean component) ESM simulation, while none formed in a low-resolution simulation with the same model. A decisive reason for this was the steeper and better resolved bathymetry of and around Maud Rise that allowed for sufficiently strong Taylor columns to form.

For the larger Weddell Sea polynyas, de Lavergne et al. (2014) and Gordon (2014) argue that such should only emerge under
pre-industrial conditions. Even though de Lavergne et al. (2014) praise the low-resolution MPI-ESM for belonging to the class of convecting models, Kurtakoti et al. (2018) explain that large-scale Weddell Sea polynyas should only occur intermittently under pre-industrial conditions and only by growing out from Maud Rise polynyas, which themselves should only occur at high model resolution ($0.1°$). Since the greenhouse gas forcing of the experiments presented here are fixed at the 1950 level, one would expect the Southern Ocean of the model to already have adjusted to the present-day situation when no Weddell Sea
polynyas are expected to occur (due to the southward shift of the precipitation rich westerlies). Strong convection and large

Weddell Sea polynyas, as implied by the perpetual large regions of excessively deep mixed layers (Fig. 11), should thus be viewed as an unrealistic behaviour.

As suggested by Timmermann and Beckmann (2004), the vertical mixing scheme affects the sensitivity of spurious deep convection in the Weddell Sea. According to Kjellsson et al. (2015) and Timmermann and Beckmann (2004), sufficient vertical mixing is required in the top 100 m of the mixed layer in the Weddell Sea to prevent polynya formation. In our simulations, the wind induced mixing decreases quadratically with an increase in sea ice cover, which may lead to deficient mixing under sea ice, thus partly explaining the deep convection in the Weddell Sea. Although the KPP scheme reduces the mixed layer depths in the Ross Sea, it enhances deep convection in the central ($HR_{kpp}$) and eastern part of the Weddell Sea ($XR_{kpp}$). This enhanced deep convection contributes to the enhanced ACC strength (Tab. 3), as it causes a steepening of the isopycnals across the ACC and thus an increased geostrophic flow (Jungclaus et al., 2013; Stössel et al., 2015; Naughten et al., 2018). This is another indication that the eddy activity is too low in the KPP simulations, so that isopycnals remain too steep and the water too weakly stratified.

Besides the resolution of the ocean component and the choice of the vertical ocean mixing scheme, a higher resolution of the atmosphere component has also a distinct effect on the simulated winter mixed layer depth (Fig. 11d versus 11b and Fig. 11e versus 11c), which is related to the reduced meridional pressure gradient (not shown) over the Weddell Sea. Stössel et al. (2015) found an improvement of the high-latitude Southern Ocean water-mass properties and winter sea ice cover in a simulation, where the high-resolution (TP6M) MPIOM was coupled to a T255 atmosphere (ECHAM6) compared to a coupled simulation with a TP6M ocean and T63 atmosphere. In terms of the ocean mixed layer depth, our results support these earlier findings, as also indicated by the reduction of the ACC to more realistic values (Tab. 3).

In all our model simulations shown here, sea ice salinity has a constant value of $5\,\mathrm{g\,kg^{-1}}$. As explained in Stössel et al. (2015), Vancoppenolle et al. (2009) and Hunke et al. (2011) argue for a sea ice salinity of about $8\,\mathrm{g\,kg^{-1}}$ for first-year ice, i.e. the kind of sea ice mostly found around Antarctica. Such a higher value would reduce the amount of brine release during ice formation, thus favoring a more stable upper-ocean water column in fall and winter. Another issue is the ice export from the coast: if too weak, it will strengthen open-ocean convection at the expense of near-boundary convection (e.g. Stössel et al., 2015; Haumann et al., 2016).

Another modelling challenge is the mixed layer depth in the Subantarctic Frontal zone equatorwards of the ACC (Rintoul and Trull, 2001). This is an important area for heat and $CO_2$ uptake and for the formation of the Subantarctic Mode Water. State-of-the-art ocean models simulate very shallow mixed layers between 40 to $60\,°\mathrm{S}$ in comparison to Argo float observations (DuVivier et al., 2018). This discrepancy is in particular large in September when the Argo float data consistently show mixed layer depths of about 400 m (see Fig.2 in DuVivier et al. (2018) and Fig. A10), even reaching depths of 700 m (Holte et al., 2017). Low-resolution models (e.g. $1\,°$), however, simulate depths of only 200 to 300 m (DuVivier et al., 2018).

The main reason is that the ocean boundary layer in the models is not penetrating deep enough into the stratified subsurface ocean, where a high salinity maximum layer is observed between 150 to 200 m depth that originates from the Agulhas retroflection. This layer is modified in a complex way by Ekman pumping/suction. This subsurface salinity maximum builds up over spring and early summer and is mixed out in September. It is expected that the mixed layer depths increase by either

increasing the horizontal resolution or by improving the vertical mixing parameterizations (DuVivier et al., 2018) allowing deeper penetrations of the ocean boundary layer into the subsurface salinity core.

The Argo data (Fig. A10a) show mixed layer depths in excess of $400\,\mathrm{m}$ in the deep mixing band. Our reference simulation ($\mathrm{HR_{pp}}$) simulates too shallow mixed layers of only about 200 to 300 m (Fig. A10b), which is in agreement with the results from

(DuVivier et al., 2018). Deeper mixed layers are simulated by either using the KPP scheme ($\mathrm{HR_{kpp}}$; Fig. A10c) or by increasing the ocean resolution ($\mathrm{ER_{pp}}$; Fig. A10f). Deeper mixed layers as in $\mathrm{ER_{pp}}$ were also found in other eddy-resolving ocean models (Lee et al., 2011; Li and Lee, 2017, R. J. Small, pers. Comm. 2019). However, the reason for improved mixed layer depth with high resolution is still unclear, and may be due to changes in circulation, local stratification or indirectly due to mixing (Lee et al., 2011; Li and Lee, 2017; DuVivier et al., 2018). As already suspected by DuVivier et al. (2018), the nonlocal transport

terms of the KPP scheme seem to favour deeper penetrations of the boundary layer into the salinity maximum layer, although this seems to happen in too wide a latitude band.

In $\mathrm{ER_{pp}}$, the deep mixed layers are sharply confined to the observed latitudinal band between 40 to $60\,°\mathrm{S}$. However, they appear to be deeper compared to the Argo float retrievals from Holte et al. (2017) (Fig.A10a), for reasons that need to be further investigated. Nevertheless, the simulation of deeper mixed layers seems to be more realistic, which gives fidelity to our models

with either an eddy-resolving resolution in the Southern Ocean or using KPP.

## 4.6 Atlantic meridional overturning circulation

The large-scale global meridional overturning circulation (MOC) is an important carrier of heat and freshwater in the climate system. The Atlantic MOC (AMOC) is considered to be the strongest part of the MOC (Trenberth and Caron, 2001). The North Atlantic contributes about $25\,\%$ of the total poleward heat flux (ocean plus atmosphere) (Srokosz and Bryden, 2015; Lozier

et al., 2017). The meridional transport of heat and salt follows the zonally integrated volume transport that, when facing west, emerges a clockwise rotating NADW cell and a counterclockwise rotating Antarctic Bottom Water (AABW) cell.

Fig. 12 shows the associated meridional overturning volume transport stream function, or AMOC, of all 5 simulations and Tab. 4 shows the time-mean AMOC strength at $26\,°\mathrm{N}$ at $1000\,\mathrm{m}$ depth, as well as the heat and salt transports across $50\,°\mathrm{N}$. The time-mean of the AMOC is about $14.9\,\mathrm{Sv}$ in $\mathrm{HR_{pp}}$ and comparable to the $16\,\mathrm{Sv}$ of the MPI-ESM1.2-HR described by

Müller et al. (2018). It is slightly lower than the observed mean value ($\pm$ one standard deviation) of $17 \pm 4.4\,\mathrm{Sv}$ (Apr 2004 to Feb 2017) from the RAPID array (McCarthy et al., 2015; Smeed et al., 2017). $\mathrm{HR_{kpp}}$ simulates a stronger AMOC of $18.9\,\mathrm{Sv}$, which is the largest value of all our simulations. A possible explanation for this is that the volume transport of the overflow waters across the Greenland-Scotland ridge are also slightly higher with the KPP scheme (Tab. 3). After the overflow waters descend along the continental slopes and mix with ambient water masses, they contribute to a stronger NADW cell (Dickson

and Brown, 1994) in the KPP simulations.

Figure 12f shows vertical profiles of the AMOC at $26.5\,°\mathrm{N}$ in comparison to the RAPID data. All simulations (except $\mathrm{XR_{pp}}$) produce transports close to the observations. The volume transport of $\mathrm{HR_{kpp}}$, however, is on the stronger side of the observations, whereas the transport of the other simulations are on the lower side of the observations. All models show a too strong southward transport of NADW below 2000 m, which suggests a too strong Deep Western Boundary Current.

The reduced wind stress from ECHAM6.3 at T255 results in the above mentioned slowdown of the AMOC in $XR_{pp}$. In this simulation, the NADW cell reaches a maximum volume transport of only about $11.0\,Sv$, which is slightly higher than the $9.0\,Sv$ reported by Putrasahan et al. (2019). This discrepancy is because we analyze an earlier period of the same $XR_{pp}$ simulation when the AMOC is still drifting to lower values. An important finding is that $XR_{kpp}$ simulates a stable AMOC ($14.6\,Sv$), despite the weak wind stress with the T255 atmosphere. In terms of volume transport, going to an eddy resolving ocean resolution ($ER_{pp}$) does not increase the strength of the NADW cell. This finding is opposite to what Hewitt et al. (2016) and Storkey et al. (2018) found.

However, the bottom (AABW) cell becomes stronger (Fig. 12e), which may be due to similar effects as described by Sein et al. (2018), who hypothesize that eddy-induced transport acts to flatten the outcropping isopycnals in the Southern Ocean. So eddies counteract a wind-induced steepening of isopycnals, while at the same time, a stronger vertical gradient between the AABW and the warmer ambient ocean is maintained. The flatter isopycnals reduce the vertical mixing because of a more stratified water column, as indicated by the reduced mixed layer depths in the Weddell Sea in ER (Fig. 11e). Reduced convection maintains denser AABW, seen by sharper gradients of temperature and salinity in ER (Fig. A4e) and it theoretically helps to build up a deep heat reservoir (Dufour et al., 2017) that is required for intermittent Weddell Sea polynyas. However, in our ER simulation, Weddell Sea polynyas still form too frequently. On the other hand, better resolved bathymetry is important for the formation of AABW over the continental shelves, which is partly resolved in ER.

We define the depth of the NADW cell as the depth where the volume transport crosses the zero line in Fig. 12f. The observed annual mean depth ($\pm$ one standard deviation) of the NADW cell (Tab. 4) from the RAPID data is about $4379 \pm 279\,m$ at $26.5\,°N$. All our simulations reveal shallower NADW cells of around $3000\,m$, but with a noticeable tendency to become deeper with the KPP scheme. A stronger AMOC deepens the NADW cell (Marshall et al., 2017), because more NADW is formed by overturning. This is consistent with the mixed layers being deeper in the KPP simulations, and with the increased overflow water from the GIN seas (Tab. 3).

In $XR_{pp}$ the NADW cell is shallower ($2665\,m$), consistent with a much weaker NADW cell. $ER_{pp}$ simulates a slightly deeper ($2941\,m$) NADW cell than $HR_{pp}$, probably because of increased overflow water from the GIN seas (Tab. 3), but still not as deep as in the simulations with the KPP scheme. The higher volume transport by the AMOC in the simulations with KPP yields a slightly enhanced heat transport and a considerably higher salt transport across $50\,°N$ (Tab. 4, Fig. A1). This larger salt input into the subpolar North Atlantic with KPP is a main reason why the overturning becomes stronger, and in particular why $XR_{kpp}$ maintains a stable AMOC, even with reduced wind stress.

The stronger deep convection in the northern North Atlantic (Labrador and Irminger Sea) and in the Nordic Seas enhance the local NADW formation that deepens the NADW cell. Note, however, that open-ocean deep convection is not directly associated with a net vertical mass transport (Marotzke and Scott, 1999; Katsman et al., 2018) and thus the location of convective mixing and of strongest downward mass transfer need not coincide.

The surplus of NADW water has to be replaced by water masses from the NAC, leading to larger volume and salt transports of this current. Once the upper cell in the Atlantic becomes stronger, a positive feedback sets in. A stronger NAC strengthens the cyclonic circulation of the subpolar gyre (Tab. 2) and the separation of water masses in the gyre centres (Labrador/Irminger

Sea) from the ambient water masses. This separation of water masses in the gyre centres enhances deep convection because of (1) increased isopycnal doming that leads to a weaker stratification of the water column and to a shallower thermocline, and (2) because of reduced mixing with ambient water, so that the water masses in the gyre centre are exposed longer to the overlaying cold atmosphere, leading to increased heat loss. Both effects reduce the surface stratification and its resistance to

erosion, favouring deep convection that again strengthens the overturning cell. In addition, increased salt input densifies the upper water masses of the northern North Atlantic and the Nordic Seas, so that convection is enhanced.

As a result of the enhanced AMOC, the adiabatic upwelling branch of the MOC south of the ACC has to become stronger too (Fig. A4). Since we use the same background diffusivities below the mixed layer in KPP as with PP, no significant differences in diapycnal diffusion occur in the Pacific (not shown). That is, the only return pathway that might be modified by KPP is via

wind-driven adiabatic upwelling in the Southern Ocean (Marshall and Speer, 2012). Indeed, the upwelling in the Pacific sector of the Southern Ocean increases with KPP (Fig. A4). An increase in upwelling in the Southern Ocean further strengthens the northern cell (Marshall et al., 2017). This feedback is however acting on longer time scales than the slowdown of the AMOC in our model. Therefore, the Southern Ocean is not the main factor in sustaining a stable AMOC in XR$_{\text{kpp}}$.

## 5   Summary

We compared control simulations of various MPI-ESM1.2 configurations following the HighResMIP protocol and investigated separately the resolution effects of the atmosphere and ocean model configurations and the effects of an alternative diapycnal ocean mixing scheme on the mean state of the atmosphere and ocean.

### 5.1   Effects of high-resolution ocean

A high-resolution ocean ($0.1°$) reduces biases in the ocean mean-state and it has a major impact on the large-scale temperature

distribution in the atmosphere. Cold temperature biases in the Southern Hemisphere, and to a lesser extent in the Northern Hemisphere, are reduced. The latter bias could not be removed by just increasing the atmospheric resolution. In the ocean, warm and saline biases in the Southern Atlantic were removed, because of the better representation of the Agulhas Current system (Putrasahan et al., 2015; Cheng et al., 2016) and because of eddy-induced upward transport of fresh and cold water masses, as described in von Storch et al. (2016). In general, swifter and narrower boundary currents are simulated in all basins

with a high resolution. In the North Atlantic, the warm and saline bias was removed because of a better simulation of the water properties of the outflowing Mediterranean Water. High resolution improves the separation of the Gulf Stream, although the NAC remained still too zonal in our simulation. Furthermore, the warm bias of the Atlantic Layer in the Arctic Ocean was removed, probably because of reduced numerical mixing due to the higher resolution, which confirms the results of Wang et al. (2018). In addition, the deep-convection centre shifted to the southeast in the Labrador Sea, and to the east in the Weddell Sea.

With the high-resolution ocean (ER$_{\text{pp}}$), the centre of deep convection in the Weddell Sea shifts to the east, to the vicinity of the Maud Rise Plateau. A high resolution was also found to improve the mixed layer depths in the Subantarctic Frontal zone in the Indian, Australian and Pacific sectors of the Southern Ocean.

## 5.2 Effects of higher resolution atmosphere

The T255 atmosphere reduced mainly the wind stress over the ocean in both hemispheres, in particular in the Labrador Sea and in the Weddell Sea. In the latter, a reduced meridional pressure gradient in the atmosphere reduces the ACC transport to realistic values, as also reported by Stössel et al. (2015). In the northern hemisphere, however, the T255 atmosphere reduces the near-surface wind speeds over the subpolar gyre, so that the subpolar gyre slows down and because of less cyclonic movement and less salt advection into the gyre centres, the deep convection diminishes (KPP scheme) or vanishes (PP scheme), as described by Putrasahan et al. (2019). In contrast to the near-surface, the jet streams, however, are stronger in the T255 atmosphere.

## 5.3 Effects of the KPP scheme

The main effects of the KPP scheme are stronger deep convection in both hemispheres, reflected by deeper mixed layers. Under convective forcing the non-local fluxes of the KPP scheme produce much higher diffusivities compared to the enhanced diffusivity parameterization that we use for the PP scheme. This stronger deep convection with the KPP scheme produces more NADW locally in the convection centres (Labrador, Irminger, and GIN Seas) which in turn strengthens the AMOC. When coupled with the T255 atmosphere, the AMOC remains stable with the KPP scheme because of this enhanced overturning, which produces sufficient NADW to maintain a strong enough upper cell. Another effect that produces deeper mixed layers is a stronger subpolar gyre that domes the isopycnals and helps to precondition the water column for convection. This is also true for the Weddell Gyre with the same effect. We further found deeper mixed layers in the Subantarctic Frontal zones, which are important for the uptake of heat and $CO_2$. The stronger AMOC transports more salt and heat into the North Atlantic, so that the cold bias in the northern hemisphere is removed.

*Code and data availability.* The MPI-ESM1.2 model code is made available under a version of the MPI-M Software License Agreement (http://www.mpimet.mpg.de/en/science/models/license; branch *mpiesm-1.2.01-cvmix* for the KPP simulations and *mpiesm-1.2.01-primavera_PP* for the PP simulations). Primary data and scripts used in the analysis, and other supplementary information that may be useful in reproducing the author's work, are archived by the Max Planck Institute for Meteorology and can be obtained by contacting publications@mpimet.mpg.de.

*Author contributions.* JJ and JS designed the experiments and DP and KL set up the model configurations and performed the simulations. OG, NB and HH have implemented the new mixing parameterizations in MPIOM. OG prepared the manuscript with contributions from all co-authors.

*Competing interests.* The authors declare that they have no conflict of interest.

*Acknowledgements.* We thank the German Computing Centre (DKRZ) for providing the computing resources. This research was funded by the EU Horizon 2020 project PRIMAVERA (grant number 641727). This paper is a contribution to the project S2 (Improved parameterisations and numerics in climate models) of the Collaborative Research Centre TRR 181 "Energy Transfer in Atmosphere and Ocean" funded by the Deutsche Forschungsgemeinschaft (DFG, German Research Foundation) - project number 274762653. JJ acknowledges support by the German BMBF RACE-II project (FKZ 03F0729D). AS acknowledges support through a Max-Planck-Society research stipend. We thank Jürgen Kröger for proofreading the manuscript. The OSI SAF data of EUMETSAT were made available from http://osisaf.met.no. The Argo data were made freely available by the International Argo Program and the national programs that contribute to it (http://www.argo.ucsd.edu, http://argo.jcommops.org). Finally, we thank Justin Small (NCAR) and one anonymous reviewer for their constructive and helpful comments.

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

**Table 1.** Overview of MPI-ESM1.2 control simulations used within this study and their horizontal resolutions. The number of vertical levels are 95 in the atmosphere and 40 in the ocean, respectively. In brackets, the nominal horizontal resolution in a Gaussian grid (approximated at the equator) is given. All models use 30 years of spin-up and are analysed for the subsequent 50 years.

| Name | Atmosphere resolution | Ocean resolution | Ocean mixing scheme | Description |
|------|----------------------|-----------------|--------------------|-------------|
| HR | T127 (0.93° or ~103 km) | TP04 (0.4° or ~44 km) | PP, KPP | reference, ocean mixing sensitivity |
| XR | T255 (0.46° or ~51 km) | TP04 (0.4° or ~44 km) | PP, KPP | increased atmospheric resolution, ocean mixing sensitivity |
| ER | T127 (0.93° or ~103 km) | TP6M (0.1° or ~11 km) | PP | increased ocean resolution |

**Table 2.** Maximum values of barotropic stream function (gyre strengths) in Sverdrup ($Sv := 10^6\,m^3\,s^{-1}$) simulated by MPI-ESM1.2 and from observations.

| Region | $HR_{pp}$ | $HR_{kpp}$ | $XR_{pp}$ | $XR_{kpp}$ | $ER_{pp}$ | Obs. | Reference |
|---|---|---|---|---|---|---|---|
| Subpolar gyre (North Atlantic) | 34.6 | 40.6 | 31.0 | 32.1 | 36.6 | 26.0 to 40.0 | Clark (1984); Bersch (1995); Bacon (1997); Lherminier et al. (2007); Holliday et al. (2009) |
| Subtropical gyre (North Atlantic) | 48.2 | 64.9 | 44.0 | 63.9 | 62.8 | 46.0 to 61.0 | Johns et al. (1995) |
| Subtropical gyre (North Pacific) | 84.1 | 116.3 | 73.6 | 95.5 | 80.7 | 42.0$\pm$2.5 | Imawaki et al. (2001) |

**Table 3.** Simulated (mean $\pm$ one standard deviation) and observed net volume transports (Sv $:= 10^6\,\mathrm{m^3\,s^{-1}}$) across sections (postive means northward).

| Section | $HR_{pp}$ | $HR_{kpp}$ | $XR_{pp}$ | $XR_{kpp}$ | $ER_{pp}$ | Obs. | Reference |
|---|---|---|---|---|---|---|---|
| Bering Strait | $0.7\pm0.1$ | $0.7\pm0.1$ | $0.5\pm0.1$ | $0.6\pm0.1$ | $0.9\pm0.1$ | 0.8 [0.7 to 1.1] | Woodgate et al. (2006, 2012) |
| Fram Strait | $-2.5\pm0.6$ | $-1.9\pm0.4$ | $-2.5\pm0.6$ | $-1.9\pm0.5$ | $-1.0\pm0.4$ | $-1.75\pm5.01$ | Fieg et al. (2010) |
| Denmark Strait | $-3.9\pm0.6$ | $-4.2\pm0.7$ | $-4.1\pm0.6$ | $-3.9\pm0.7$ | $-4.6\pm0.4$ | $-4.6$ | Hansen et al. (2008) |
| | | | | | | $-3.4\pm1.4$ | Jochumsen et al. (2012) |
| | | | | | | $-3.2\pm0.5$ | Jochumsen et al. (2017) |
| Iceland – Scotland | $4.0\pm0.8$ | $5.0\pm1.0$ | $4.2\pm0.8$ | $4.4\pm1.0$ | $5.5\pm0.6$ | 4.8 | Hansen et al. (2008) |
| | | | | | | $4.6\pm0.25$ | Rossby and Flagg (2012) |
| | | | | | | $3.8\pm0.6$ | Kanzow and Zenk (2014) |
| Florida Current | $14.6\pm0.7$ | $15.5\pm0.7$ | $12.4\pm0.6$ | $14.1\pm0.6$ | $24.7\pm0.8$ | 31.7 | Kanzow et al. (2010) |
| | | | | | | $31.6\pm2.7$ | McDonagh et al. (2015) |
| Strait of Gibraltar | $0.04\pm0.01$ | $0.04\pm0.01$ | $0.04\pm0.01$ | $0.04\pm0.01$ | $0.05\pm0.01$ | $0.038\pm0.007$ | Soto-Navarro et al. (2010) |
| | | | | | | 0.041 | Bryden et al. (1994) |
| Indonesian Throughflow | $8.5\pm0.8$ | $9.5\pm0.9$ | $8.0\pm0.5$ | $8.5\pm0.8$ | $13.0\pm0.8$ | 11.6 to 15.7 | Gordon et al. (2010) |
| Mozambique Channel | $8.8\pm1.7$ | $6.5\pm2.0$ | $8.0\pm1.3$ | $5.3\pm1.9$ | $13.6\pm1.2$ | 5.0 to 26.0 | DiMarco et al. (2002) |
| | | | | | | $16.7\pm8.9$ | Ridderinkhof et al. (2010) |
| Drake Passage | $161.7\pm3.0$ | $191.9\pm2.6$ | $150.1\pm4.1$ | $170.2\pm3.0$ | $140.9\pm3.0$ | $134.0\pm14.0$ | Nowlin Jr. and Klinck (1986) |
| | | | | | | $137.0\pm8.0$ | Cunningham et al. (2003) |
| | | | | | | $136.7\pm6.9$ | Meredith et al. (2011) |
| | | | | | | $173.3\pm10.7$ | Donohue et al. (2016) |

**Table 4.** Time-mean AMOC volume transports ($\pm$ one standard deviation of annual means) at $26\,°$N in $1000\,$m depth simulated by MPI-ESM1.2 and the depth of the North Atlantic Deep Water (NADW) cell at $26.5\,°$N (defined where the stream function crosses zero). The observed annual mean ($\pm$ on standard deviation) NADW cell depth from the RAPID-MOCHA-WBTS array (Smeed et al., 2017) is $4379 \pm 279\,$m. Further, the time-mean ($\pm$ one standard deviation of annual means) heat and salt transports across $50\,°$N are shown (positive means northward transport).

| Property | $HR_{pp}$ | $HR_{kpp}$ | $XR_{pp}$ | $XR_{kpp}$ | $ER_{pp}$ |
|---|---|---|---|---|---|
| AMOC volume (Sv) | $14.9 \pm 3.5$ | $18.9 \pm 4.0$ | $11.0 \pm 3.8$ | $14.6 \pm 3.9$ | $14.9 \pm 3.6$ |
| NADW cell depth (m) | $2865 \pm 270$ | $3176 \pm 334$ | $2665 \pm 287$ | $2979 \pm 489$ | $2941 \pm 265$ |
| Atl. heat transport across $50\,°$N (PW) | $0.60 \pm 0.04$ | $0.63 \pm 0.06$ | $0.42 \pm 0.06$ | $0.52 \pm 0.05$ | $0.57 \pm 0.03$ |
| Atl. salt transport across $50\,°$N ($10^6\,\mathrm{kg\,s^{-1}}$) | $0.28 \pm 1.89$ | $0.64 \pm 2.18$ | $-1.04 \pm 2.54$ | $0.4 \pm 2.11$ | $-0.22 \pm 1.27$ |

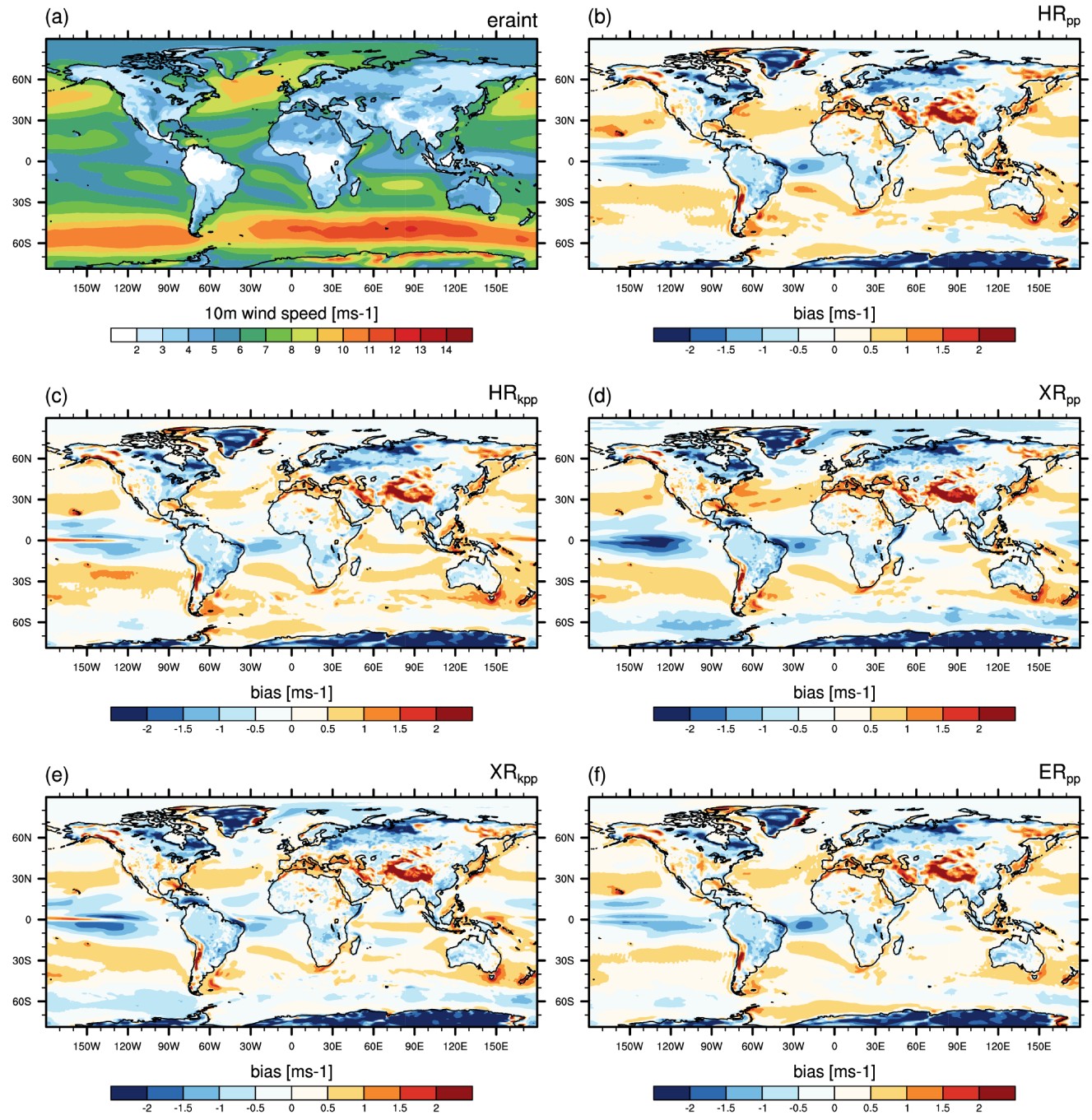

**Figure 1.** Annual mean 10 m wind speed from (a) ERA-Interim (1979–2005) and the bias of: (b) $HR_{pp}$, (c) $HR_{kpp}$, (d) $XR_{pp}$, (e) $XR_{kpp}$, and (f) $ER_{pp}$.

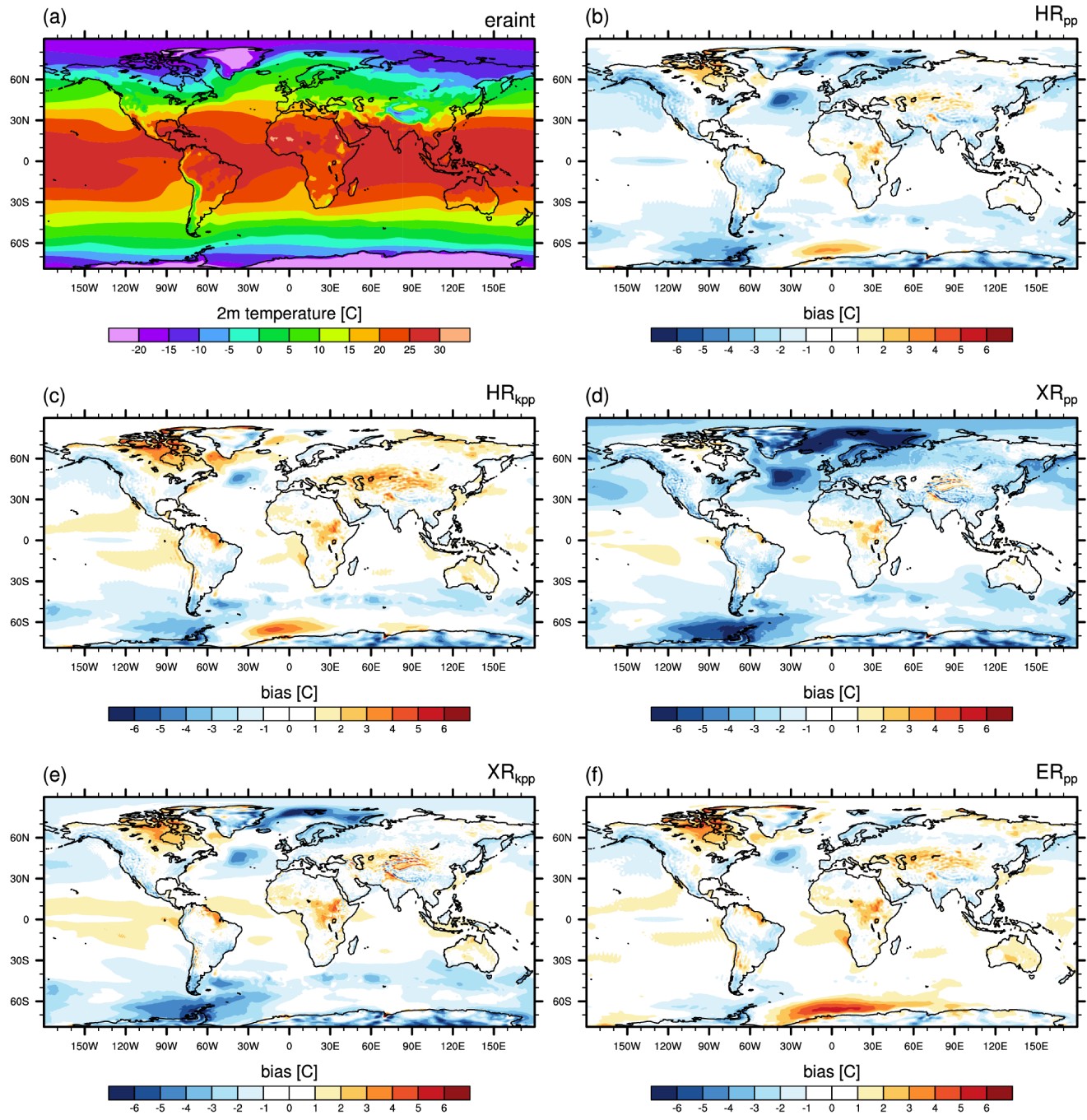

**Figure 2.** Annual mean 2 m temperature from (a) ERA-Interim (1979–2005) and the bias of: (b) $HR_{pp}$, (c) $HR_{kpp}$, (d) $XR_{pp}$, (e) $XR_{kpp}$, and (f) $ER_{pp}$.

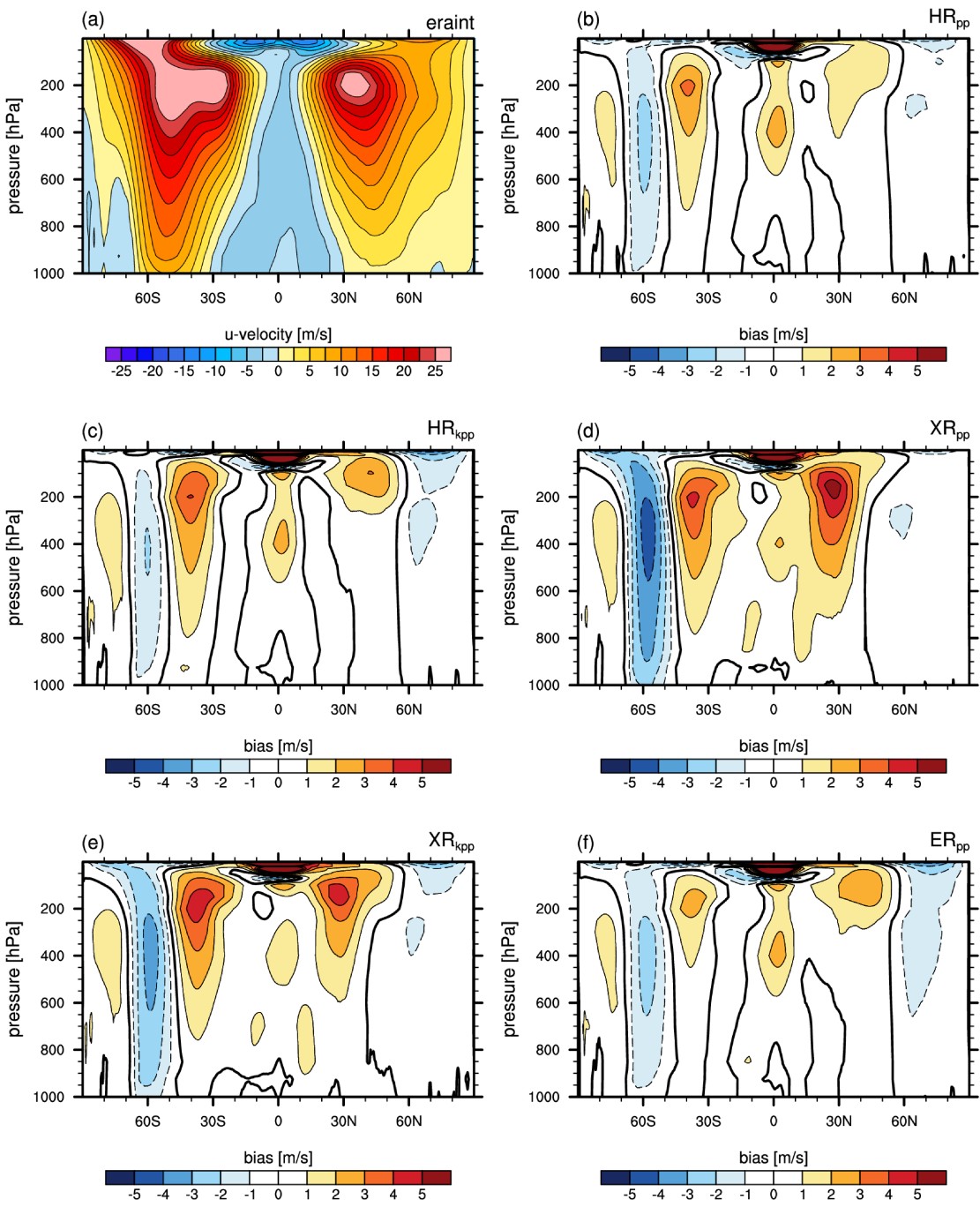

**Figure 3.** Global zonally-averaged u-velocity from (a) ERA-Interim (1979–2005) and the bias (MPI-ESM1.2 minus ERA-Interim) of: (b) $HR_{pp}$, (c) $HR_{kpp}$, (d) $XR_{pp}$, (e) $XR_{kpp}$, and (f) $ER_{pp}$. The zero contour line is shown as a thick solid line; negative (positive) contours are dashed (solid).

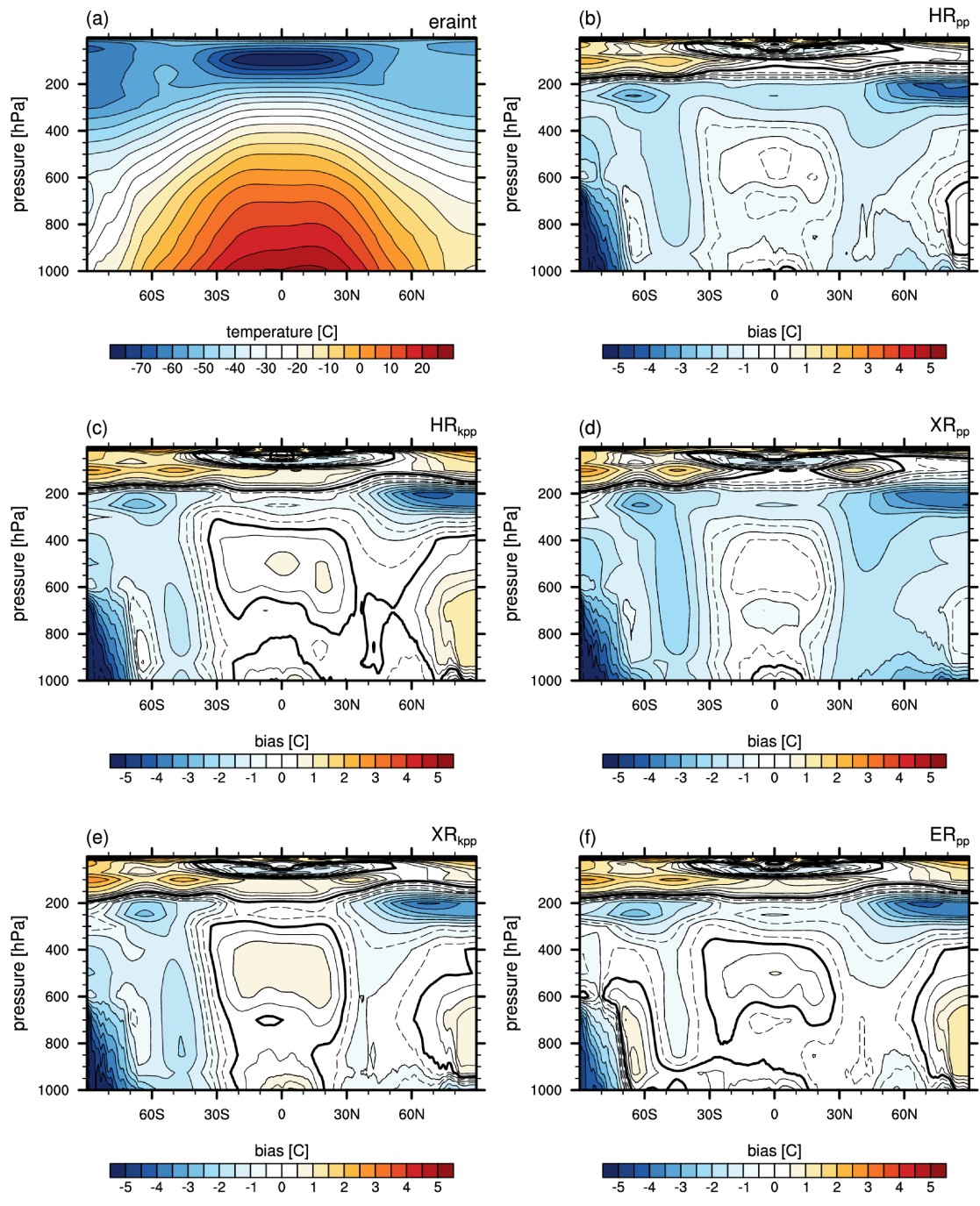

**Figure 4.** Global zonally-averaged temperature from (a) ERA-Interim (1979–2005) and the bias (MPI-ESM1.2 minus ERA-Interim) of (b) HR$_{pp}$, (c) HR$_{kpp}$, (d) XR$_{pp}$, (e) XR$_{kpp}$, and (f) ER$_{pp}$. The contour lines in b-f span $\pm0.75$ with an interval of 0.5K, and of 1.0K outside that range. The zero contour line is shown as a thick solid line; negative (positive) contours are dashed (solid).

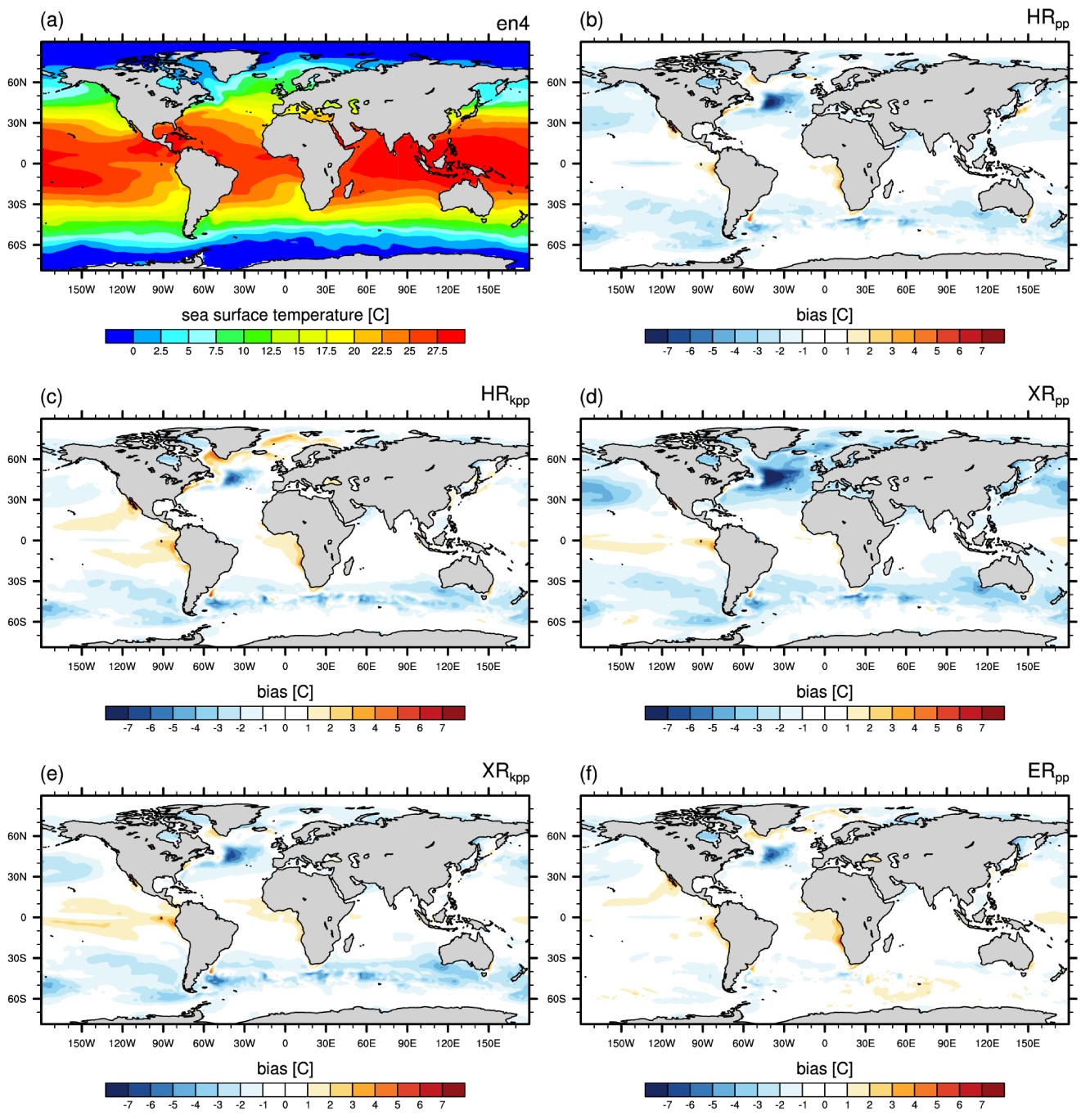

**Figure 5.** Sea surface temperature (°C) from (a) EN4 (averaged over 1945–1955) and differences: MPI-ESM1.2 minus EN4 for (b) HR$_{pp}$, (c) HR$_{kpp}$, (d) XR$_{pp}$, (e) XR$_{kpp}$, and (f) ER$_{pp}$.

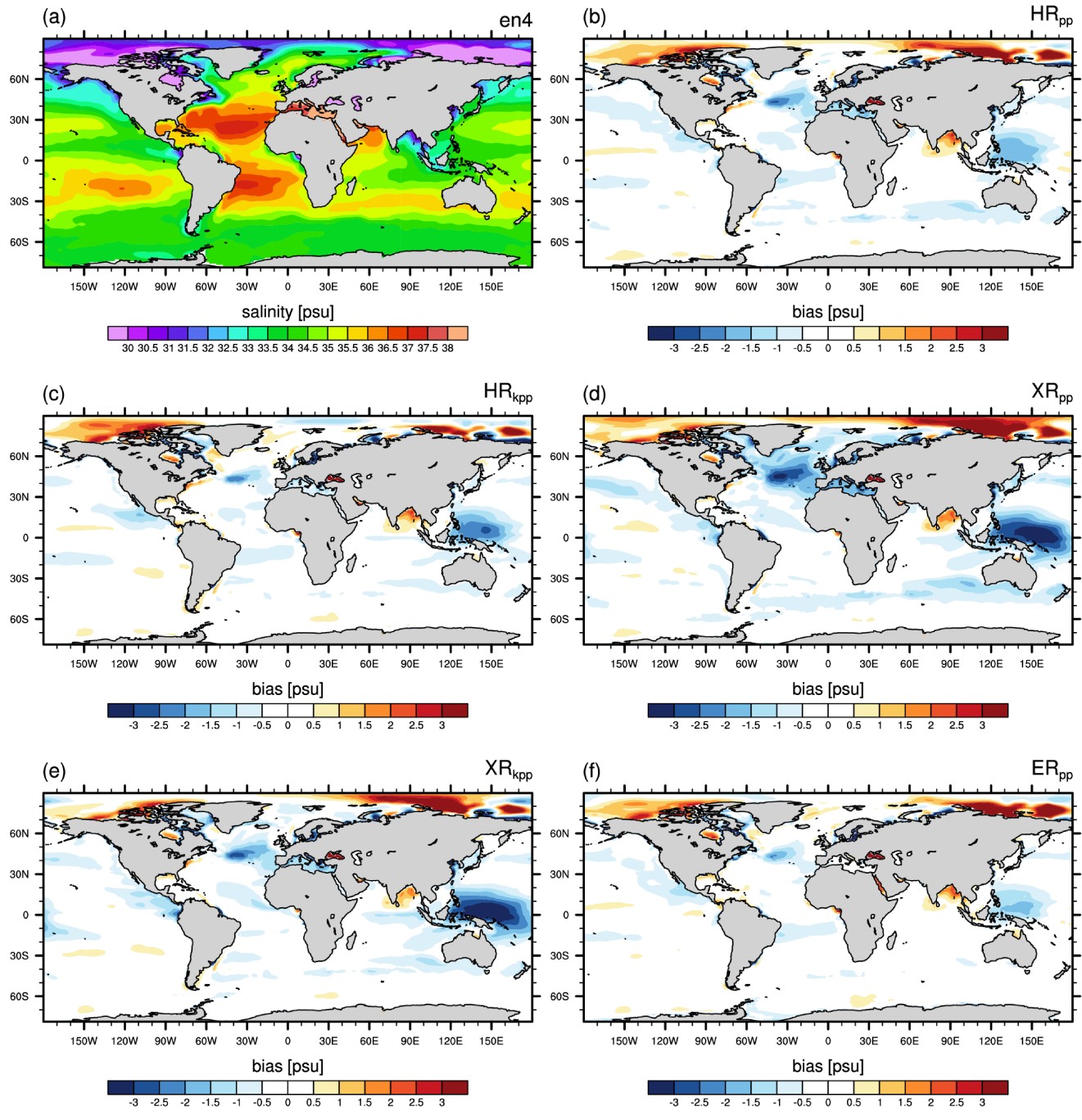

**Figure 6.** Sea surface salinity (psu) from (a) EN4 (averaged over 1945–1955) and for the differences: MPI-ESM1.2 minus EN4 for (b) HR$_{pp}$,(c) HR$_{kpp}$, (d) XR$_{pp}$, (e) XR$_{kpp}$, and (f) ER$_{pp}$.

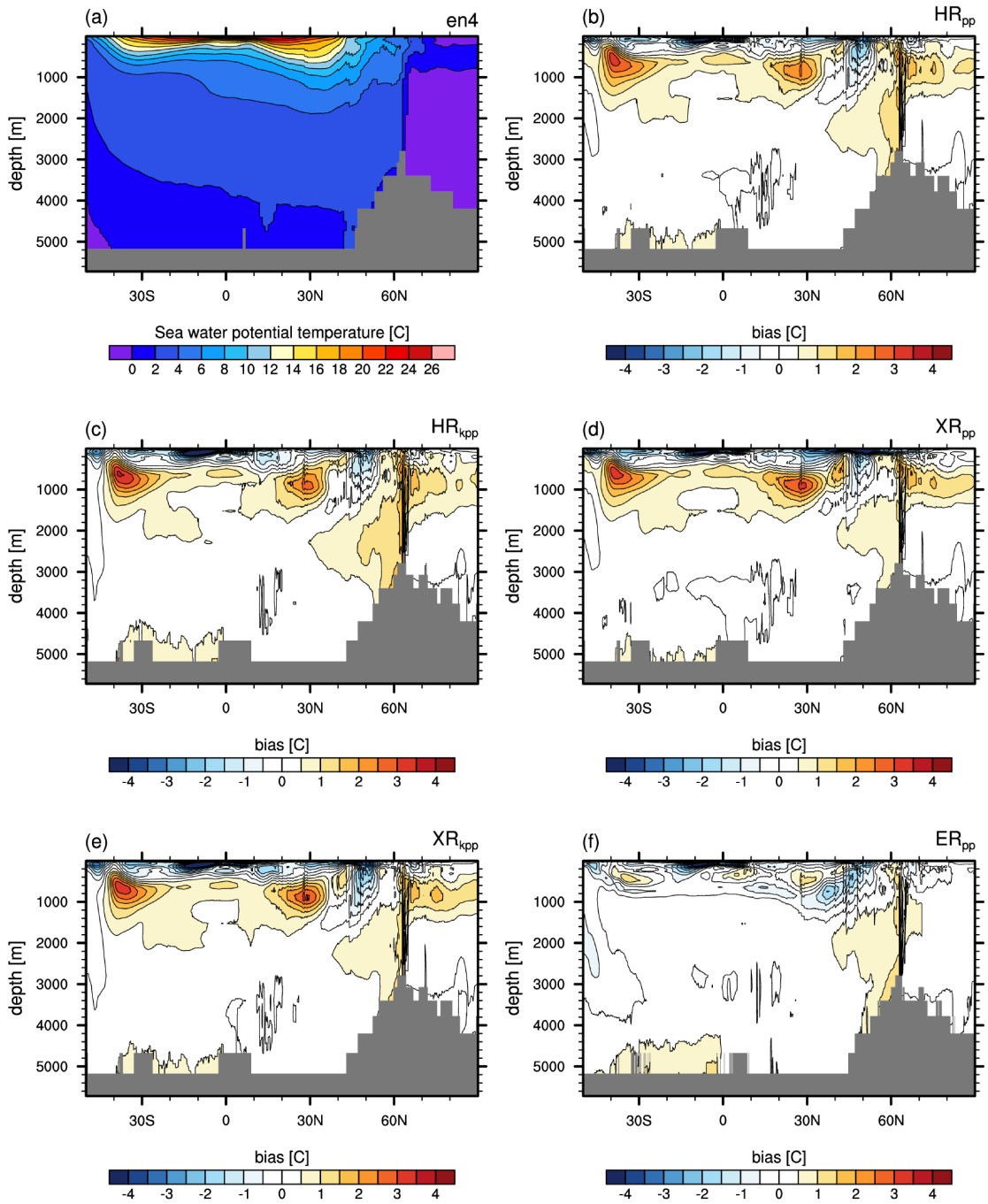

**Figure 7.** Zonal mean temperature transect through the Atlantic basin and the Arctic Ocean of (a) EN4 (averaged over 1945–1955) and the bias (MPI-ESM1.2 minus EN4) of (b) HR$_{pp}$, (c) HR$_{kpp}$, (d) XR$_{pp}$, (e) XR$_{kpp}$, and (f) ER$_{pp}$. Contour levels (b-f) begin with $\pm0.5\,^{\circ}$C.

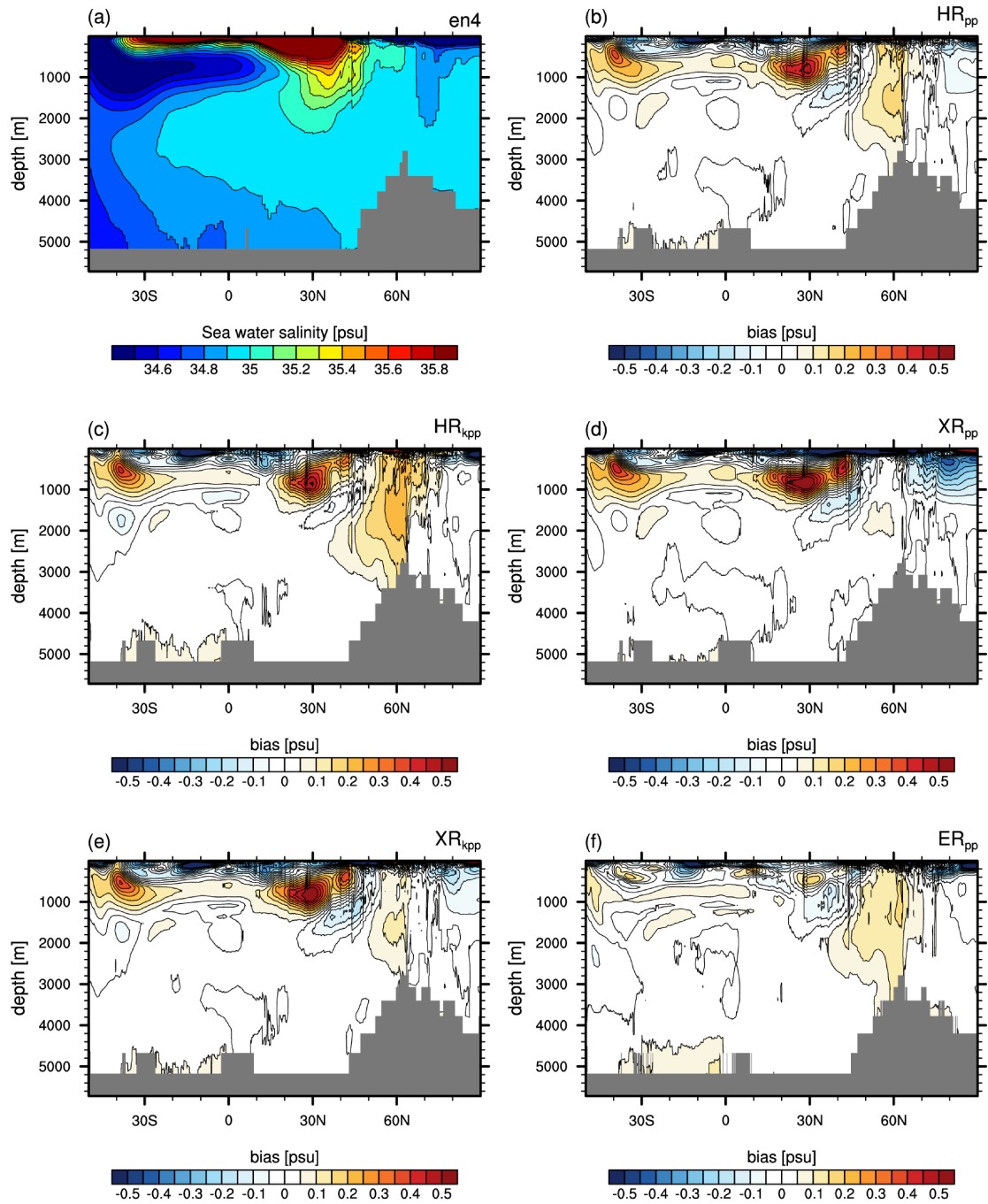

**Figure 8.** Zonal mean salinity transect through the Atlantic basin and the Arctic Ocean of (a) EN4 (averaged over 1945–1955) and the bias (MPI-ESM1.2 minus EN4) (b) HR$_{kpp}$, (c) XR$_{pp}$, and (d) XR$_{kpp}$, (e) XR$_{kpp}$, and (f) ER$_{pp}$. Contour levels (b-f) begin with $\pm 0.05$ psu.

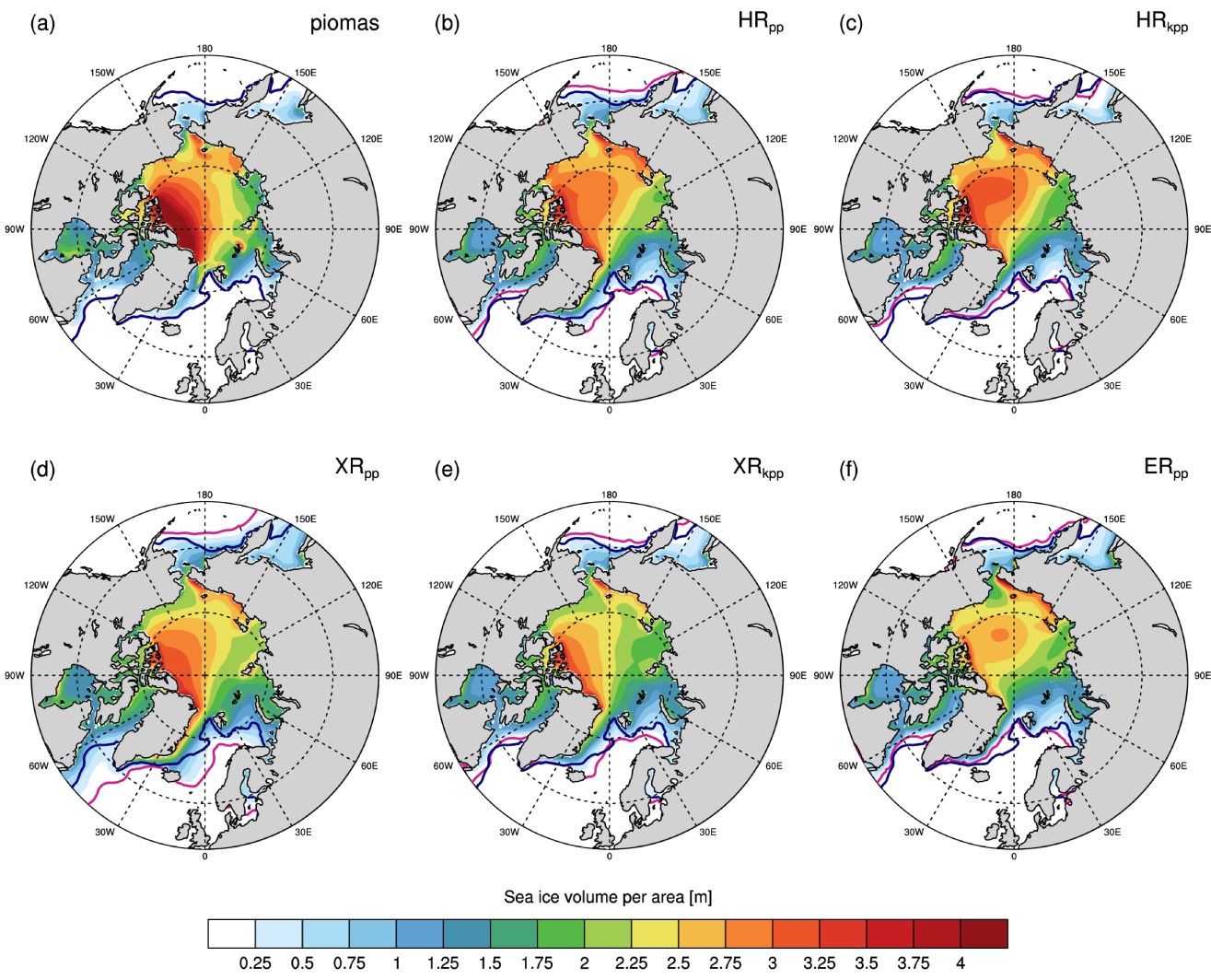

**Figure 9.** Time-averaged Arctic sea ice thickness in March for (a) PIOMAS reanalysis (Zhang and Rothrock, 2003), (b) HR$_{pp}$, (c) HR$_{kpp}$, (d) XR$_{pp}$, (e) XR$_{kpp}$, and (f) ER$_{pp}$. Simulations include their 15 % sea ice concentration contour in magenta and all figures the EUMETSAT OSI SAF observed ice edge (15% contour) in dark blue (averaged March 1979–2005).

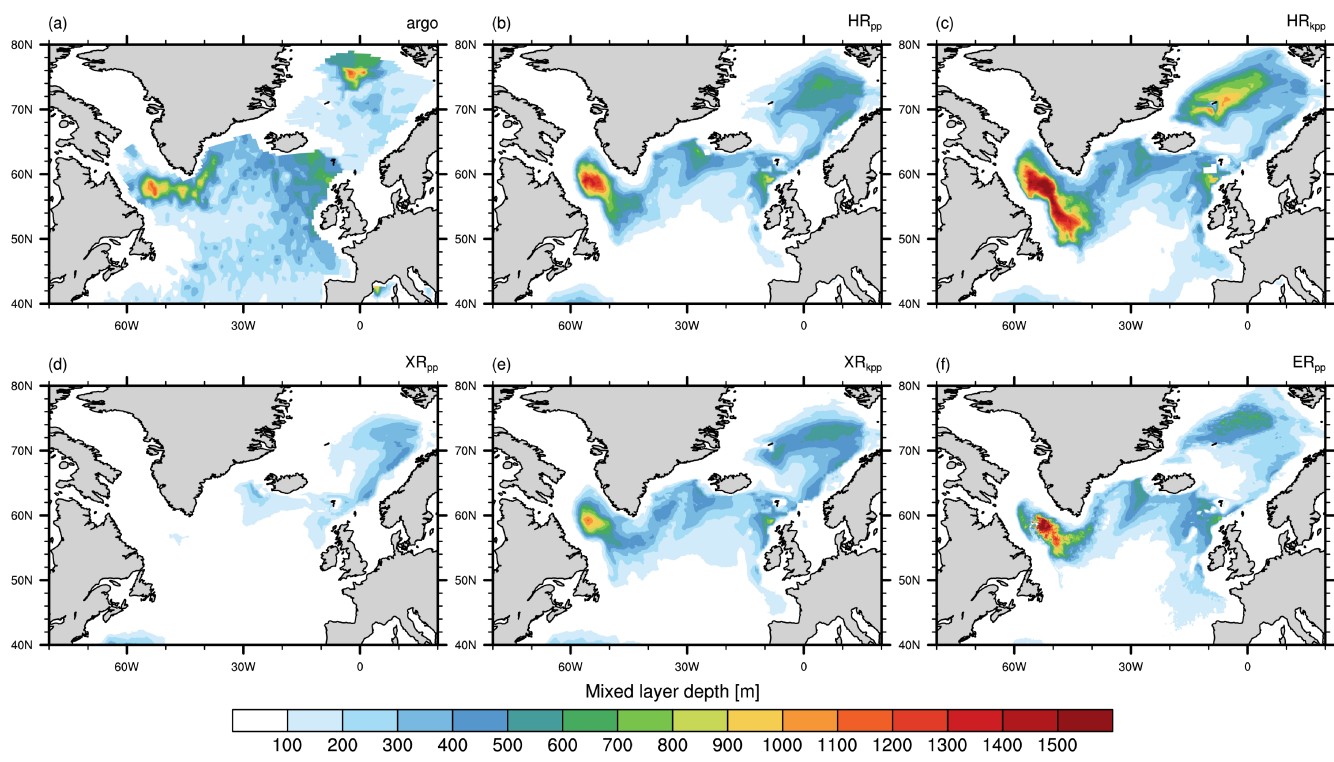

**Figure 10.** Time-averaged mixed layer depth ($\sigma_t = 0.01\,\mathrm{kg\,m^{-3}}$) in March in the North Atlantic and the Nordic Seas from (a) $1° \times 1°$ gridded Argo float data (Holte et al., 2017) and from (b) $\mathrm{HR_{pp}}$, (c) $\mathrm{HR_{kpp}}$, (d) $\mathrm{XR_{pp}}$, (e) $\mathrm{XR_{kpp}}$, and (f) $\mathrm{ER_{pp}}$.

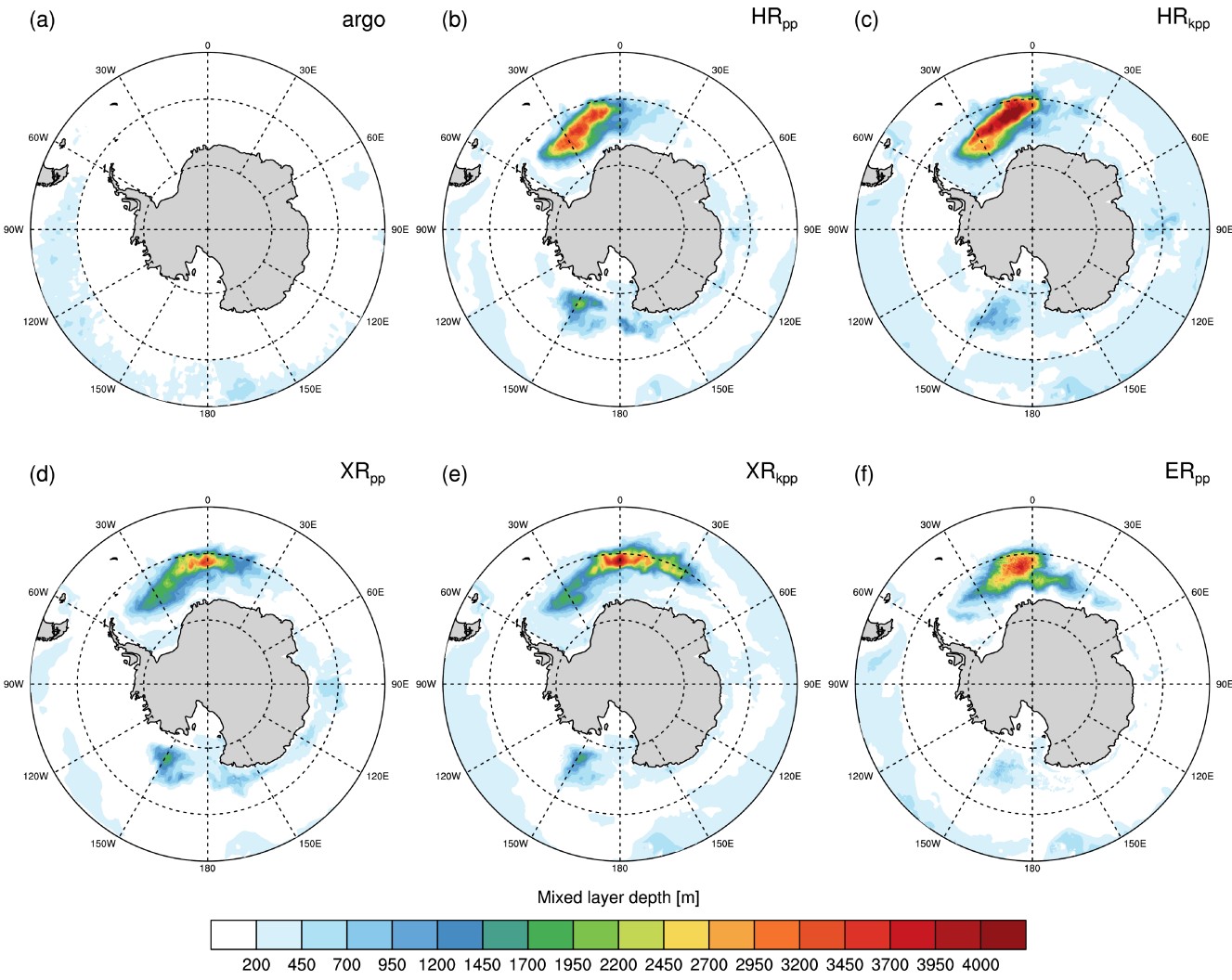

**Figure 11.** Time-averaged mixed layer depth ($\sigma_t = 0.03\,\mathrm{kg\,m^{-3}}$) in September in the Southern Ocean from (a) $1° \times 1°$ gridded Argo float data (Holte et al., 2017) and from (b) HR$_{pp}$, (c) HR$_{kpp}$, (d) XR$_{pp}$, (e) XR$_{kpp}$, and (f) ER$_{pp}$.

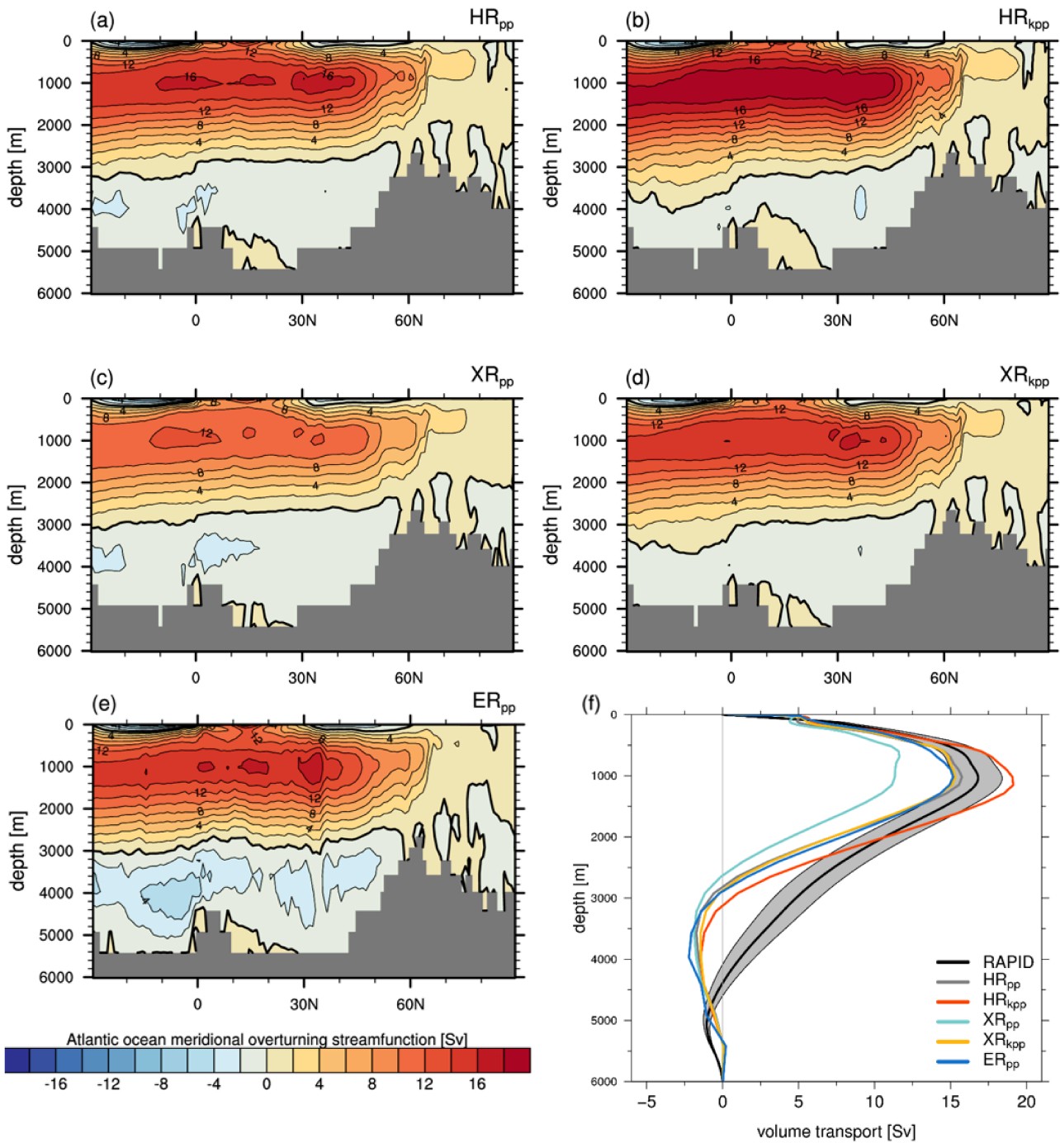

**Figure 12.** Eulerian stream function (Sv := $10^6 \, \mathrm{m}^3 \, \mathrm{s}^{-1}$) of the Atlantic Meridional Overturning Circulation (AMOC) for (a) $HR_{pp}$, (b) $HR_{kpp}$, (c) $XR_{pp}$, (d) $XR_{kpp}$, and (e) $ER_{pp}$. The zero contour is drawn as a thicker line. In (f) annual mean profiles of the AMOC at 26.5 °N are shown as observed from Apr 2004 to Feb 2017 by the RAPID-MOCHA-WBTS array (± one standard deviation marked by grey shading) (Smeed et al., 2017) and simulated by MPI-ESM1.2.

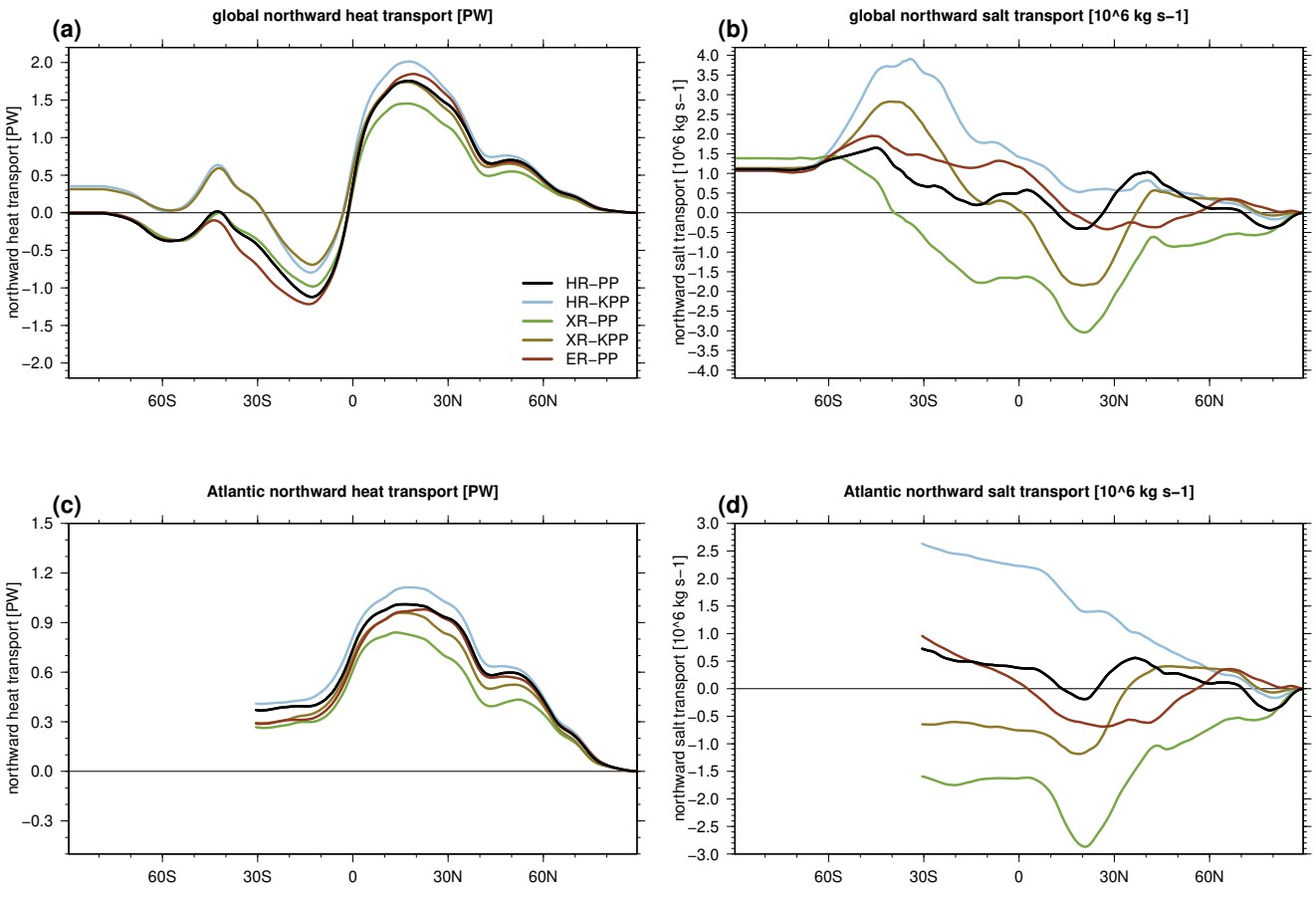

**Figure A1.** Time-averaged northward heat (PW) and salt transport ($10^6 \, \mathrm{kg \, s^{-1}}$) in the global ocean (a,b) and in the Atlantic basin (c,d). Note the different scaling in (c) and (d).

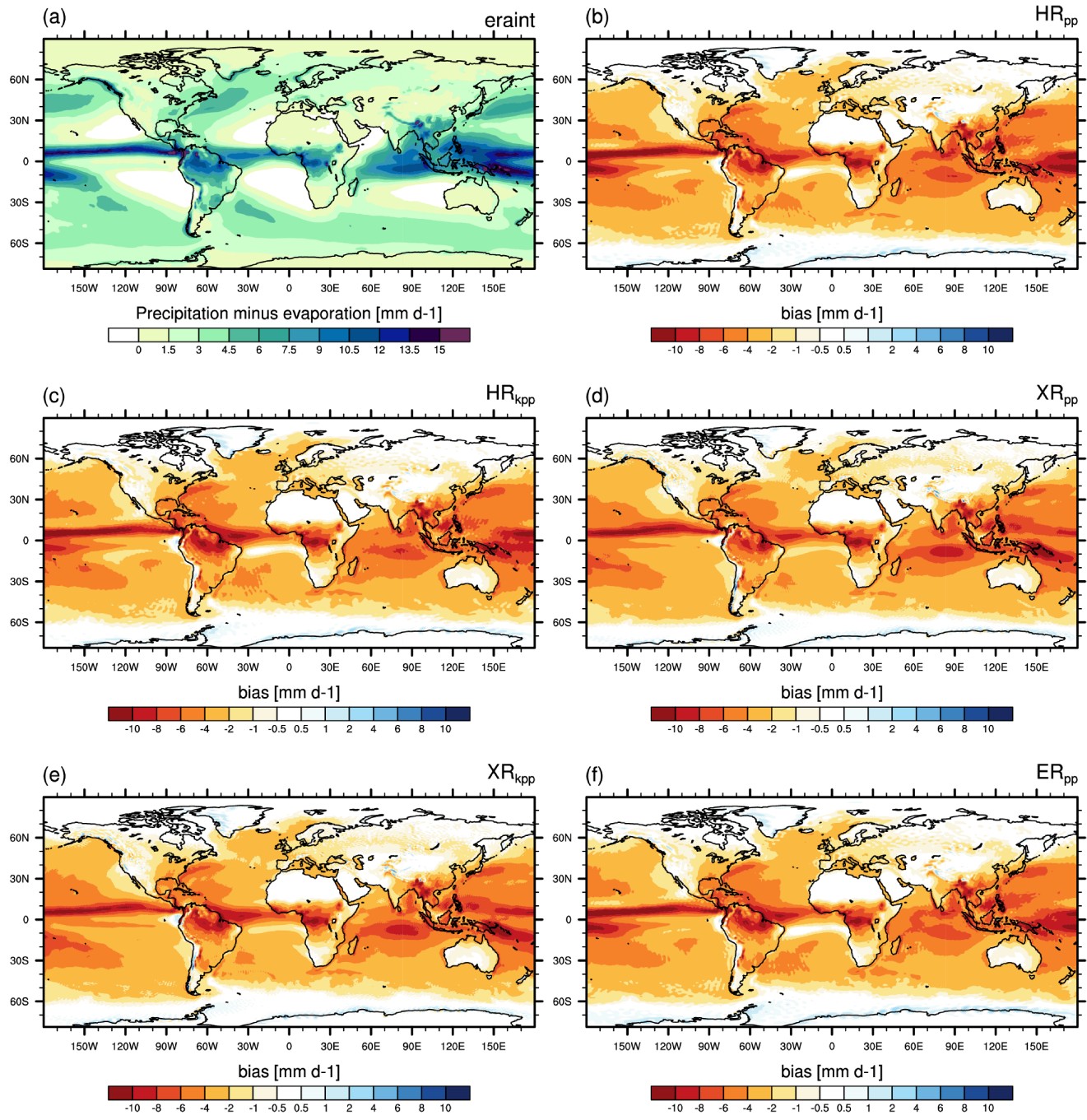

**Figure A2.** Time-averaged precipitation minus evaporation from (a) ERA-Interim (1979–2005) and the bias of: (b) HR$_{pp}$, (c) HR$_{kpp}$, (d) XR$_{pp}$, (e) XR$_{kpp}$, and (f) ER$_{pp}$.

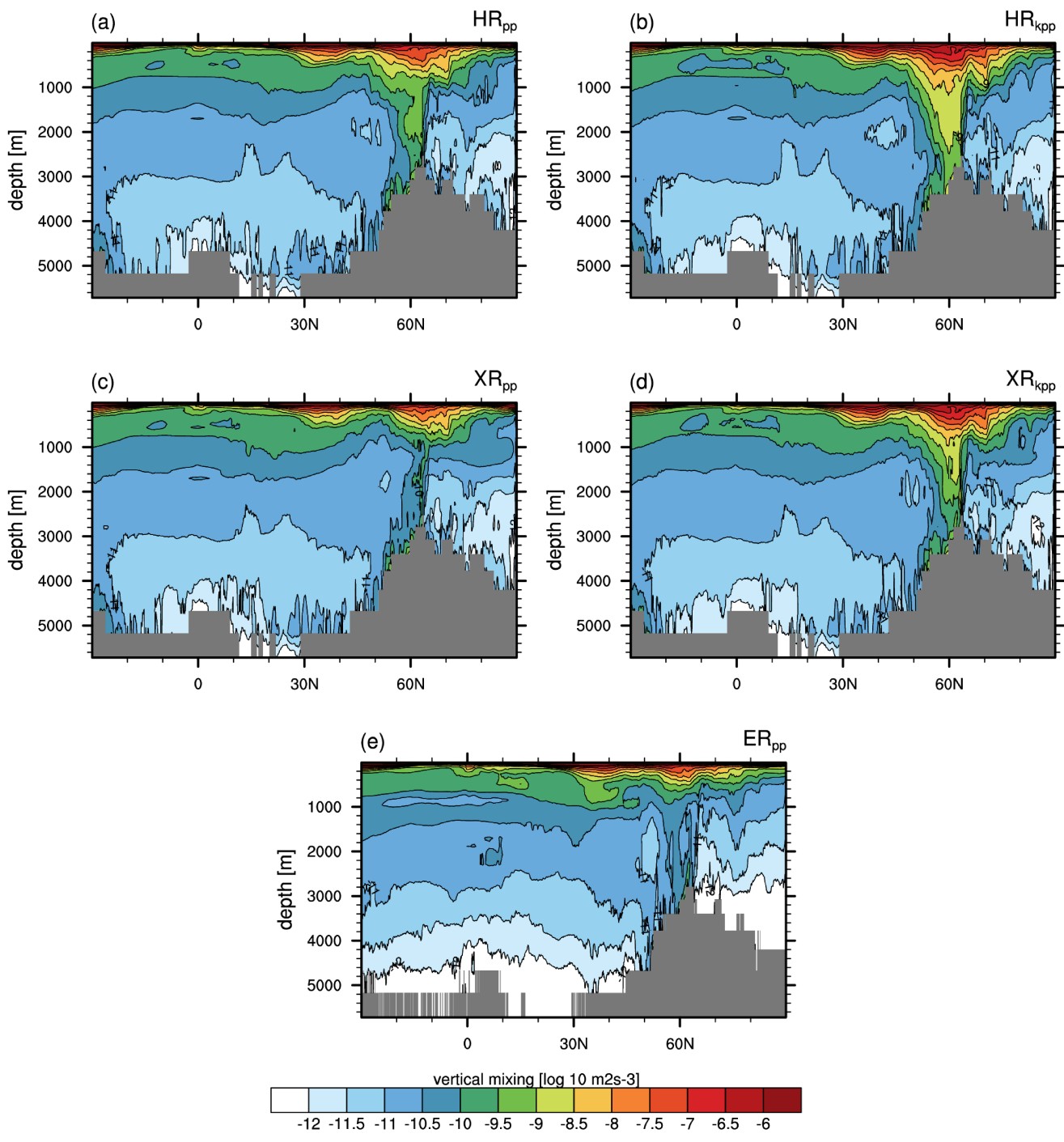

**Figure A3.** Transect of zonal mean vertical mixing ($log_{10}(k_v N^2)$) through the Atlantic basin and the Arctic Ocean of (a) $HR_{pp}$, (b) $HR_{kpp}$, (c) $XR_{pp}$, (d) $XR_{kpp}$, and (e) $ER_{pp}$.

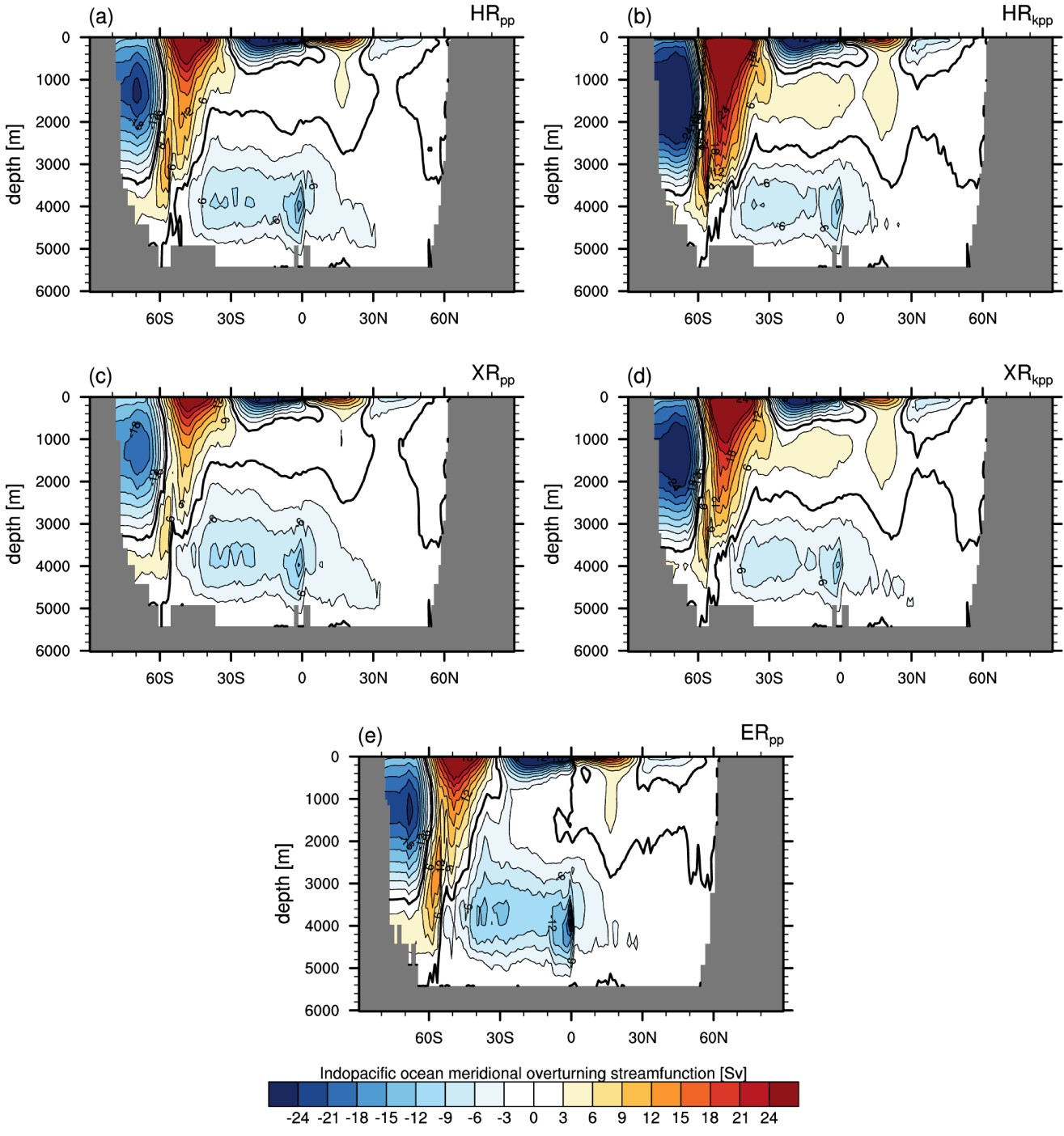

**Figure A4.** Eulerian stream function ($Sv := 10^6 \, m^3 \, s^{-1}$) of the Pacific meridional overturning circulation for (a) $HR_{pp}$, (b) $HR_{kpp}$, (c) $XR_{pp}$, (d) $XR_{kpp}$, and (e) $ER_{pp}$. The zero contour is drawn as a thicker line.

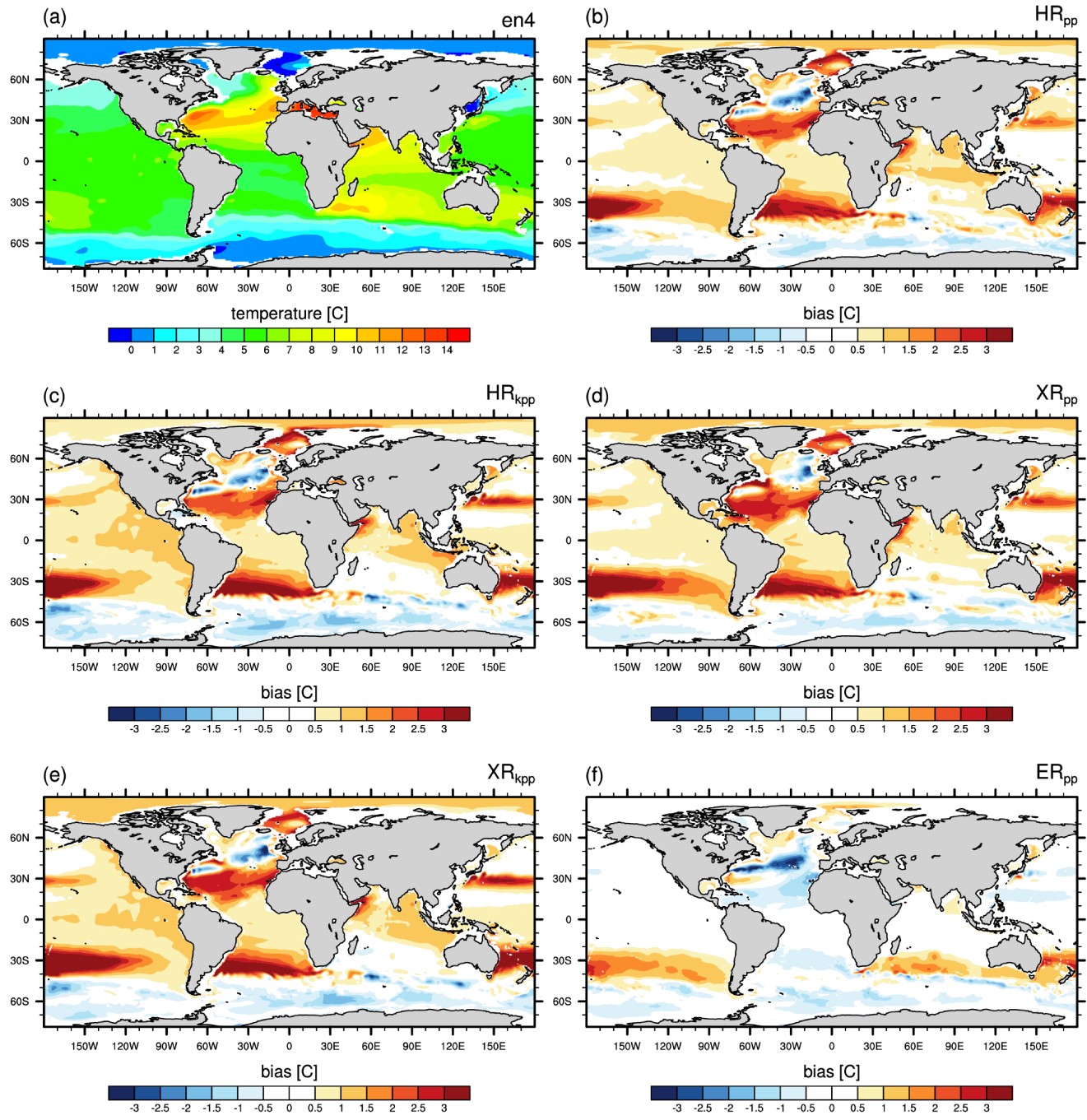

**Figure A5.** Sea water potential temperature ($^{\circ}$C) at a depth of 740 m from (a) EN4 (averaged over 1945–1955) and differences: MPI-ESM1.2 minus EN4 for (b) $HR_{pp}$,(c) $HR_{kpp}$, (d) $XR_{pp}$, (e) $XR_{kpp}$, and (f) $ER_{pp}$.

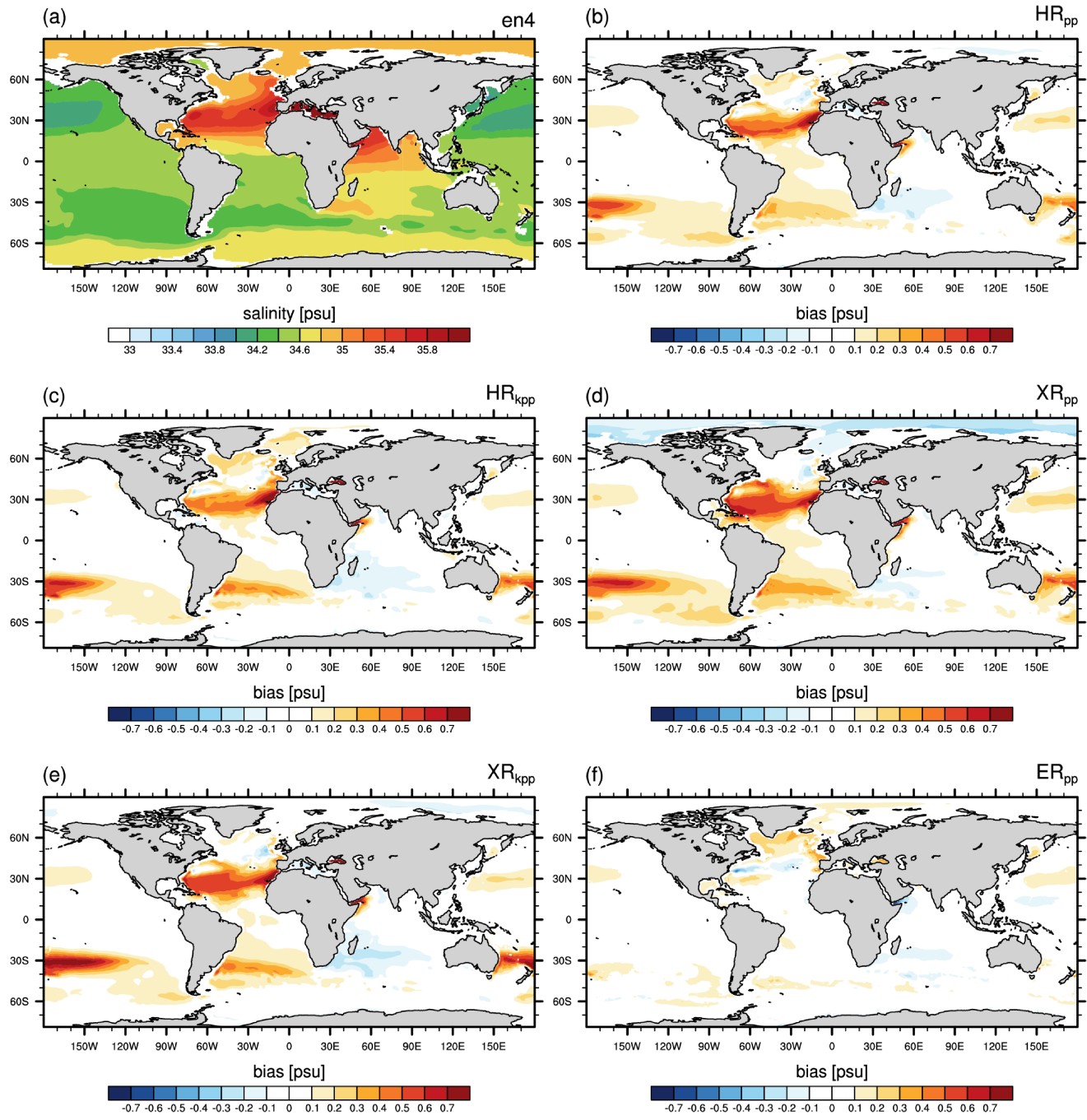

**Figure A6.** Sea water salinity (psu) at a depth of 740 m from (a) EN4 (averaged over 1945–1955) and differences: MPI-ESM1.2 minus EN4 for (b) HR$_{pp}$,(c) HR$_{kpp}$, (d) XR$_{pp}$, (e) XR$_{kpp}$, and (f) ER$_{pp}$.

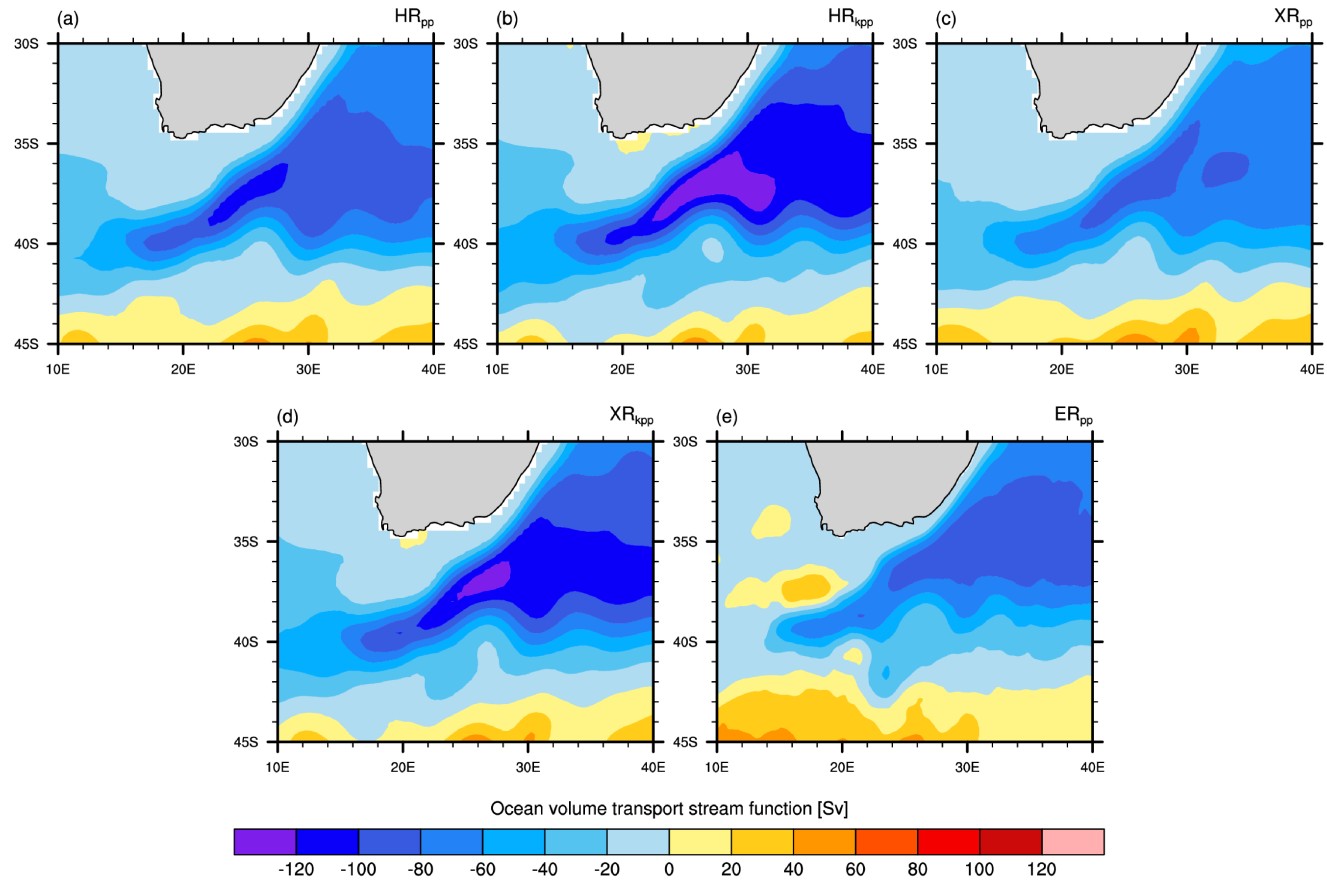

**Figure A7.** Time-averaged barotropic volume transport (Sv) stream function of the Agulhas Current system simulated by (a) HR_pp, (b) HR_kpp, (c) XR_pp, (d) XR_kpp, and (e) ER_pp.

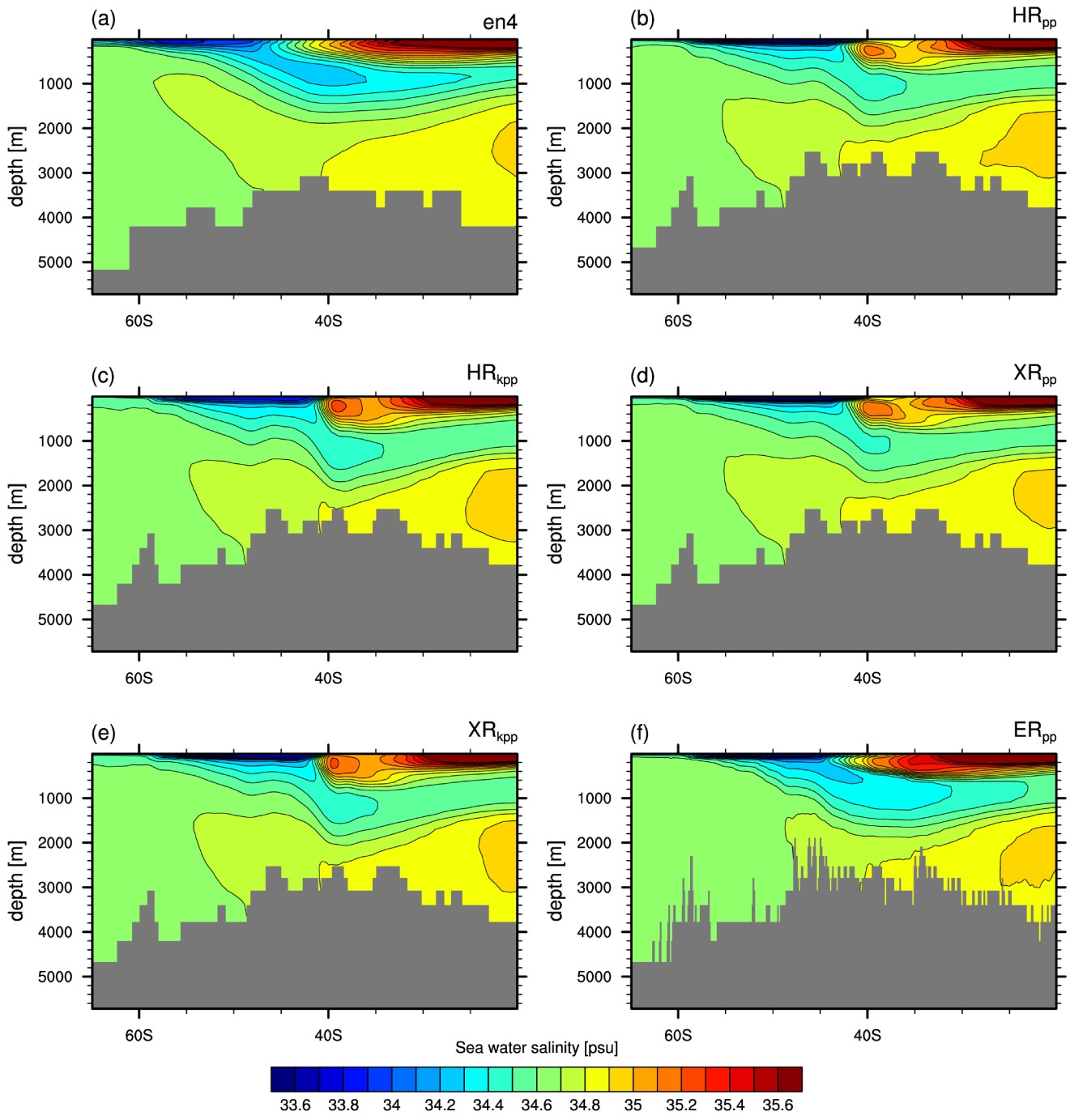

**Figure A8.** Time-averaged salinity section along $15\,^\circ$W (from $65\,^\circ$S to $20\,^\circ$S) in the Southern Ocean of (a) EN4 (1945-1955), (b) HR$_{pp}$, (c) HR$_{kpp}$, (d) XR$_{pp}$, (e) XR$_{kpp}$, and (f) ER$_{pp}$.

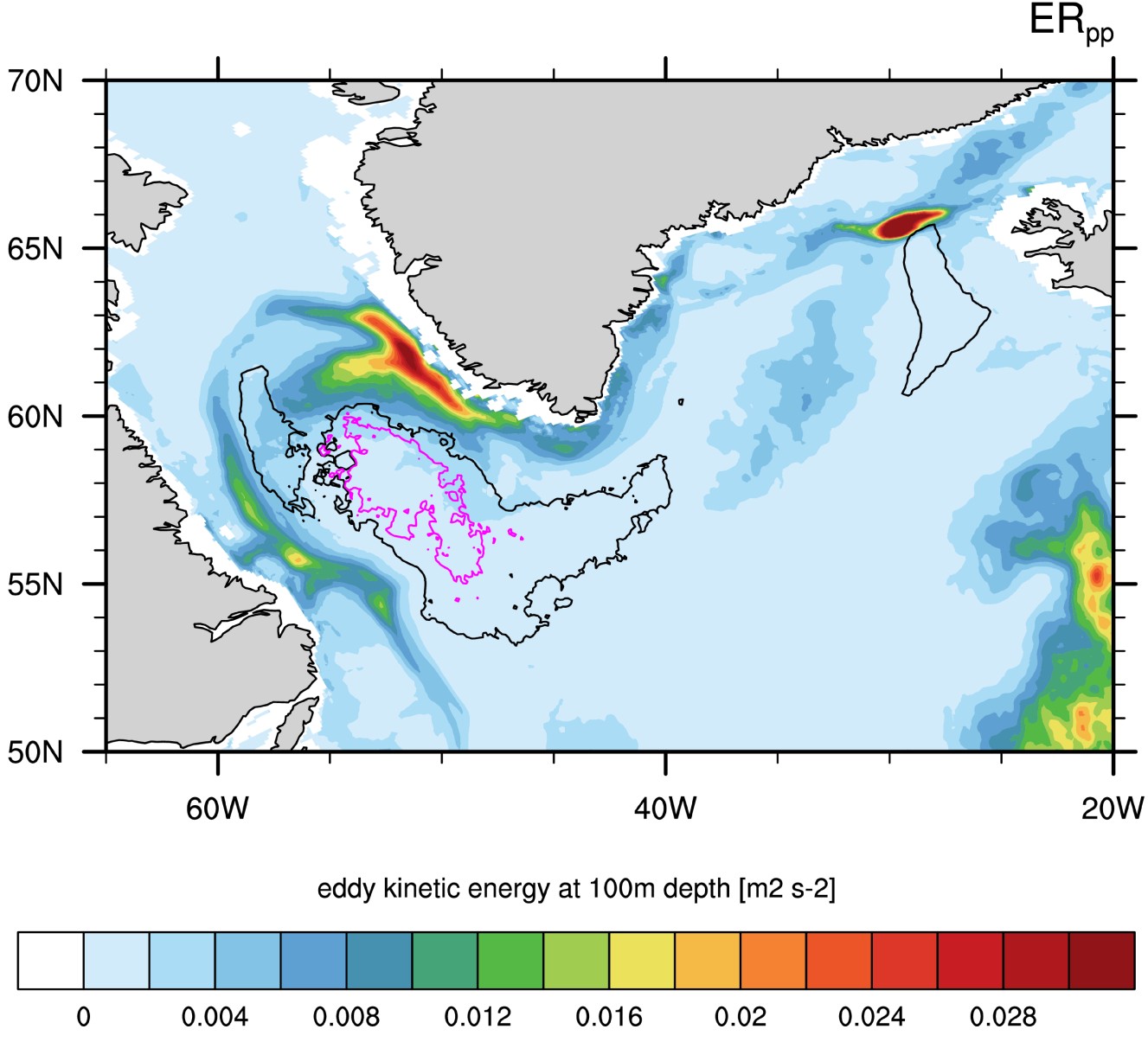

**Figure A9.** Time-averaged eddy kinetic energy ($eke = \frac{1}{2}(\overline{u'^2 + v'^2})$ in $\mathrm{m^2\,s^{-2}}$) in March simulated by $\mathrm{ER_{pp}}$. In addition, the mean mixed layer depths in March are shown as contour lines (black: 500 m and magenta: 1000 m).

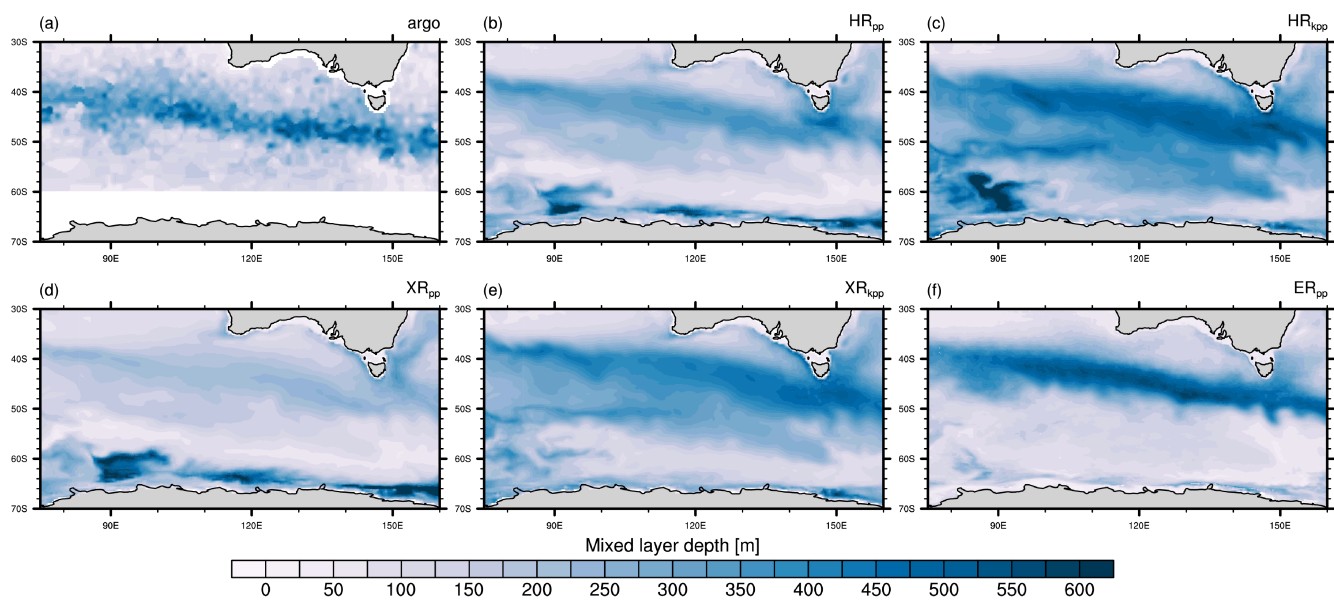

**Figure A10.** Time-averaged mixed layer depths ($\sigma_t = 0.03\,\mathrm{kg\,m^{-3}}$) across the Subantarctic Frontal zone in September from (a) $1° \times 1°$ gridded Argo float data (Holte et al., 2017) and from (b) $\mathrm{HR_{pp}}$, (c) $\mathrm{HR_{kpp}}$, (d) $\mathrm{XR_{pp}}$, (e) $\mathrm{XR_{kpp}}$, and (f) $\mathrm{ER_{pp}}$.