# Peer review of "Max Planck Institute Earth System Model (MPI-ESM1.2) for High-Resolution Model Intercomparison Project (HighResMIP)"

_Geoscientific Model Development, 2018_

## Referee Comment (RC1) · Anonymous Referee #1 · 17 Feb 2019

The paper compares climate simulations with MPI-ESM1.2 under HighResMIP protocol. The roles of different resolutions in both, ocean and atmosphere as well as the effects from using two vertical mixing schemes in the ocean, namely PP and KPP, are addressed. The work presents a big value for the climate modelling community and contains material which is worth publishing in GMD. I recommend this paper for publishing after minor revisions. Please see below my comments and concerns.

First of all I was positively surprised seeing this paper submitted to GMD because to my knowledge the developments of both, ocean and atmosphere components reported here were announced to be discontinued. I still believe that a lot of things can be done

[Figure]

using this system.

Regarding MPIOM in the middle of the chapter 2.1 describing the setups I was surprised by the choice of the the GM coefficient which is 250 m2/s for 400km cell width. I find this value too small as compared to what is practiced by other models. For instance, in paper by Marshall, J et at. 2017 which is cited by the authors the value of 850 m2/s is reported for 1° resolution in the control run. It is about 13 times larger than what is used in this paper. The same, although not that extend, is true for the Redi diffusion. All this implies that the mesoscale eddies are basically neither parameterized nor resolved in simulations denoted with HR and XR. Going through other papers using MPIOM I found that the choice made by the authors is indeed canonical for this model. From this one may conclude that the baroclinic instability in low resolution setups is "indirectly parameterized" by some diabatic processes taking place in the model. These can be for instance the explicitly specified or numerical diffusion. According to Marshall, J et al. 2017 the MOC/AMOC are far too large when the small values are used although in the ocean only configurations.

XR_pp is characterized by a very small AMOC due to very low winds in ECHAM/T255 and associated collapse of the deep convection in the Labrador Sea. A similar behaviour regarding Labrador Sea MLD collapse has already been reported for several completely different climate models. In line 21, page 8 the authors discard the effect from the Southern Hemisphere saying that the collapse happens on a fast timescale. Although I tend to believe that the authors are right one still could speculate that the time range from 50 to 100 years is already not that fast. The colorbar range in Figure 5 is huge but one could see that the cold bias over ACC is coincidently the largest in XR_pp. Alternatively one may look at the difference between subplots in Figure 8 to see whether the slope of the halocline in the SO changes between simulations. Otherwise it is hard to guess it by eye. How do the interranual timeseries of the MLD in the Labrador Sea look like? After which year does the first Labrador Sea MLD collapse happen?

In Figure 9 I would expect to see the sea ice in the Labrador Sea in XR_pp. Is it masked by the choice of colorbar. Honestly, I would add even more material regarding the Labrador Sea collapse as it has been often discussed in literature. See for instance the paper by Moriaki Yasuharaet al. 2008 in PNAS titled as "Abrupt climate change and collapse of deep-sea ecosystems".

The authors solved the problem with too cold North Atlantic in XR_pp by changing the mixing scheme from PP to KPP. I speculate that it would be sufficient to increase the upper mixing coefficient in PP to parameterize for the wind induced turbulence but still encourage using KPP as there is more physics in it.

Even though XR_kpp is a reasonable simulation, the NADW is still not as well simulated as in ER_pp where the bottom cell in AMOC is nicely visible (Fig. 12). Most probably, playing with GM is still required in order to further improve the quality of XR_kpp. So far if all I wrote above regarding GM is correct, ER_pp is the only simulation which physically consistent deals with the baroclinic instability. Surprisingly, ER_pp looks fine even with PP.

In addition to showing the differences to climatology at the surface I would also suggest to plot these differences at other depths (1000 or 2000 meters). The drawback of using KPP might be that it propagates the model error further to the deeper ocean than it was with PP.

I would suggest the authors to elaborate a bit more on the text considering what I have written above. It will be not that descriptive then. The text is easy to read but I would recommend that some native speaker will read it. I found sentences containing things like "cold bias in the North Atlantic improves" in the text.

---

## Referee Comment (RC2) · Anonymous Referee #2 · 26 Feb 2019

The paper examines a contribution to the HighResMIP project from the MPI. The climate state in MPI-ESM with different atmosphere and ocean resolutions are examined. The paper is quite clear and thorough. Major results include the fact that increasing atmosphere horizontal resolution degraded some aspects of the solution (forming a cold bias in NH) , but this was fixed by changing the ocean vertical mixing scheme: and that using an eddy-resolving ocean fixed much of the remaining bias in ocean and atmosphere temperature. These results (when well-explained) are valuable to the community trying to decide where to invest computer time: on different resolution or ensembles or improved physics. Another aspect of interest to the community is the freeze over of the Labrador Sea and steps made to alleviate it.
I have some comments, one of which could involve some major restructuring/re-ordering, others more minor.

Major

The paper starts by discussing wind, Tair and SST biases etc., and refers to many results shown later in the paper, such as the following in section 3.1.2: line 2 of page 6 "and briefly outlined in section 4" line 18-19 North Atlantic Current (section 4.1.1) line 25 (see section 4.5) line 29-30 (see section 4) The references to later in the paper are not ideal, and I think it would read better if either i) these references were removed or, probably much better, ii) the paper re-arranged. As many properties are related to AMOC, you could start with a discussion of the ocean circulation (4.3) followed by 4.6 (AMOC) followed by 4.5 (Mixed layer Depth) followed by 4.4 (Sea ice). Then perhaps describe the rest of the ocean state (sections 4.1, 4.2) and then move to section 3 (Atmosphere state). Hopefully this or something similar would make the paper flow better.

Minor comments 0. Some of the English language style could be improved. Please check thoroughly.

1. As the acronyms for model resolution are a bit hard to follow, you should refer to Table 1 much earlier, on page 2 (section 1) and again in 2.1. You may also want to clearly state: "atmosphere resolution is contrasted between HR and XR: ocean resolution between HR and ER or between XR and ER: ocean mixing physics between pp and kpp.

2. Page 5, lines 9-10. "diffusivities not matched at base of mixed layer" Do you think that this has any adverse effects?

3. line 20. Presumably you refer to the time mean of scalar wind speed, not the magnitude of the time-mean wind vector?

4. line 26. Changes in subtropics are very small, perhaps delete the comment

page 6, line 18 "partly caused" - I don't think it is solely the NAC problem.

page 7, line 1. I think it should be compared to HRpp, not HRkpp.

line 19. "does not modify the mean zonal wind " where are you referring to?

line 25. Another reference to a later section...

Fig. 4. Please add more contour levels to b-f), such as some combination of +/- 0.25, 0.5, 0.75K.

Line 28. "We conclude that eddies play a major role" - but it could be other processes, such as resolution of boundary currents, and eddies could play a direct or indirect (via effect on mean flow) effect

page 9 lines 12-15. This explanation is not backed up by results (of salt transport). It could be put in Discussion.

line 25. "less" to "little"

line 26-29. It would be useful to include maps of precipitation (P) and evaporation (E) , or P-E to i) look at their biases and ii) see if they relate to ocean salinity bias.

page 10, lines 8-12. We also see big effects of improving the Agulhas in our ocean simulation at different resolutions. I would like to see a map of the mean ocean currents (or just zonal velocity) in the Agulhas/Retroflection region in HR and XR and ER. Also, do you see similar changes (ER relative to HR/XR) in other basins?  If not, it may discount the eddy-induced cooling hypothesis.

Also, it may be useful to show (as Supp. Figs) the temperature and salinity biases globally at around 700m depth.

page 11 line 24. Delete "slightly", it looks large.

line 26 add "compared to HRpp" at end of line

line 32-33. I did not follow "explain the positive salinity bias in N. Atlantic" Where and

why?

Page 12 line 12. "model resolution or using KPP, which increase..."

Fig. 9 The Lab. Sea freeze-over is hard to see - the south-west labrador Sea has small sea ice volume. Perhaps you can also show sea-ice extent, if only for case XRpp?

On this subject, other climate model centers have battled with Lab. Sea freeze-over. For example, in Community Earth System Model 2 development, several cases obtained freeze-over. CESM uses kpp. So, you show it is sensitive to using pp or kpp, while with CESM it seems to be very sensitive to run-off and surface salinity. (It is not fully understood).

page 15, line 2. "in XRkpp" compared to?

Line 11-13. Convection in the Labrador and Irminger Sea is governed by a number of complex factors in addition to those listed. The effect of eddies may be quite different in the eddy-permitting case than in the eddy-resolved case. An idea of the complexity of the eddies and convection can be found from Kawasaki and Hasumi 2014 (Ocean Modelling) and DuVivier et al 2016 (J. Climate).

Section 4.5.2. Although I understand you want to focus on high latitude convection and relationship to AMOC, it would also be interesting to look at MLD in the SubAntatarctic Frontal Zone (which ranges from latitudes of 40deg. in the South Indian Ocean to 60deg. near Drake Passage). Standard resolution models have a shallow MLD bias (Sallee et al 2013, DuVivier et al 2018, JGR Oceans https://doi.org/10.1029/2018JC014275) but there is a hint that high resolution models do better (Lee et al 2011, 24, 3830-3849, J. Climate, Li and Lee 2017, 47, 2755-2772, J. Climate.)

Page 17, line 32, "featuring reduced surface winds" where?

Line 24. "winter upper layer"?

Page 19, line 15-16. This sentence is awkward. It could be reworded like "Cold temperature biases in the Southern Hemisphere, and to a lesser extent in the Northern Hemisphere, are reduced"

Line 26-27. But doesn't XRpp, with T255 , have negligible MLD in the sub-polar gyre?
* * *

---

## Author Comment (AC1) · 16 Apr 2019

**1   Answers to reviewer 1**

First of all, we thank reviewer 1 for the positive attitude to continuing model developments of MPI-ESM. It is however true that there will be no further developments on MPI-ESM1.2 (all effort is on the new ICON model), except for advanced vertical mixing schemes in the ocean, which have considerable effects on the ocean circulation that might become of importance also for other high-resolution earth system models.

1. **RC1**: "Regarding MPIOM in the middle of the chapter 2.1 describing the setups I

was surprised by the choice of the the GM coefficient which is 250 m2s-1 for 400 km cell width. I find this value too small as compared to what is practiced by other models. For instance, in paper by Marshall, J et at. 2017 which is cited by the authors the value of 850 m2/s is reported for 1° resolution in the control run. The same, although not that extend, is true for the Redi diffusion. All this implies that the mesoscale eddies are basically neither parameterized nor resolved in simulations denoted with HR and XR. Going through other papers using MPIOM I found that the choice made by the authors is indeed canonical for this model. From this one may conclude that the baroclinic instability in low resolution setups is "indirectly parameterized" by some diabatic processes taking place in the model. These can be for instance the explicitly specified or numerical diffusion. According to Marshall, J et al. 2017 the MOC/AMOC are far too large when the small values are used although in the ocean only configurations."

**AC**: There are two reasons why we chose a value of 250 m2s-1 for a 400km grid cell for the GM coefficient. First, we did not want to change this GM-parameter to be consistent with previous MPI-ESM simulations. Second, the GM coefficient was used to tune the strength of the AMOC, which was too weak in the TP04 configuration when higher values were used (as mentioned by von Storch et al. 2016). Finding an optimal configuration is, however, challenging for intermediate resolutions and is tackled differently by individual modelling centres. We note that mesoscale variability (eddies) is partly resolved and partly parameterized in our HR/XR models, which can be seen by looking at e.g. eddy kinetic energy.

We admit that the GM coefficient is lower compared to what is used by Marshall et al. (2017); however, there are strategies to turn off the GM parameterization completely for such hybrid resolutions (see for instance the GFDL model, where the GM parameterization was switched off (Delworth et al., 2012; doi=10.1175/JCLI-D-11-00316.1). In fact, Treguier (2006; Ch.3 in Chassignet and Verron (Eds.) Ocean Weather forecasing, p.100-102) discusses this issue. Although a full GM parameterization is justified for e.g. a 1/3° model, for instance in the Labrador Sea, it is not suited for all parts of the
ocean because of the varying Rossby radius. A value of 800 m$^2$s-1 for instance would destroy the eddy activity in the Gulf Stream. Thus it is still an open question how to represent the unresolved mesoscale eddy spectrum in eddy-permitting ocean models.

MPIOM uses a weighted average of both central-differences and upstream methods (Marsland, 2003). It exploits the benefits of the upstream scheme to limit spurious tracer sources and sinks, while it avoids numerical diffusion in areas with strong gradients in the tracer field. Usually the central-differences scheme has more weight, except in cases where strong gradients occur. In such a case, the numerical mixing might still be too large. However, as mentioned above, the resulting AMOC is in good agreement with observations, so that we did not change the GM coefficients for our simulations.

2. **RC1**: "XRpp is characterized by a very small AMOC due to very low winds in ECHAM/T255 and associated collapse of the deep convection in the Labrador Sea. A similar behaviour regarding Labrador Sea MLD collapse has already been reported for several completely different climate models. In line 21, page 8 the authors discard the effect from the Southern Hemisphere saying that the collapse happens on a fast timescale. Although I tend to believe that the authors are right one still could speculate that the time range from 50 to 100 years is already not that fast. The colorbar range in Figure 5 is huge but one could see that the cold bias over ACC is coincidently the largest in XRpp. Alternatively one may look at the difference between subplots in Figure 8 to see whether the slope of the halocline in the SO changes between simulations. Otherwise it is hard to guess it by eye. How do the interranual timeseries of the MLD in the Labrador Sea look like? After which year does the first Labrador Sea MLD collapse happen?"

**AC**: We discard the Southern Ocean (SO) to be the reason for the collapse of the AMOC and rely on the study of Putrasahan et al. (JAMES, now accepted), who explicitly demonstrated that the SO is not the reason for the collapse. We did not go into details here, because the Putrasahan et al. paper is all on exploring the collapse of the Labrador Sea convection that occurs within the first 20 years of the simulation (Fig.7

in their manuscript). Unfortunately, the Putrasahan et al. paper was not publically accessible at the time we wrote our manuscript. A brief note on the topic: the weaker westerlies over the SO decrease the upwelling in the SO which in turn reduces the return flow and thus the AMOC strength. In their paper, they performed a sensitivity simulation in which the wind bias over the SO was corrected (by adding the mean wind stress from HR) to test this hypothesis. As a result, they find only a negligible effect from the SO and conclude that biases from the SO might affect the AMOC only on longer time-scales, but not within a few decades. From Fig. A8 (revised manuscript; A4 in the first submission), we conclude that the slopes of the halocline are similar in all HR and XR simulations, only in ER is the slope less steep, which better agrees with EN4 data. Additionally, the cold bias in the SO remains even when using KPP in the XR setup, which indicates that the sensitivity of deep convection in the Labrador Sea is mainly responsible for the stability of the AMOC in XR. The time series of the mixed layer depth in the Labrador Sea (averaged over 58-48°W, 55-65°N, attached Fig. 1) show that the mixed layers in XR-PP are shallower than in HRpp in the first two decades, and drop to almost zero afterwards.

3. **RC1**: "In Figure 9 I would expect to see the sea ice in the Labrador Sea in XRpp. Is it masked by the choice of colorbar. Honestly, I would add even more material regarding the Labrador Sea collapse as it has been often discussed in literature. See for instance the paper by Moriaki Yasuharaet al. 2008 in PNAS titled as "Abrupt climate change and collapse of deep-sea ecosystems"."

**AC**: Thank you very much for this comment. We have added the contour lines for the 15% sea-ice concentration from the simulations (magenta) and from the EUMETSAT OSI SAF observational product (dark blue) to Fig. 9 (attached Fig. 2). We also have slightly adjusted the colour bar range. We decided to not include a discussion of the impacts of a Labrador Sea deep-convection collapse into the manuscript, because it is not the topic for this overview manuscript. Thus, although the Yasuharaet et al. 2008 paper is of interest for sure, it is to specific (ecosystems) for our purpose.

4. **RC1**: "The authors solved the problem with too cold North Atlantic in XRpp by changing the mixing scheme from PP to KPP. I speculate that it would be sufficient to increase the upper mixing coefficient in PP to parameterize for the wind induced turbulence but still encourage using KPP as there is more physics in it."

**AC**: Although we did not explicitly perform such an experiment, we think it will not solve the collapse. Putrasahan et al. performed a sensitivity experiment where they increased the wind stress received by the ocean by a factor of 1.5, which is comparable to increasing the wind mixing part of the PP scheme. It did however not rescue the AMOC. Therefore, and because the KPP scheme was on our agenda anyways, we decided to only perform a KPP simulation with XR. We speculate that the non-local transport terms in KPP better reflects the convection (probably due to entrainment of salt in the upper ocean during absence of wind forcing) than the enhanced wind-mixing parameterization for the upper ocean diffusivity used with the PP scheme. Furthermore, the diffusivity in the KPP scheme also depends on the mixed layer depth to reflect that boundary layer eddies become larger with deeper mixed layers. This dependency might further cause higher diffusivities than with the PP scheme, which then sustains sufficient mixing and convection in the Labrador Sea.

5. **RC1**: "Even though XRkpp is a reasonable simulation, the NADW is still not as well simulated as in ERpp where the bottom cell in AMOC is nicely visible (Fig. 12). Most probably, playing with GM is still required in order to further improve the quality of XRkpp. So far if all I wrote above regarding GM is correct, ERpp is the only simulation which physically consistent deals with the baroclinic instability. Surprisingly, ERpp looks fine even with PP."

**AC**: That is correct; the bottom cell is well simulated in ERpp. A comparison with HRpp suggests that the higher ocean resolution improves the bottom cell, while the KPP scheme does not. Both use the same atmosphere (T127) which does not produce the negative wind bias of the T255 atmosphere, so that a stable AMOC is simulated also with the PP scheme. It is likely that further retuning (including GM) is required

for the XRkpp to improve the bottom cell; however, the HighResMIP protocol has a strong preference for not retuning of the models in order to see the pure effects coming from either increased resolution or improved phyiscs (KPP scheme in our case). This procedure received a consensus by all modelling groups within the HighResMIP. If not, the effect of subgrid-scale parameterization of ocean eddies would obscure the effect and benefit of higher model resolution.

6. **RC1**: "In addition to showing the differences to climatology at the surface I would also suggest to plot these differences at other depths (1000 or 2000 meters). The drawback of using KPP might be that it propagates the model error further to the deeper ocean than it was with PP."

**AC**: We plotted the temperature and salinity bias (attached) at 1000m and 2000m depth (see attached Fig. 3 and Fig. 4). As also shown in Fig. 7 and Fig. 8 in the manuscript (zonal mean transects), the bias at depth is of similar magnitude in XRkpp and XRpp. However, at 1050m depth (attached Fig. 3) both HRkpp and XRkpp enhance the warm bias in the Southern Ocean around 30°S and in the Indian Ocean and Pacific. In the North Atlantic, however, the bias is similar to HRpp. At 2080m depth (attached Fig. 4) the difference of the bias between PP and the KPP simulations is much smaller. Concerning salinity, the bias of HRpp and HRkpp are comparable, with HRkpp slightly saltier in the subpolar gyre and in the GIN seas (attached Fig. 5). Although similar for large parts of the ocean, the positive salinity bias in XRpp is, however, slightly less in than in XRkpp. This salinity reduction along with a reduced wind forcing is then the reason for the collapse of deep-convection in the Labrador Sea. In addition, the Arctic Ocean freshens in XRpp, so that also more freshwater is exported into the North Atlantic, further reducing the deep-convection. At 2080m depth (attached Fig. 6) the salinity bias mostly vanishes, except for a positive bias in the subpolar gyre. The bias is slightly stronger in the HR and XR simulations with KPP, which might be due to saltier Mediterranean outflow waters into the North Atlantic. It certainly helps to sustain deep-convection in the subpolar gyre in addition to the

physically more realistic KPP scheme convection.

7. **RC1**: "I would suggest the authors to elaborate a bit more on the text considering what I have written above. It will be not that descriptive then. The text is easy to read but I would recommend that some native speaker will read it. I found sentences containing things like "cold bias in the North Atlantic improves" in the text."

**AC**: Thank you very much, we will add content following your comments and we hope we did correct the flaws in the text.

**2   Answers to reviewer 2**

We thank the reviewer for the constructive comments and thorough reading of our manuscript.

**2.1   Major comment**

1. **RC2**: "The paper starts by discussing wind, Tair and SST biases etc., and refers to many results shown later in the paper, such as the following in section 3.1.2: line 2 of page 6 "and briefly outlined in section 4" line 18-19 North Atlantic Current (section 4.1.1) line 25 (see section 4.5) line 29-30 (see section 4) The references to later in the paper are not ideal, and I think it would read better if either i) these references were removed or, probably much better, ii) the paper re-arranged. As many properties are related to AMOC, you could start with a discussion of the ocean circulation (4.3) followed by 4.6 (AMOC) followed by 4.5 (Mixed layer Depth) followed by 4.4 (Sea ice). Then perhaps describe the rest of the ocean state (sections 4.1, 4.2) and then move to section 3 (Atmosphere state). Hopefully this or something similar would make the paper flow better."

**AC**: In our initial draft (before submission), we used a structure as you propose. In the course of writing however, we changed the structure to what it is now. The reason is that from a narrative point of view, we find it easier to follow when the atmosphere is described first, with its reduction of the near-surface wind speeds in the XR setup, which is independent of the ocean vertical mixing scheme or the resolution of the underlying ocean model. This wind reduction then forms the base from which we develop the collapse of the AMOC with the PP scheme, while with KPP the AMOC remains stable. However, as we are dealing with coupled models, it is hard to avoid anticipating any information that is presented in detail later in the text. For instance, if we began describing the AMOC slowdown, we would have to anticipate that wind forcing reduces with T255, which is then shown in a later section. We are grateful to the reviewer for thinking the structure of the paper through and giving us these suggestions. In the end it is probably a matter of personal taste where to describe the complex coupled system. We hope this explains why we started with the atmosphere. You can rest assured that we have long thought about the structure of the paper. The initial references to later in the paper may not be ideal, but together with the additional information we provide at this stage, it is sufficient information to understand the consequences for the atmosphere, and it prepares the reader for what to expect in the course of the manuscript.

**2.2 Minor comments**

1. **RC2**: "As the acronyms for model resolution are a bit hard to follow, you should refer to Table 1 much earlier, on page 2 (section 1) and again in 2.1. You may also want to clearly state: "atmosphere resolution is contrasted between HR and XR: ocean resolution between HR and ER or between XR and ER: ocean mixing physics between pp and kpp."

**AC**: We agree and we now refer to Table 1 much earlier (first half of Introduction) and make explicitly clear what the difference is between the settings (summary at the end

of the Introduction).

2. **RC2**: "Page 5, lines 9-10. "diffusivities not matched at base of mixed layer" Do you think that this has any adverse effects?"

**AC**: A match of the diffusivities at the base of the ocean boundary layer (OBL) in KPP allows that interior processes can influence the diffusivity within the OBL, which adds complexity to the KPP scheme. As this matching differs for every tracer, it also implicates that the non-dimensional shape function (G) in the KPP scheme is no longer a universal function, but depends on the field that is transported. In addition, it is not straightforward how to do this smoothing for a discretized vertical ocean grid. It also means that the non-local transport terms do not smoothly go to zero at the base of the mixed layer. In unfavourable conditions, it was found that the non-local fluxes can even become larger than the surface fluxes (Griffies, 2013) near the OBL base, even though they are supposed to just redistribute the surface fluxes in the vertical. This 'overshooting' then might produce extrema in the tracer field. One example where this might happen is when in unstable conditions below the OBL the enhanced diffusivity parameterization results in very large diffusivity just beneath the OBL. In that case a matching would result in a very large shape function (G > 1.0) at the base of the mixed layer. Although there are simpler techniques of matching, referring to Griffies (2013) they were not fully tested yet.

For these reasons we decided to not match the diffusivities and accept that there might be jumps at the base of the boundary layer, if not both the OBL and the interior diffusivities go to zero at the OBL base. On the other hand, the OBL diffusivities are still influenced by processes related to shear at its base, as the calculation of the OBL depth in KPP involves the resolved and turbulent shear in the Richardson number. One such area is for instance the equatorial Pacific where the near-surface underlying Equatorial Undercurrent has a strong vertical shear that might deepen the OBL and with that enhance the OBL diffusivities, which are a non-local function of OBL depth (and other OBL properties).

Interactive
comment

3. **RC2**: "line 20. Presumably you refer to the time mean of scalar wind speed, not the magnitude of the time-mean wind vector?"

**AC**: Yes we refer to the scalar wind speed here. We have revised the sentence accordingly.

4. **RC2**: "line 26. Changes in subtropics are very small, perhaps delete the comment."

**AC**: We agree the change is very small, and have deleted the sentence.

5. **RC2**: "page 6, line 18 "partly caused" - I don't think it is solely the NAC problem."

**AC**: Yes, absolutely correct. We add your suggestion, and we have added additional reasons for the SST cold bias in the extratropical NA: "Here, all simulations show a cold bias, which is a common error in state-of-the-art ESMs (Randall et al., 2007), which is partly caused by a too zonal North Atlantic Current (NAC) (Dengg et al., 1996), or by insufficient northward heat transport by the AMOC (Wang et al., 2014a). Drews et al. (2015) could demonstrate that correcting the flow field for biases they were able to remove the cold bias in the North Atlantic."

6. **RC2**: "page 7, line 1. I think it should be compared to HRpp, not HRkpp."

**AC**: That is true, we have revised and restructured the sentence.

7. **RC2**: "line 19. "does not modify the mean zonal wind " where are you referring to?"

**AC**: There is a missing part in that sentence for which we apologize. We have revised this paragraph and the one before to clarify our statements.

8. **RC2**: "line 25. Another reference to a later section..."

**AC**: See our general comment to the major point on structure above.

9. Fig. 4. Please add more contour levels to b-f), such as some combination of +/- 0.25, 0.5, 0.75K.

**AC**: We have added more contour lines to the plot as you suggest (attached Fig. 7).

While producing the new figure however, we noticed that we did process only the zonal-mean bias over the Atlantic, and not the global zonal-mean as intended. Thus, we replaced the figure with the correct one (attached Fig. 1). Same error happened for Fig. 3, which we corrected (attached Fig. 8). We have also modified the text in manuscript accordingly, in particular with respect to the cold bias above the Antarctic continent, which was cut off from the figure before.

10. **RC2**: "Line 28. "We conclude that eddies play a major role" - but it could be other processes, such as resolution of boundary currents, and eddies could play a direct or indirect (via effect on mean flow) effect."

**AC**: This is true, we rephrased the sentence emphasizing that the eddy-resolving resolution in general improves the mean-state of the ocean and atmosphere: "We conclude that an eddy-resolving ocean resolution plays a major role for the mean-states of the large-scale temperature distribution in the atmosphere."

11. **RC2**: "page 9 lines 12-15. This explanation is not backed up by results (of salt transport). It could be put in Discussion."

**AC**: That is correct. We have inserted a supplement figure (Fig. A1 in revised manuscript; attached Fig. 9), showing the northward heat and salt transports at every latitude. The figure clearly shows the enhanced northward heat and salt transport in the KPP simulations.

12. **RC2**: "line 25 "less" to "little.""

**AC**: We have corrected the sentence.

13. **RC2**: " line 26-29. It would be useful to include maps of precipitation (P) and evaporation (E) , or P-E to i) look at their biases and ii) see if they relate to ocean salinity bias."

**AC**: We have added P-E maps as a suppl. figure A2 in the revised manuscript (attached Fig. 10). All models simulate too little precipitation or too much evaporation, resulting in

negative biases for most parts of the globe. However, in the western Pacific where the upper ocean is simulated much too fresh in the XR simulations, we do not see much of a difference to HR. On the contrary, the XR models simulate even less precipitation than the HR models. Thus our conclusion that too less salt is advected by the ocean currents.

14. **RC2**: "page 10, lines 8-12. We also see big effects of improving the Agulhas in our ocean simulation at different resolutions. I would like to see a map of the mean ocean currents (or just zonal velocity) in the Agulhas/Retroflection region in HR and XR and ER. Also, do you see similar changes (ER relative to HR/XR) in other basins? If not, it may discount the eddy-induced cooling hypothesis."

**AC**: We have plotted the velocity at 100m depth and the barotropic volume transport stream function of the Agulhas Current system (attached Fig. 11 and 12). ER-PP simulates a swifter and narrower Agulhas Current (attached Fig. 11) and a stronger retroflection (attached Fig. 12; this figure we included now as an suppl. figure (Fig. A8) in the revised manuscript), compared to the HR/XR simulations. This reduces the inflow of warm and salty water from the Indian Ocean into the South Atlantic, which is why the positive salinity bias at 500 to 700m depth is reduced in ER-PP (Fig. 8 of the manuscript). On the other hand, we notice that the simulations with KPP increase the inflow from the Indian Ocean, which is why both the warm bias and the salinity bias enhance slightly. In general, ER-PP simulates swifter and narrower boundary currents in all basins.

15. **RC2**: "Also, it may be useful to show (as Supp. Figs) the temperature and salinity biases globally at around 700m depth."

**AC**: We have added your suggested plots as supplement figures (Fig. A5 and A6 in the revised manuscript) (attached Fig. 13 and 14), and discuss these plots also in the manuscript.

16. **RC2**: "page 11 line 24. Delete "slightly", it looks large."

**AC**: We have removed 'slightly' from the sentence.

17. **RC2**: "line 26 add "compared to HRpp" at end of line."

**AC**: We have completed the sentence.

18. **RC2**: "line 32-33. I did not follow "explain the positive salinity bias in N. Atlantic" Where and why?"

**AC**: We have changed the sentence to "In the case of HRkpp and XRkpp the too strong volume transport of the subtropical gyre might further contribute to the positive salinity bias in the subpolar gyre at a depth of 500 to 1000 m (Fig. 8 and Fig. A6)."

19. **RC2**: "Page 12 line 12. "model resolution or using KPP, which increase..."."

**AC**: We have corrected the sentence.

20. **RC2**: "Fig. 9 The Lab. Sea freeze-over is hard to see - the south-west labrador Sea has small sea ice volume. Perhaps you can also show sea-ice extent, if only for case XRpp? On this subject, other climate model centers have battled with Lab. Sea freeze-over. For example, in Community Earth System Model 2 development, several cases obtained freeze-over. CESM uses kpp. So, you show it is sensitive to using pp or kpp, while with CESM it seems to be very sensitive to run-off and surface salinity. (It is not fully understood)."

**AC**: That is true and we have added the 15% sea-ice extent contour line in Fig. 9 for all models (magenta) and from the EUMETSAT OSI SAF observational product (dark blue) (see attached Fig. 2). We have also slightly reduced the colourbar range. The freeze-over in XRpp is now clearly visible. We think that the Labrador freeze-over, although occurring across different models, is very model specific for its origin. In the XRpp the primary reason is the reduced wind forcing that results in a reduced salinity advection by the subpolar gyre into the Labrador Sea. Freshening from increased surface runoff seems not to play a major role in that respect and Putrasahan et al. could demonstrate that the collapse is caused because of the reduced wind forcing and

salt advection. In that sense, a too fresh upper ocean due to reduced salt input causes the freeze-over in MPI-ESM-XR. In our XRkpp setup, it seems that KPP maintains a stronger overturning, predominantly because of its non-local formulation and the addition of the non-local flux transports. These cause a stronger deep convection that in turn steepens the isopycnals in the Labradror Sea. The steeper ispoycnals sustain a sufficiently strong subpolar gyre, even though the wind forcing is weak, so that a sufficient amount of salt is advected into the Labrador Sea, where in turn the water column overturns due to negative buoyancy. Another aspect might be that our XR setup is probably in a transition zone, where the vertical mixing scheme is important for the AMOC stability. Nevertheless, we also found that the wind bias in ECHAM was resistant to tuning efforts (as shown in Putrasahan et al., 2018, accepted for JAMES), which is also not fully understood yet.

21. **RC2**: "page 15, line 2. "in XRkpp" compared to?"

**AC**: We have corrected the sentence to: "However, even with a T255 atmosphere the resolution is too coarse to fully simulate Greenland tip jets (e.g. Martin and Moore, 2007; DuVivier and Cassano, 2016; Gutjahr and Heinemann, 2018), which impact the triggering of deep convection in the Irminger Sea due to strong associated turbulent heat fluxes."

22. **RC2**: "Convection in the Labrador and Irminger Sea is governed by a number of complex factors in addition to those listed. The effect of eddies may be quite different in the eddy-permitting case than in the eddy-resolved case. An idea of the complexity of the eddies and convection can be found from Kawasaki and Hasumi 2014 (Ocean Modelling) and DuVivier et al 2016 (J. Climate)."

**AC**: We have added a sentence to the revised manuscript: "However, the processes that lead to deep convection in the Irminger Sea are complex, and is still not fully understood how eddies affect the preconditioning/triggering of convection and where their main formation area is (Fan et al., 2013; DuVivier and Cassano, 2016)."

23. **RC2**: "Section 4.5.2. Although I understand you want to focus on high latitude convection and relationship to AMOC, it would also be interesting to look at MLD in the SubAntatarctic Frontal Zone (which ranges from latitudes of 40deg. in the South Indian Ocean to 60deg. near Drake Passage). Standard resolution models have a shallow MLD bias (Sallee et al 2013, DuVivier et al 2018, JGR Oceans https://doi.org/10.1029/2018JC014275) but there is a hint that high resolution models do better (Lee et al 2011, 24, 3830-3849, J. Climate, Li and Lee 2017, 47, 2755-2772, J. Climate.)"

**AC**: Thank you very much for this suggestion. We did not look too much into Southern Ocean mixed layer depths (apart from the very deep mixed layers simulated in the Weddell Sea in MPI-ESM1.2). However, we produced the mixed layer depths plots for the Subantarctic Frontal zone (attached Fig. 15), which we also added as a suppl. Fig. A10 to the revised manuscript. We used the same colors as in the Fig.2 of DuVivier et al. (2018) for a better comparison with the Argo floats MLDs. There clearly is a deepening of the mixed layers with the KPP scheme, probably due to the nonlocal transport terms as suspected by DuVivier et al. (2018), so that the boundary layer penetrates deeper into the stratified, salinity maximum at deeper layers. Further, the higher ocean resolution increases the MLDs as well while confining it to a narrower band, which is in better agreement with the observations from the Argo floats, as shown by DuVivier et al. (2018) (Fig. 2 therein). However, as also noted by DuVivier et al. (2018) for their higher resolved CESM simulation, the mixed layer depths are too deep compared to the Argo float observations. We have added a paragraph at the end of section 4.5.2 discussing these results.

24. **RC2**: "Page 17, line 32, "featuring reduced surface winds" where?"

**AC**: We have corrected the sentence to: "An important finding is that XRkpp simulates a stable AMOC (14.6 Sv), despite the weak wind stress with the T255 atmosphere."

25. **RC2**: "Line 24. "winter upper layer"?"

**AC**: We have corrected the sentence.

26. **RC2**: "Page 19, line 15-16. This sentence is awkward. It could be reworded like "Cold temperature biases in the Southern Hemisphere, and to a lesser extent in the Northern Hemisphere, are reduced"."

**AC**: Thank you we have rephrased the sentence accordingly.

27. **RC2**: "Line 26-27. But doesn't XRpp, with T255, have negligible MLD in the sub-polar gyre?"

**AC**: You are correct, what we meant is in case of XRkpp. We have however rephrased the whole paragraph.

Please also note the supplement to this comment:
https://www.geosci-model-dev-discuss.net/gmd-2018-286/gmd-2018-286-AC1-supplement.pdf
* * *
**Mixed layer depth in Labrador Sea (48–38°W, 55–65°N)**

$\sigma_t = 0.03\ \mathrm{kgm}^{-3}$

HR–PP
XR–PP

mixed layer depth [m]

model year

**Fig. 1.** Time series of the spatially averaged mixed layer depth ($\sigma$_t=0.03 kg m-3) in the Labrador Sea (58-48°W, 55-65°N) for the first 50 models years of HRpp and XRpp.

(a) piomas

(b) HR_pp

(c) HR_kpp

(d) XR_pp

(e) XR_kpp

(f) ER_pp

Sea ice volume per area [m]

0.25 0.5 0.75 1 1.25 1.5 1.75 2 2.25 2.5 2.75 3 3.25 3.5 3.75 4

**Fig. 2.** Time-averaged Arctic sea ice volume in March. The 15% sea ice concentration contour line line is shown in magenta for the MPI-ESM1.2 simulations and in dark blue for the EUMETSAT OSI SAF observation.

[Figure]

**Fig. 3.** Time-averaged bias of simulated potential temperature at 1085m depth (averaged over model years 30 to 80) with respect to EN4 (averaged over 1945-1955).

**Fig. 4.** same as Figure 3 but at 2080m depth.

**Fig. 5.** Time-averaged bias of simulated salinity at 1085m depth (averaged over model years 30 to 80) with respect to EN4 (averaged over 1945-1955).
**Fig. 6.** same as Figure 5 but at 2080m depth.

[Figure]

[Figure]

**Fig. 7.** Global zonally-averaged temperature. The contour lines in b-f span ±0.75 with an interval of 0.5K, and of 1.0K outside that range.

[Figure]

**Fig. 8.** Global zonally-averaged u-velocity. The zero contour line is shown as a thick solid line; negative (positive) contours are dashed (solid).

**(a)** global northward heat transport [PW]

**(b)** global northward salt transport [10^6 kg s–1]

**(c)** Atlantic northward heat transport [PW]

**(d)** Atlantic northward salt transport [10^6 kg s–1]

Legend:
- HR–PP
- HR–KPP
- XR–PP
- XR–KPP
- ER–PP

**Fig. 9.** Time-averaged northward heat (PW) and salt transport (106 kg s-1) in the global ocean (a,b) and in the Atlantic basin (c,d). Note the different scaling in (c) and (d).

[Figure]

**Fig. 10.** Time-averaged precipitation minus evaporation from (a) ERA-Interim (1979-2005) and the bias (MPI-ESM1.2 minus ERA-Interim) of (b) HRpp, (c) HRkpp, (d) XRpp, (e) XRkpp, and (f) ERpp.

**Fig. 11.** Time-averaged velocity at 100m depth of the Agulhas Current system simulated by (a) HRpp, (b) HRpp, (c) HRkpp, (d) XRpp, (e) XRkpp, and (f) ERpp. The length of the reference vector is 0.2 m s-1.

[Figure]

**Fig. 12.** Time-averaged barotropic volume transport (Sv) stream function of the Agulhas Current system simulated by (a) HRpp, (b) HRpp, (c) HRkpp, (d) XRpp, (e) XRkpp, and (f) ERpp.

**Fig. 13.** Sea water potential temperature (°C) at 740m depth from (a) EN4 (averaged over 1945–1955) and differences: MPI-ESM1.2 minus EN4 for (b) HRpp,(c) HRkpp, (d) XRpp, (e) XRkpp, and (f) ERpp.

**Fig. 14.** Sea water salinity (psu) at 740m depth from (a) EN4 (averaged over 1945–1955) and differences: MPI-ESM1.2 minus EN4 for (b) HRpp,(c) HRkpp, (d) XRpp, (e) XRkpp, and (f) ERpp.

[Figure]

**Fig. 15.** Time-averaged mixed layer depths ($\sigma$\_t=0.03 kg m-3) across the Subantarctic Frontal zone in September simulated by (a) HRpp,(b) HRkpp, (c) XRpp, (d) XRkpp, and (e) ERpp.

**Supplement:**

[revised manuscript text omitted]

* * *
[111] removed: slightly

[112] removed: Subtropical Gyre

[113] removed: only

[114] removed: stronger transport

[115] removed: this might explain

[116] removed: North Atlantic.Similar findings results for the Subtropical Gyre

[117] removed: grid

[118] removed: values

found [..[119] ]a systematic increase of the overflow volumes by either increasing the [..[120] ]ocean resolution or using KPP. [..[121] ]In particular the eddy-resolving ocean ($ER_{pp}$) produces more overflow waters. In the case of $HR_{kpp}$, this higher transport is caused by enhanced deep convection in the Nordic Seas, particularly in the Greenland Sea (Fig. 10).

[revised manuscript text omitted]

---

## Referee Report (RR1)

Review of Gutjahr, Putrasahan, Lohmann, Jungclaus, von Storch, Bruggeman, Haak and Stössel, Max Planck Institute Earth System Model MPI-ESM1.2 for High-resolution Model Intercomparison Project, revised for GMD.

The authors have addressed my comments very thoroughly. I think the paper can be accepted subject to the following minor comments:

Page 2, line 18. Start new paragraph here on increasing atmosphere resolution.

Page 4, lines 7-11. I have read the other Reviewers comments on this, and your response, and I think you should put more of your response text into the main text here.

Page 5 line 27. Typo on "configurations"

Page 6. Lines 8-9. "…the wind speed is too weak…" "but not as weak as in …" ?

Line 22. Delete "for biases" Also, did Drews et al partly, or totally, remove the bias?

Page 7, line 3. "due to resolved eddies and better mean flow"?

Page 11, line 7. "24km wide" – how does this compare with the lower resolution, and real bathymetry (approximately, no need to be too precise)

Line 11. "the cold bias at 740m"

Page 12, line 22. Reword "the relative differences among the simulations are similar" to something more easy to understand. (I think I know what you mean.)

Page 13, final sentence. This is a bit speculative – I suggest move to Discussion or delete.

Page 14 line 27 "in general much less sea ice volume"

Figs 10, 11, A9. There are modern estimates of observed MLD, e.g. Holte et al., MIMOC (PMEL) https://www.pmel.noaa.gov/mimoc/ etc, using data from ARGO. Although data sparcity of ARGO is still an issue in some locations, can you add spatial maps of these products to the figures?

Page 16 line 31. According to the literature, ocean eddies act to restratify many areas of the Labrador Sea, leaving a small region of deep mixed layers possibly where eddies are not active. It might be interesting to add a plot of SSH variability to see if ERpp adds anything to the eddy-permitting regime.

Page 18, line 32. Low resolution MLD may only be 200-300m in the same regions in DuVivier et al etc.

Page 19, line 8. You could place Lee et al and Li and Lee references on this line: on the lines above I don't think Small et al 2014 discussed MLD, but you could put (R. J. Small, pers. Comm. 2019)

Line 26. " A possible explanation for this is that the volume transport…" ?

Page 22, line 4-5. The reference to ERpp is out of place as this paragraph is on atmosphere resolution.

Justin Small

---

## Author Response (AR2)

**Answers to Reviewer:**

Dear Justin Small, thank you very much for your constructive and helpful comments. We hope we could improve our manuscript following your recommendations.

**1. RC:** Page 2, line 18. Start new paragraph here on increasing atmosphere resolution.

**AC:** We started a new paragraph as suggested.

**2. RC:** Page 4, lines 7-11. I have read the other Reviewers comments on this, and your response, and I think you should put more of your response text into the main text here.

**AC:** We added more information on why we chose this GM coefficient in section 2.1: "We had two reasons for keeping this rather low GM coefficient compared to what is used by e.g. Marshall et al. (2017). First, we want to be consistent with previous MPI-ESM simulations, and second, the GM coefficient was used to tune the AMOC, which became too weak in the TP04 configuration with higher values (von Storch et al., 2016). We note that finding an optimal configuration is challenging and still an open issue, in particular for hybrid grids as our TP04, which is eddy-permitting so that eddies are partly resolved and partly parameterized. There are also strategies to completely switch off GM for such hybrid grids (Delworth et al., 2012)."

Furthermore, we have added more information on the diffusivity and non-local transport in KPP (section 4.5.1).
"We speculate that the non-local transport terms in KPP cause a more efficient convection than the enhanced wind-mixing parameterization of our PP scheme. The diffusivity in KPP is further enhanced as it also depends on the mixed layer depth, which reflects that boundary layer eddies become larger with deeper mixed layers."

**3. RC:** Page 5 line 27. Typo on "configurations"

**AC:** Corrected.

**4. RC:** Page 6. Lines 8-9. "…the wind speed is too weak…" "but not as weak as in …" ?

**AC:** Corrected.

**5. RC:** Line 22. Delete "for biases" Also, did Drews et al partly, or totally, remove the bias?

**AC:** We deleted "for biases" and added "… removed the bias … almost completely" to the sentence.

**6. RC:** Page 7, line 3. "due to resolved eddies and better mean flow"?

**AC:** We have corrected the sentence.

**7. RC:** Page 11, line 7. "24km wide" – how does this compare with the lower resolution, and real bathymetry (approximately, no need to be too precise).

**AC:** We added information about the real bathymetry and the sill depth in our model configurations:
" The reason for this major improvement is the better resolved bathymetry of the Strait of Gibraltar, which is 12km wide and has a sill depth of ~ 300m in the present day real world. In the two ocean configurations discussed in this paper this Strait is about 24km wide with a shallowest sill depth of about 230m in the TP6M grid, compared to about 54km and same sill depth in TP04 "

**8. RC:** Line 11. "the cold bias at 740m"

**AC:** We added this to the sentence.

**9. RC:** Page 12, line 22. Reword "the relative differences among the simulations are similar" to something more easy to understand. (I think I know what you mean.)

**AC:** We rephrased this sentence to:
"The results for the subtropical gyre of the North Pacific reveal a similar picture as in the North Atlantic with stronger transports in the KPP simulations. One exception is a markedly reduced gyre strength in ERpp"

**9. RC:** Page 13, final sentence. This is a bit speculative – I suggest move to Discussion or delete.

**AC:** We decided to delete this sentence.

**10. RC:** Page 14 line 27 "in general much less sea ice volume"

**AC:** We added "in general a much lower sea ice volume" to the sentence.

**11. RC:** Figs 10, 11, A9. There are modern estimates of observed MLD, e.g. Holte et al., MIMOC (PMEL) https://www.pmel.noaa.gov/mimoc/ etc, using data from ARGO. Although data sparcity of ARGO is still an issue in some locations, can you add spatial maps of these products to the figures?

**AC:** We have added the mixed layer depth from the Holte et al. (2017) monthly mean climatology (using the density threshold method (0.03 kg/m³)) to the figures. We now relate our results to the Argo mld.

**12. RC:** Page 16 line 31. According to the literature, ocean eddies act to restratify many areas of the Labrador Sea, leaving a small region of deep mixed layers possibly where eddies are not active. It might be interesting to add a plot of SSH variability to see if ERpp adds anything to the eddy-permitting regime.

**AC:** Instead of SSH variability, we calculated the eddy kinetic energy at 100m depth from all models. As expected, there is (almost) no eke in the models with TP04. Only ERpp resolves eddies in the Labrador Sea, so that we show only the eke from ERpp in an additional appendix figure (now Fig.A9). To better show how the eddies limit the northward extension of the convection area, we also added contour lines for the 500m and 1000m MLD.

We have added these results in section 4.5.1 and give reference to Rieck et al. (2019) for the interaction of deep convection and restratification by eddies in the Labrador Sea.

**13. RC:** Page 18, line 32. Low resolution MLD may only be 200-300m in the same regions in DuVivier et al etc.

**AC:** We have added a sentence to this: "Low-resolution models (e.g. 1°), however, simulate depths of only 200 to 300m (DuVivier et al., 2018)."

and later:
"Our reference simulation (HRpp) simulates mixed layers of only about 200 to 300m (Fig. A9a), which is in agreement with the results from (DuVivier et al., 2018)."

**14. RC:** Page 19, line 8. You could place Lee et al and Li and Lee references on this line: on the lines above I don't think Small et al 2014 discussed MLD, but you could put (R. J. Small, pers. Comm. 2019).

**AC:** We changed the two sentences accordingly.

**15. RC:** Line 26. " A possible explanation for this is that the volume transport…" ?

**AC:** We linked this and the following sentence as recommended.

**16. RC:** Page 22, line 4-5. The reference to ERpp is out of place as this paragraph is on atmosphere resolution.

**AC:** Indeed, with moved the sentence to section 5.1.

[revised manuscript text omitted]

* * *
[14]removed: a

[15]removed: and up to $600\,\mathrm{m}$

[16]removed: c

[17]removed: b

On the other hand, $XR_{kpp}$ (Fig. 10[..[18] ]e) simulates shallower mixed layers compared with $HR_{kpp}$. These shallower mixed layers result from the reduced wind stress of the T255 atmosphere by means of two processes: (1) less positive wind stress curl spins down the subpolar gyre, so that the slower cyclonic circulation reduces the isopycnal doming and the horizontal salt advection to the gyre centres (Tab. 4), leading to a more stratified surface layer; and (2) lower near-surface wind speeds reduce

5    the turbulent air-sea fluxes via the bulk formula and the surface friction velocity ($u_*$). Lesser heat fluxes in turn reduce directly the non-local fluxes of the KPP scheme in convection areas, and lower $u_*$ reduces the turbulent vertical velocity scales, which results in lower vertical diffusivities and viscosities.

Based on these results, increasing the atmospheric resolution reduces the mixed layer depths over the North Atlantic and the Nordic Seas, whereas KPP deepens them. By combining both, the T255 atmosphere and the KPP scheme, the above effects

10   compensate each other ($XR_{kpp}$; Fig. 10[..[19] ]e). In contrast to $XR_{pp}$, where the convection ceases in the Labrador and GIN seas, the combination of T255 and KPP ($XR_{kpp}$) produces more realistic mixed layers depths even with reduced wind stress.

Overall, the KPP scheme modifies the large-scale circulation by simulating a stronger subpolar gyre, which in turn provides favourable conditions for deep convection in the Labrador Sea, Irminger Sea, and Nordic Seas. For this reason, $HR_{kpp}$ simulates enhanced deep convection compared with $HR_{pp}$, in particular in the Labrador and GIN Seas. In the Irminger Sea, mixed layer

15   depths of about 400 to 500 m are simulated by both $HR_{kpp}$ and $XR_{kpp}$, which is consistent with retrievals from observations (e.g. Pickart et al., 2003; Våge et al., 2008, 2011)[..[20] ], although too shallow compared to Argo (Fig. 10a). One explanation for these too shallow mixed layers are that even the T255 atmosphere is too coarse to fully simulate Greenland tip jets (e.g. Martin and Moore, 2007; DuVivier and Cassano, 2016; Gutjahr and Heinemann, 2018)[..[21] ]. The tip jets have a considerable impact on triggering deep convection in the Irminger Sea due to strong associated [..[22] ]wind stress curls driving the Irminger

20   Gyre, [..[23] ]and turbulent fluxes of heat and momentum removing the near-surface stratification. Because of the unresolved tip jets the mixed layer depth may be underestimated in winters with high tip jet activity.

The mixed layer depths in the Labrador Sea are nevertheless too deep (excluding $XR_{pp}$). A possible explanation is the neglect of tidal mixing in MPI-ESM1.2. As shown by Müller et al. (2010), tidal mixing improves the recirculation of the Labrador Current. By entraining more freshwater into the surface layer of the Labrador Sea, it becomes more stratified which in turn

25   reduces deep convection. Another shortcoming is probably insufficient eddy activity in the Labrador Sea so that too little freshwater is transported from the West Greenland Current into the interior of the Labrador Sea (e.g. Eden and Böning, 2002; Kawasaki and Hasumi, 2014).

In $ER_{pp}$ (Fig. 10f) the mixed layer depths are to a large extent similar to our reference simulation $HR_{pp}$. However, [..[24] ]the convection centre in the Labrador Sea is confined to a more southeastern area with deeper mixed layers in $ER_{pp}$. This is due

30   to resolved eddies, in particular Irminger Rings, that flatten the isopycnals thereby limiting the northward extent of the
* * *
[18]removed: d

[19]removed: d

[20]removed: . However,

[21]removed: , which

[22]removed: turbulent heat and momomentum fluxes

[23]removed: so that

[24]removed: in $ER_{pp}$

[revised manuscript text omitted]